# TCD-Arena: Assessing Robustness of Time Series Causal Discovery Methods Against Assumption Violations

**Gideon Stein, Niklas Penzel, Tristan Piater, Joachim Denzler**
Computer Vision Group Jena
Friedrich Schiller University Jena
Jena, Thuringia, 07743, DE
`gideon.stein@uni-jena.de`

## Abstract

Causal Discovery (CD) is a powerful framework for scientific inquiry. Yet, its practical adoption is hindered by a reliance on strong, often unverifiable assumptions and a lack of robust performance assessment. To address these limitations and advance empirical CD evaluation, we present **TCD-Arena**, a modularized, highly customizable, and extendable testing kit to assess the robustness of time series CD algorithms against stepwise more severe assumption violations. For demonstration, we conduct an extensive empirical study comprising around 30 million individual CD attempts and reveal nuanced robustness profiles for 33 distinct assumption violations. Further, we investigate CD ensembles and find that they have the potential to improve general robustness, which has implications for real-world applications. With this, we strive to ultimately facilitate the development of CD methods that are reliable for a diverse range of synthetic and potentially real-world data conditions.

## 1 Introduction

Causal Discovery (CD) holds great potential for addressing scientific hypotheses in fields where randomized control trials are difficult or impossible Glymour et al. (2019). Despite this promise, the widespread adoption of CD methods by practitioners remains limited. Recent works (Brouillard et al., 2024; Yi et al., 2025; Faller et al., 2024) attribute this to mainly two key factors: First, existing CD methods often rely on strong, idealized assumptions (e.g., no hidden confounders or stationarity) that are difficult to validate or, in real-world scenarios, simply unverifiable, even if they underpin theoretical guarantees. Second, empirical evaluations of CD methods predominantly use idealized synthetic data, which can overestimate performance and offer limited insight into robustness under imperfect but realistic conditions. Consequently, practitioners hesitate to adopt CD methods where their output reliability is limited (Kaiser & Sipos, 2021; Nastl & Hardt, 2024; Poinsot et al., 2025). To overcome this issue, there has been a recent push towards more benchmarking as it is the de facto golden standard in Machine Learning (Neal et al., 2023; Stein et al., 2024a; Wang, 2024; Mogensen et al., 2024; Herdeanu et al., 2025). However, the scarcity of real-world datasets with known causal ground truth continues to hinder a full reliance on empirical validation of CD methods. As a possible alternative to real-world benchmarks, recent studies investigate CD performance when specific assumptions are violated (Montagna et al., 2023a; Yi et al., 2025; Ferdous et al., 2025). Furthermore, the robustness of CD methods to hyperparameter selection has recently been highlighted (Machlanski et al., 2024). Building on these emerging efforts in empirical evaluation and aiming to unify them, we present **TCD-Arena**, a modularized, fully customizable testing kit to assess CD robustness against assumption violations. Next to an unprecedented scale, TCD-Arena focuses on three so far sporadically addressed aspects: **(1) temporal data**, that introduces additional challenges and opportunities; **(2) stepwise violation intensities**, crucial for capturing nuanced performance degradation rather than binary pass/fail outcomes; and **(3) a focus on violations often encountered in real-world settings.** In this paper, we demonstrate the benefits of TCD-Arena by conducting an extensive empirical study on the robustness of CD algorithms. Specifically, we evaluate ten CD algorithms that cover all four common CD archetypes Assaad et al. (2022b). We evaluate these

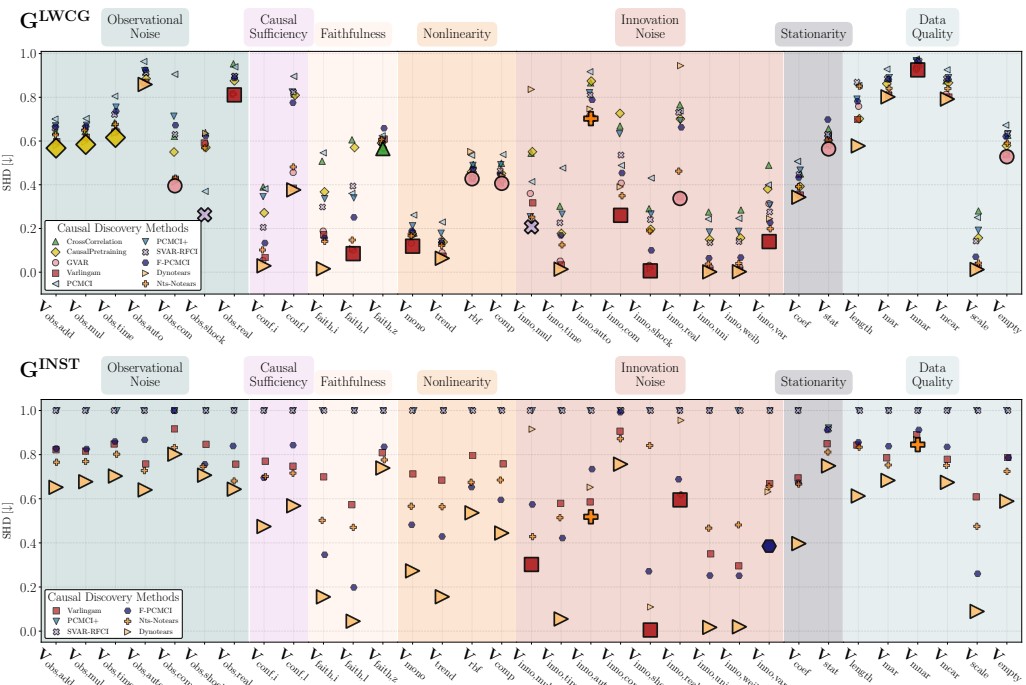

Figure 1: Robustness profile for $G^{\text{LWCG}}$ and $G^{\text{INST}}$ and for ten Causal Discovery algorithms against a multitude of stepwise more severe assumption violations. We measure robustness as the average normalized SHD over various data regimes and violation levels. See Fig. 17 for $G^{\text{LSG}}$.

algorithms across 33 real-world-inspired, partly semi-synthetic assumption violations, each scaled in intensity. By performing around 30 million individual CD attempts, we find that different methods differ in their ability to handle assumption violations. Additionally, we investigate hyperparameter sensitivities with respect to robustness and model misspecification, two aspects we believe are critical for applications to novel real-world data. Further, we investigate ensembles of CD methods, which have received little attention in the literature, and conclude that they can improve general robustness. Standing with Poinsot et al. (2025) and recognizing the pressing need for more nuanced CD evaluation, we attempt to further establish robustness analysis as an alternative to traditional benchmarking and theoretical analysis. With this, we hope to ultimately facilitate the development of CD methods that are reliably applicable to real-world data. In summary, this paper makes the following contributions:

1. The introduction of TCD-Arena, an open-source, customizable toolkit for quantifying the robustness of CD across diverse time series data that fosters long-term comparability.

2. A large-scale empirical study that evaluates the robustness of ten time series CD methods against 33 stepwise intensified, partly semi-synthetic, assumption violations.

3. An investigation into ensembling CD methods with respect to violation robustness.

## 2 BACKGROUND AND THEORETICAL PRELIMINARIES

To ground our empirical investigation, we begin by selectively revisiting the relevant theoretical background. Let $\boldsymbol{X} \in \mathbb{R}^{D \times T}$ be a $D$-variate time series comprising $T$ samples from $D$ interacting variables, generated by an unknown underlying causal process. The objective of time series Causal Discovery (CD) is to infer the causal relationships among $D$ variables from the observed data $\boldsymbol{X}$. These relationships are commonly represented as a Structural Causal Model (SCM) (Peters et al., 2017). For each variable $X_{i,t}$, the SCM contains assignments of the form:

$$X_{i,t} = f_i\left(\text{Pa}(X_{i,t}), \epsilon_{i,t}\right), \tag{1}$$

where $\text{Pa}(X_{i,t})$ is the set of direct causal parents of $X_{i,t}$, $f_i$ is a causal mechanism, and $\epsilon_{i,t}$ is independent innovation noise. The set of assignments within an SCM defines a directed graph $G = (V, E)$. In this work, we distinguish between contemporaneous ($X_{j,t}$ for $j \neq i$) and lagged effects ($X_{j,t-k}$ for $k > 0$) by evaluating the recovery of the following three distinct graph structures: First, the lagged window causal graph ($G^{\text{LWCG}}$) provides lag-specific causal dependencies up to a maximum lag $L$. Here $V$ includes each variable at time step $t$ and at all past lags: $V = \{X_{i,t-l} \mid i \in \{1, \ldots, D\}, l \in \{1, \ldots, L\}\}$. A directed edge $X_{j,t-l} \rightarrow X_{i,t}$ exists in $G^{\text{LWCG}}$ if $X_{j,t-l}$ is in $\text{Pa}(X_{i,t})$. Note that in this representation, edges only connect past variables to variables at step $t$. Second, the lagged summary graph ($G^{\text{LSG}}$) summarizes time-lagged relationships. Its vertices are defined as $V = \{X_1, \ldots, X_D\}$. A directed edge $X_j \rightarrow X_i$ exists in $G^{\text{LSG}}$ if $X_{j,t-l} \in \text{Pa}(X_{i,t})$ for at least one $l > 0$. Third, the instantaneous graph ($G^{\text{INST}}$) captures only contemporaneous relationships. It is a directed graph with vertices $V = \{X_1, \ldots, X_D\}$. A directed edge $X_j \rightarrow X_i$ exists in $G^{\text{INST}}$ iff $X_{j,t}$ is in $\text{Pa}(X_{i,t})$. While this framework formalizes causal interactions, the identifiability of any $G$ from $\boldsymbol{X}$ requires a number of assumptions about Eq. (1). For time series, the direction of time Bauer et al. (2016) (effects cannot precede causes) aids with identifying lagged relationships ($G^{\text{LWCG}}$ and $G^{\text{LSG}}$), generally requiring fewer restrictive assumptions. However, recovering $G^{\text{INST}}$ is more challenging (and is not addressed by all CD methods), resembling causal discovery from i.i.d. data. We refer to (Pearl, 2009; Peters et al., 2017) for a comprehensive introduction as well as to (Spirtes et al., 2001) concerning constraint-based algorithms.

Despite the fact that many specific assumptions underpinning CD methods can be relaxed individually, a core set of strong, partly implicit, assumptions generally remains necessary to guarantee the identifiability of any SCM, as the causal hierarchy levels almost never collapse Bareinboim et al. (2022). Further, even if these assumptions can be perfectly met in synthetic data, real-world data will often violate many of them, which can degrade the performance of CD algorithms Kaiser & Sipos (2021); Nastl & Hardt (2024). On top, many assumptions are not verifiable without having access to the full SCM, e.g., the appropriate conditional-independence test Shah & Peters (2020). For widespread practical adoption, it is therefore essential to assess method performance under suboptimal conditions Poinsot et al. (2025). In response to these challenges, and mirroring trends in other machine learning domains, there is a growing emphasis on developing real-world Stein et al. (2024a); Mogensen et al. (2024) as well as semi-synthetic Cheng et al. (2023); Herdeanu et al. (2025) benchmarks, or kits such as Muñoz-Marí et al. (2020) or Zhou et al. (2024) for CD. Notably, while real-world and semi-synthetic datasets are essential, they come with a tradeoff of often having an at least partly unknown data generation process (outside of the causal ground-truth), which leaves room for extensive synthetic benchmarks Poinsot et al. (2025) Furthermore, as aggregating extensive real-world causal ground truth is notoriously challenging, alternative approaches have been introduced to enable the empirical evaluation of CD method performance. For instance, Schkoda et al. (2024) proposes leave-one-out cross-validation to assess the predictive performance of CD algorithms. Faltenbacher et al. (2025) introduces an internal consistency score, specifically for PC-style algorithms Spirtes et al. (2001), to help validate the inferred graph. Moreover, Machlanski et al. (2024) advocates evaluating hyperparameter sensitivity, which has implications for method selection in practical applications. Closely related to our work, Yi et al. (2025) and Montagna et al. (2023a) test the performance of i.i.d. sample-based CD methods for fully violated assumptions. Further, Ferdous et al. (2025) provides an insightful study that investigates the impact of five different real-world complications on the performance of CD methods. Notably, robustness has also been explored in a general statistical context as well Rasch & Guiard (2004); Zimmerman (2014). Nevertheless, the impact of such violations remains under-investigated, particularly for time series data and across differences in violation severity. Finally, ensembling strategies, as a practical tool to improve robustness in other machine learning domains Arpit et al. (2022); Mienye & Sun (2022), remain largely unexplored in the CD literature. While recent work has investigated ensembling over variable subsets to recover large graphs Wu et al. (2024), the potential to improve resilience against assumption violations has not yet been studied.

## 3 STEPWISE INCREASING ASSUMPTION VIOLATIONS

While the CD literature explores relaxing certain assumptions, identifying a causal structure from $\boldsymbol{X}$ alone typically relies on a core set of assumptions to guarantee identifiability. Although prior work partially analyzes resilience to binary assumption violations Yi et al. (2025); Montagna et al.

(a) In our experiments, the sources of randomness for the observational noise variables $\zeta_{i,t}$ are standard normally distributed $\left(\mathcal{N}(0,1)\right)$ random variables $\eta_{i,t}$ and $\eta_t$, which are consequently influenced by various factors, e.g., the signal strength ($\mathbf{V}_{\text{obs,mul}}$). Both $\alpha$ and $\beta$ denote hyperparameters (details in **Apx. B.1**). For $\mathbf{V}_{\text{obs,real}}$ is entirely dependent on an external (possibly processed) time series.

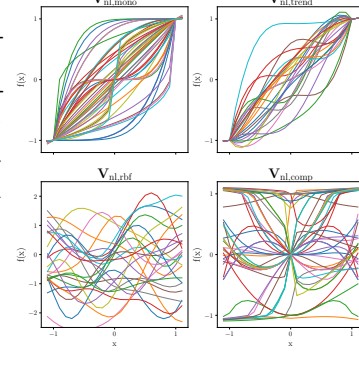

| Violation | Definition of $\zeta_{i,t}$ | Depends On |
|---|---|---|
| $\mathbf{V}_{\text{obs,add}}$ | $\zeta_{i,t} = \eta_{i,t}$ | — |
| $\mathbf{V}_{\text{obs,mul}}$ | $\zeta_{i,t} = X_{i,t} \cdot \eta_{i,t}$ | the signal $X_{i,t}$ |
| $\mathbf{V}_{\text{obs,time}}$ | $\zeta_{i,t} = \eta_{i,t} \cdot (1 + \alpha t) \cdot \sin(2\pi t/\beta)$ | time step $t$ |
| $\mathbf{V}_{\text{obs,auto}}$ | $\zeta_{i,t} = \alpha \cdot \zeta_{i,t-1} + (1-\alpha) \cdot \eta_{i,t}$ | autoregressive |
| $\mathbf{V}_{\text{obs,com}}$ | $\zeta_{i,t} = \eta_t \quad \text{for all } i \text{ at each } t$ | — |
| $\mathbf{V}_{\text{obs,shock}}$ | $\zeta_{i,t} \sim \begin{cases} S & \text{with prob. } p_{\text{shock}}, \\ 0 & \text{else.} \end{cases}$ | fixed scalar $S$, shock prob. $p_{\text{shock}}$ |
| $\mathbf{V}_{\text{obs,real}}$ | $z_{i,t}$ | exog. real-world TS $Z^{I,T}$ |

(b) Note that the coefficients $A_{i,d,l}$ can be negative, resulting in negative trends.

Figure 2: Details for violation types $\mathbf{V}_{\text{obs}}$ and $\mathbf{V}_{\text{nl}}$. Left: Observational noise violations. Right: Functional distributions that we deploy to sample $f_{i,d,l}$ used in Eq. (2).

(2023a); Ferdous et al. (2025), a pertinent question arises: **How robust are certain CD methods against different severities of assumption violation?** This question is critical in applied settings. For instance, the mere existence of observational noise is less informative than understanding its robustness to its presence. Addressing this requires a framework for varying the severity of these violations. In this study, we introduce TCD-Arena for this purpose. In total, we implement 33 distinct assumption violations, each parameterized to allow for a stepwise increase of its severity. We individually describe these in the sections to come. Generally, we focus on covering commonly made assumptions Runge (2018) along with real-world complications that are practically relevant. We also focus on providing multiple alternative implementations of a specific violation if applicable. Further, we restrict our exploration to the following restriction for Eq. (1):

$$X_{i,t} = \sum_{d=1}^{D} \sum_{l=0}^{L} A_{i,d,l} \cdot f_{i,d,l}(X_{d,t-l}) + \epsilon_{t,i}, \tag{2}$$

where $A$ specifies a coefficient matrix and $f_{i,d,l}$ an edge-specific univariate function and $\epsilon_{t,i}$ independent innovation noise. Crucially, any non-zero element in $A$ denotes a corresponding edge in $G$. Further, for violations that do not concern the causal mechanisms, $f_{i,d,l}$ is the identity function, and all interactions are linear. To help with clarity, we mark individual violations as $\mathbf{V}_{\text{type}}$.

Notably, we keep the following violation descriptions brief and include a summary table, graphical depictions, specific violation step sizes, and detailed design choices for each violation in **Apx. A** and **Apx. B**. Finally, while the experiments in this paper focus on assessing the effect of each implemented violation individually, TCD-Arena allows for freely combining all implemented violations to create highly customized data scenarios. We further discuss this in **Apx. D.6**.

**Observational Noise ($\mathbf{V}_{\text{obs}}$)** Many theoretical guarantees in causal discovery assume noise-free measurements, despite the fact that measurement errors are practically unavoidable and can introduce discrepancies that distort true causal relationships Scheines & Ramsey (2016). In an additive form, observation noise can be defined as: $\hat{X}_{i,t} = X_{i,t} + \zeta_{i,t}$, where $\zeta_{i,t}$ denotes observational noise. While standard independent additive noise ($\mathbf{V}_{\text{obs,add}}$) is prevalent, other noise types can occur depending on the measurement process. To this end, we investigate the impact of five additional types of observational noise structures on CD. **Fig. 2a** contains a concrete list. Additionally, details, hyperparameters, and discussions can be found in **Apx. B.1**. In particular, we include multiplicative, signal-dependent noise ($\mathbf{V}_{\text{obs,mul}}$), with real-world examples such as temperature sensors that exhibit lower precision at higher values Bentley (1984) and speckle noise in image processing Liu et al. (2014). We include time-dependent noise ($\mathbf{V}_{\text{obs,time}}$) to simulate cycles or linear sensor drift. Further, we model autoregressive noise structures ($\mathbf{V}_{\text{obs,auto}}$), i.e., disturbances in the measurements that persist for multiple time steps. Similarly, we include common observational noise ($\mathbf{V}_{\text{obs,com}}$), which affects multiple variables simultaneously, e.g., due to weather events. Next, we include shock noise

($\mathbf{V}_{\text{obs,shock}}$) to model infrequent events such as measurement failures. Finally, we include observational noise, directly consisting of processed exogenous real-world time series that can be chosen freely ($\mathbf{V}_{\text{obs,real}}$). To systematically vary the level of intensity for any of these observational noise structures, we adjust the signal power of $\zeta$ to control the corresponding Signal-to-Noise Ratio (SNR) with respect to the data $\boldsymbol{X}$. To isolate the influence of the noise structure, we use the same decreasing SNR levels for all observational noise violations in **Fig. 2a**.

**Causal Sufficiency ($\mathbf{V}_{\text{conf}}$)**   Causal sufficiency posits that for any pair of observed variables $X_i$ and $X_j$, there are no unobserved common causes (hidden confounders). That is, there is no unmeasured variable $U$ such that $U \rightarrow X_i$ and $U \rightarrow X_j$. Such latent confounders can induce spurious correlations among observed variables, potentially leading to incorrect or misleading causal inferences. While some advanced methods aim to address specific types of confounding Trifunov et al. (2019); Chen et al. (2024); Li & Liu (2024), the presence of unmeasured confounders remains a major practical challenge, as it is rarely feasible to measure all relevant variables in complex systems. To simulate varying degrees of confounding and assess its impact, we employ two distinct strategies targeting lagged and contemporaneous confounding: First, concerning instantaneous confounding $\mathbf{V}_{\text{conf,inst}}$, we introduce a set of $N$ exogenous variables $\mathcal{Z} = Z_1, \dots, Z_N$, where each $Z_{n,t} \sim \mathcal{N}(0,1)$. These exogenous variables are not causally influenced by any variable in $\boldsymbol{X}$ but can act as common causes to multiple variables in step $t$. The severity of this type of confounding is controlled by progressively increasing the probability that an observed variable $X_{i,t}$ becomes dependent on any of the exogenous variables $Z_n$. This, in turn, increases the probability that two variables in X share a hidden parent at $t$. Second, for lagged confounding, we introduce an additional variable, $X_C$, designated as the potential confounder ($\mathbf{V}_{\text{conf,lag}}$). This variable $X_C$ is allowed to causally influence, and can be influenced by, other observed variables $X_i$ with lagged effects up to a specified maximum lag $L$. The severity of confounding is controlled by stepwise increasing the probability that $X_C$ is in the parent set of any other variable in $\boldsymbol{X}$. After sampling, the time series, $X_C$ is removed from the observed data $\mathbf{X}$, rendering it a hidden confounder. We include an in-depth description and the concrete stepwise probabilities for both types in **Apx. B.2**.

**Faithfulness ($\mathbf{V}_{\text{faith}}$)**   Faithfulness asserts that all conditional independencies observed in the data are precisely those implied by $d$-separation of the DAG $G$ Scheines (1997). Formally, for any disjoint sets of variables $\mathscr{X}, \mathscr{Y}, \mathscr{Z}$, $\mathscr{X} \perp\!\!\!\perp \mathscr{Y} | \mathscr{Z}$ holds in the observed distribution iff $\mathscr{X}$ and $\mathscr{Y}$ are d-separated by $\mathscr{Z}$ in $G$. Violations of faithfulness can lead to indistinguishable causal structures as they generate no dependencies in $\boldsymbol{X}$. Some works that examine this assumption and offer alternatives are (Zhang & Spirtes, 2008; Andersen, 2013; Lin & Zhang, 2020; Ng et al., 2021). Typically, unfaithfulness is implemented through causal structures like $X_{j,t} \rightarrow X_{i,t} \leftarrow X_{k,t} \leftarrow X_{j,t}$, where effects from $X_{j,t}$ to $X_{i,t}$ cancel out through appropriate parameter configurations in $A$. We implement this case for instantaneous effects ($\mathbf{V}_{\text{faith,inst}}$) as well as a lagged structure of the form $X_{j,t-2} \rightarrow X_{i,t} \leftarrow X_{k,t-1} \leftarrow X_{j,t-2}$ ($\mathbf{V}_{\text{faith,lag}}$). Furthermore, we implement an alternative approach to generate unfaithful connections by scaling down the effect sizes of causal relationships to near zero ($\mathbf{V}_{\text{faith,zero}}$), making effects harder to detect statistically. Notably, this case is related to the notion of $\lambda$ strong faithfulness Zhang & Spirtes (2012). For the first two cases, we stepwise scale the intensities of both violations by updating the parameter configurations in $A$ to approach full path cancellation. For the third case, we stepwise reduce the range of the parameter distribution for values in $A$. Further, as this is only one case of violating faithfulness Montagna et al. (2023b), we include a discussion of this design choice, along with visual examples, in **Apx. B.3**.

**Functional Assumptions ($\mathbf{V}_{\text{nl}}$)**   While the general SCM in Eq. (2) is agnostic to the functional forms $f_{i,d,l}$ between variables $X_{d,t-l} \rightarrow X_{i,t}$, many discovery algorithms assume specific interactions, e.g., linear-additive relationships $X_t = \sum_{l=1}^{L} A_l \cdot X_{t-l} + \epsilon_t$ Hyvärinen et al. (2010); Pamfil et al. (2020). In real-world systems, such assumptions are often violated or are only approximations. Thus, it is crucial to study the consequences of corresponding violations, besides attempting to relax them Runge et al. (2019); Monti et al. (2020); Wu et al. (2022). To better emulate the variety found in practical scenarios and simulate data diversity, we employ a range of function-generation techniques with distinct characteristics. In particular, we sample individual univariate functions $f_{i,d,l}$ from four distinct distributions: (1) Monotonic nonlinear functions ($\mathbf{V}_{\text{nl, mono}}$), (2) Non-monotonic functions with a linear trend ($\mathbf{V}_{\text{nl,trend}}$) (3) A Gaussian process with an RBF kernel ($\mathbf{V}_{\text{nl,rbf}}$) following related robustness studies Montagna et al. (2023a); Yi et al. (2025), and (4) Random combinations of a set

of base functions, e.g., $\sin(\cdot)$ or $e^{(\cdot)}$ ($\mathbf{V}_{\text{nl,comp}}$). Example functions are depicted in **Fig. 2b**, and we describe the exact distributions from which we sample in **Apx. B.4**. To stepwise increase the violations, we rely on two distinct procedures. First, for $\mathbf{V}_{\text{nl,mono}}$ and $\mathbf{V}_{\text{nl,trend}}$, we stepwise adapt the functional distributions such that sampled functions become on average increasingly nonlinear Emancipator & Kroll (1993) (**Apx. B.4**). We sample all interactions $f_{i,d,l}$ in Eq. (2) from the corresponding distributions. Second, for $\mathbf{V}_{\text{nl,rbf}}$ and $\mathbf{V}_{\text{nl,comp}}$, we stepwise increase the probability of any $f_{i,d,l}$ to be drawn from the nonlinear distribution instead of being equal to the identity $f_{i,d,l}(\cdot) = \text{id}(\cdot)$.

**Independent Innovation Noise ($\mathbf{V}_{\text{inno}}$)**   Independent additive innovation noise ($\epsilon_{i,t}$ in Eq. (2)) is crucial for CD, as it ensures dependencies are attributed to causal links, not shared noise. However, this assumption is often violated in practice, as noise can incorporate unmeasured, dependent effects, and its true distribution is typically unknown or is fully deterministic Li et al. (2024). To evaluate how alternative innovation noise distributions might affect the performance of CD algorithms, we deploy the same six noise structures that we use for observational noise, i.e., $\mathbf{V}_{\text{inno,mul}}, \mathbf{V}_{\text{inno,auto}}, \mathbf{V}_{\text{inno,com}}, \mathbf{V}_{\text{inno,time}}, \mathbf{V}_{\text{inno,shock}}, \mathbf{V}_{\text{inno,real}}$. However, for innovation noise scaling, the SNR (compared to the observation noise) is nontrivial as it is part of the signal itself. Therefore, we control the violations by blending standard normal noise with each of the five noise terms to stepwise move away from independent additive conditions (details in **Apx. B.5**). Notably, autoregressive $\mathbf{V}_{\text{inno,auto}}$, common $\mathbf{V}_{\text{inno,com}}$ and likely real-world innovation noise ($\mathbf{V}_{\text{inno,com}}$) fundamentally violate the Markov condition Peters et al. (2017). Additionally, since some identifiability guarantees assume non-Gaussian noise (Shimizu et al., 2006), we test the effect of stepwise moving towards a non-Gaussian distribution. Specifically, we simulate this by starting from a Gaussian distribution (see **Apx. B.5**) and progressively shifting towards either a uniform ($\mathbf{V}_{\text{inno,uni}}$) or a Weibull distribution ($\mathbf{V}_{\text{inno,weib}}$). Finally, as some works assume equal noise variances, e.g., Peters & Bühlmann (2014), we implement a strategy to move away from this assumption ($\mathbf{V}_{\text{inno,var}}$). For this, we draw $\epsilon_{i,t}$ from $\mathcal{N}(0, \sigma_i^2)$, where the variance $\sigma_i^2$ is individually sampled for each $X_i$. We then increase the corresponding ranges stepwise.

**Stationarity ($\mathbf{V}_{\text{coef}}, \mathbf{V}_{\text{stat}}$)**   CD methods typically aim to uncover a single $G$ from observations $\boldsymbol{X}$. Hence, a common assumption is that the SCM remains unchanged, i.e, stationary, over time or across regions. However, in many real-world scenarios, causal relationships can be heterogeneous across different populations or evolve over time Nastl & Hardt (2024). Here, works such as Huang et al. (2020); Günther et al. (2024); Ahmad et al. (2024) attempt to identify causal relationships in systems where parts of the SCM are changing. We include two violation cases to simulate violation of stationarity. First, we keep the causal skeleton (the nonzero elements in $A$) fixed and redraw the coefficients multiple times. We scale the violation by increasing the maximum allowed distance from the previous value. Second, we fully resample $A$ multiple times throughout the sampling process, violating the principle of causal consistency. Here, any element in $A$ that is non-zero at any $t$ is treated as a causal effect. To stepwise scale the violation, we increase the number of times we resample $A$ when generating $\boldsymbol{X}$ (see **Apx. B.6** for more details).

**Sufficient Sample Sizes ($\mathbf{V}_{\text{length}}$)**   Causal discovery algorithms necessitate a sufficient sample size to reliably detect patterns and estimate relationships Shen et al. (2020); Castelletti & Consonni (2024). For example, statistical tests used to identify conditional independencies may lack power with limited data Spirtes & Zhang (2016). To the best of our knowledge, no work has yet conducted an extensive study on the relationship between CD performance and sample size. To remedy this, we test the effect of a stepwise reduction of the length of the sampled time series $\boldsymbol{X}$ ($\mathbf{V}_{\text{length}}$, **Apx. B.7**).

**Data Quality ($\mathbf{V}_{\text{mcar}}, \mathbf{V}_{\text{mar}}, \mathbf{V}_{\text{mnar}}, \mathbf{V}_{\text{empty}}$)**   To simulate measurement disturbances beyond observational noise, we introduce four types of quality degradations for $\boldsymbol{X}$. First, for missing data ($\mathbf{V}_{\text{mcar}}, \mathbf{V}_{\text{mar}}, \mathbf{V}_{\text{mnar}}$), we remove a number of samples "completely at random", "at random" (depending on an external real-world time series), or "not at random" (depending of the values of the time series itself) Heitjan & Basu (1996) and fill the resulting NaNs via linear interpolation, a common approach for practitioners. To increase the severity of the violation, we stepwise increase the total amount of missing values. Second, we model sensor failures ($\mathbf{V}_{\text{empty}}$) by setting all parent sets to $\varnothing$ for periods, simulating false, zero-information measurements. We then stepwise increase the length of these periods to scale the effect (see **Apx. B.8** for more details).

**Data Scaling ($V_{scale}$)**    Recent works show that synthetically generated data can introduce artifacts in the causal order, which can be exploited by CD methods Reisach et al. (2021); Kaiser & Sipos (2021); Ormaniec et al. (2025). By rescaling $X$, these artifacts can be partly removed. To investigate how robust methods are against scaling, we allow for a stepwise scaling of the generated time series. In particular, we blend the original time series with its standardized version, a transformation that is reported to affect CD performance in Reisach et al. (2021); Kaiser & Sipos (2021) (**Apx. B.9**).

**Acyclicity and Sampling Rate**    Finally, we comment on the assumption of acyclicity, which we deliberately do not address in this work. While central to many algorithms, this assumption can be violated in two primary ways: by genuine feedback loops inherent to the system (e.g., in differential equations) or by apparent cycles that emerge as artifacts of temporal aggregation. The latter occurs when a coarse measurement resolution makes a lagged effect appear as a contemporaneous, bidirectional relationship Runge (2018). While the first case is connected to a fundamentally different data generation process, the second creates a fundamental ambiguity: A system may be acyclic at one temporal scale but cyclic at another, making a single ground truth non-trivial to define. Additionally, as we find it problematic to define a meaningful stepwise increase in acyclicity, we refrain from investigating this assumption in this work.

## 4    EXPERIMENTS

To evaluate the robustness of CD methods and to showcase the functionality of TCD-Arena, we conducted a large empirical study using synthetic data across all previously described violations. For each violation, we systematically increase its intensity across five discrete levels and test the ability of CD methods to recover $G^{LWCG}$, $G^{INST}$, and $G^{LSG}$. The severity levels for each violation were individually calibrated to span from negligible impact to a level at which the baseline method's performance (see below) degrades to chance (if the violation type allows it). A complete list of the configurations and further discussion of this methodology are provided in **Apx. A.2**. Our experiments covered a range of data conditions to ensure the generalizability of our findings. We vary the number of time steps ($T \in \{250, 1000\}$) and the number of variables $D$ (we call the two configurations *small* and *big*), together with the maximum causal lag $L$ in the true SCM $\big((D, L) \in \{(5, 3), (7, 4)\}\big)$. For each setting, we generated datasets with both sparse and dense causal graphs, and both with and without instantaneous effects. This resulted in 16 distinct data-generating conditions, which we call "data regimes". Generally, we use standard normal innovation noise ($\epsilon_{i,t} \sim \mathcal{N}(0, 1)$, eq. (2)). For each violation type, severity level, and data regime, we generated 100 independent structural causal models (SCMs) and a corresponding time series. In total, the evaluation for each violation type comprises 8,000 unique time series instances. Further details on the data-generating process are available in **Apx. C.1**. Notably, while our experiments focus on assessing each violation individually, TCD-Arena allows for a free combination of all violations to generate multi-violation scenarios. We demonstrate this capability shortly in **Apx. D.6**. Finally, to encourage practitioners and authors of novel methods to build and extend on this protocol, we include a guide on this topic in **Apx. C.7**.

Regarding CD methods, we conduct experiments on 10 strategies, including the direct Cross Correlation matrix, which serves as a baseline for predicting causal relationships. Further, we include the following nine approaches: We leverage Granger-causal ideas and deploy a vector autoregressive model Granger (1969) where we either rely on p-values or absolute model coefficients to predict causal relationships (GVAR), Varlingam Hyvärinen et al. (2010), PCMCI and PCMCI+ Runge et al. (2019), Dynotears (Pamfil et al., 2020), NTS-NOTears (Sun et al., 2023), FPCMCI Castri et al. (2023), SVARRFCI Gerhardus & Runge (2021) and CausalPretraining (Stein et al., 2024b). With this, we cover all common CD paradigms Assaad et al. (2022b). Note, we discuss possible reasons for method exclusion in **Apx. C.4**. Under ideal linear conditions with no assumption violations, all included methods are capable of recovering $G^{LWCG}$ from Eq. (2). Additional details and a list of assumptions for each method are provided in **Apx. C.2**. Notably, because all methods were evaluated on the same datasets, any potential theoretical non-identifiability issues affect all algorithms equally. This ensures a fair comparison of their relative robustness. Details on reproducibility can be found in **Apx. C.6**.

To ensure a fair performance comparison, we adopt an evaluation protocol with three key components. First, to mitigate bias from suboptimal hyperparameters, we perform a hyperparameter search for each method (**Apx. C.3**). Second, we selected the minimum normalized Structured-Hamming Distance as our primary, threshold-agnostic performance metric :

$$\text{SHD} = \min_{\tau \in \mathcal{T}} \left( \frac{\text{SHD}(G, \hat{G}_\tau)}{|A^G|} \right). \tag{3}$$

Here, $\tau$ specifies an arbitrary decision boundary. This allows us to evaluate how well a method distinguishes causal links from non-dependence, without the need to select a specific decision threshold. Notably, depending on the graph structure, $G$ and $A$ correspond to either instantaneous, lagged, or summary effects. For completeness, we also report several alternative metrics (AUROC, F1, Accuracy) in **Apx. D.1**. Third, to measure the robustness of a method's specific hyperparameter configuration with respect to a violation $\mathbf{V}_{\text{type}}$, we average the SHD scores across all data regimes and violation levels. This aggregation accounts for potential variations in the optimal decision boundary across different experimental conditions. Also, compared with Ferdous et al. (2025), this protocol, although computationally heavier, enables a more reliable estimation of robustness, as the considered data distributions exhibit greater variability. We include a visual overview of this experimental protocol in **Apx. A.3**.

**General Robustness**  First, we illustrate the robustness of each method against individual violation types for each graph structure in **Fig. 1**. Furthermore, **Fig. 3a** summarizes the average robustness scores across all assessed violations. Here, we identify a single hyperparameter configuration for each CD method that maximizes average robustness across all violations. We believe this better reflects a practical scenario than optimizing for each violation individually, a protocol used by Montagna et al. (2023a); Yi et al. (2025). For complementary purposes, we also report individually optimized results and worst-case performance in **apx. D.1**. Given the discovery of lagged effects ($G^{\text{LWCG}}$), we find that Varlingam and Dynotears generally show the highest robustness, with Varlingam achieving the highest total robustness across all violations. Furthermore, we observe that constraint-based approaches (Especially PCMCI) tend to be less robust. Concerning uncovering $G^{\text{INST}}$ (**Fig. 1**), we find that continuous optimization-based methods (Dynotears and Nts-Notears) display the highest robustness. Notably, as PCMCI+ and SVARRFCI predict undirected graphs, their normalized SHD cannot be smaller than 1 as long as $A$ is constant. Therefore, additional metrics that punish False Positives less harshly should be considered (**Apx. D.1**) Furthermore, as our data generation process does not strictly enforce non-Gaussian, independent, additive noise, Varlingam's performance on $G^{\text{INST}}$ aligns with theoretical expectations. Its superior performance on $G^{\text{LWCG}}$ is consistent with this finding, as the non-Gaussian assumption is primarily required to identify instantaneous links rather than lagged effects. We, however, find it interesting that explicitly modeling instantaneous links facilitates the identification of lagged effects, outperforming GVAR despite it estimating the same model for those lagged components. Further, even in scenarios where we explicitly introduced non-Gaussian noise components (i.e., $\mathbf{V}_{\text{weib}}$ and $\mathbf{V}_{\text{uni}}$), VarLiNGAM did not exhibit improved robustness. This suggests that the innovation noise must diverge sufficiently from a Gaussian distribution to enable effective identification." Concerning uncovering $G^{\text{LSG}}$, we find that GVAR depicts the highest average robustness. Notably, this is consistent with our previously published results reported in Stein et al. (2024a), a large-scale real-world CD benchmark. Here, GVAR is also shown to be the most robust CD method for uncovering summary causal graphs, suggesting that TCD-Arena can capture the complexities of real-world applications.

**Model Misspecification**  Second, as we previously assumed a known maximum lag $L$, we further investigate the performance of all tested algorithms under two additional scenarios: (i) The model is allowed to search for causes up to a lag greater than the true maximum lag ($L \in \{3, 4\}$ while $L_{\text{model}} \in \{5, 6\}$). (ii) The model's search space is restricted to lags shorter than the true maximum lag ($L \in \{3, 4\}$ while $L_{\text{model}} \in \{1, 2\}$). We denote these cases with $\uparrow L$ and $\downarrow L$, and report the effect of these misspecifications in **Table 1** (additional visualizations in **Apx. D.2**). In the $\downarrow L$ condition, we observe a sharp decline in performance across all methods. We also find that GVAR becomes the most robust method in this regime. Conversely, performance in the $\uparrow L$ regime is much more stable. However, it is notable that GVAR's performance on $G^{\text{LSG}}$ degrades more significantly than Varlingam's, resulting in a reversal of their rankings. While synthetic results warrant caution, these observations suggest that, in practice, overestimating the maximum lag $L_{\text{model}}$ may be beneficial. Finally, CausalPretraining offers a distinct advantage: It does not require specifying $L_{\text{model}}$, as it is fixed to the maximum lag during training (3). This explains its superior robustness in the $\downarrow L$ regime but has limited implications for real-world applications.

(a) By ensembling different CD methods, general robustness can be improved. **\*** marks required knowledge about the underlying SCM. † marks methods that do not discover $G^{\text{INST}}$. We highlight superior performance with **green** 🟢 .

| Method | $G^{\text{LWCG}}$ | $G^{\text{INST}}$ | $G^{\text{LSG}}$ |
|---|---|---|---|
| CrossCorrelation | $.582_{\pm.23}$ | † | $.453_{\pm.20}$ |
| CausalPretraining | $.530_{\pm.25}$ | † | $.440_{\pm.23}$ |
| GVAR | $.424_{\pm.28}$ | † | $\mathbf{.330}_{\pm.23}$ |
| Varlingam | $\mathbf{.408}_{\pm.29}$ | $.692_{\pm.20}$ | $.334_{\pm.24}$ |
| PCMCI | $.601_{\pm.25}$ | † | $.447_{\pm.22}$ |
| PCMCI+ | $.539_{\pm.26}$ | $.998_{\pm.01}$ | $.405_{\pm.23}$ |
| SVAR-RFCI | $.517_{\pm.28}$ | $.997_{\pm.01}$ | $.407_{\pm.25}$ |
| F-PCMCI | $.499_{\pm.30}$ | $.657_{\pm.24}$ | $.434_{\pm.26}$ |
| Dynotears | $.445_{\pm.32}$ | $\mathbf{.515}_{\pm.28}$ | $.365_{\pm.27}$ |
| Nts-Notears | $.445_{\pm.28}$ | $.674_{\pm.13}$ | $.358_{\pm.24}$ |
| Ensemble$_{\text{Avg.}}$ | $.387_{\pm.30}$ | $.550_{\pm.23}$ | $.300_{\pm.24}$ |
| Ensemble$_{\text{Linear}}$ | $\mathbf{.362}_{\pm.20}$ | $.527_{\pm.19}$ | $\mathbf{.281}_{\pm.16}$ |
| Ensemble$_{\text{MLP}}$ | $.445_{\pm.06}$ | $.523_{\pm.20}$ | $.343_{\pm.15}$ |
| Ensemble$_{\text{Transformer}}$ | $.415_{\pm.20}$ | $.587_{\pm.16}$ | $.324_{\pm.16}$ |
| Ensemble$_{\text{Pareto}}$ | $.376^*_{\pm.29}$ | $\mathbf{.470}^*_{\pm.27}$ | $.296^*_{\pm.24}$ |

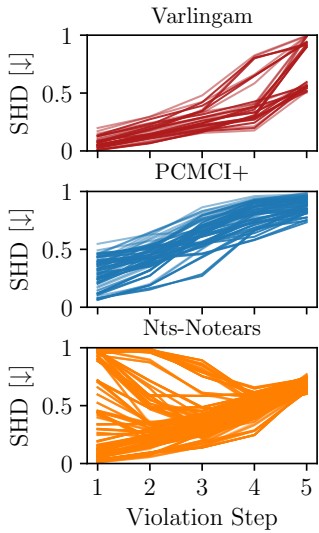

(b) Each curve depicts SHD changes of a particular hyperparameter configuration and a particular data regime for $\mathbf{V}_{\text{inno,com}}$.

Figure 3: Left: Average robustness of the best hyperparameter configuration per CD method and per ensemble for $G^{\text{LWCG}}$, $G^{\text{INST}}$, and $G^{\text{LSG}}$. Right: Hyperparameter variations with respect to $\mathbf{V}_{\text{scale}}$.

**Hyperparameter Sensitivity** Third, in line with Machlanski et al. (2024) and acknowledging that optimal hyperparameters are unknown in real-world applications, we report the average robustness across all hyperparameters in **Table 1** and examine a fine-grained pattern of hyperparameter and data regime influence in **Fig. 3b**. While robustness decreases for all Causal Discovery (CD) methods when averaged across hyperparameters, methods with larger hyperparameter sets (e.g., Dynotears and Nts-Notears) exhibit a notably steeper decline and higher standard deviation. Although these results depend in part on our search space, they underscore the practical challenges of applying these models to real-world data. Additionally, **Fig. 3b** shows that different CD methods can exhibit distinct sensitivity patterns. Relying on $\mathbf{V}_{\text{inno,common}}$ as an example, we observe distinctive behaviors when uncovering $G^{\text{LSG}}$. While some methods (PCMCI+) show a monotonic decrease in performance across all combinations of data regime and hyperparameter configuration, others (Varlingam) begin with a higher performance that eventually diverges. Again, other (Nts-Notears) exhibit high variance (instability) across hyperparameter settings, particularly at low violation levels, even though their performance at extreme violation steps is not poor and relatively consistent. We provide a full list of these patterns for each violation and method in **Apx. D.3**. While we estimate that the ideal sensitivity profile depends on the specific application, we argue that such comparisons that extend beyond simple optimal performance metrics are essential for a rigorous evaluation of CD methods. To conclude, we find several empirical differences across CD methods, raising the question of whether combining multiple CD methods can improve overall robustness. We investigate this question in the next section.

## 4.1 ENSEMBLING CD TO IMPROVE ROBUSTNESS

Given the variability in robustness across different CD methods, we investigate the potential of ensembling techniques to achieve improved general robustness. While ensembling is a cornerstone of modern machine learning Arpit et al. (2022); Mienye & Sun (2022), its potential to enhance the robustness of time series CD methods remains underexplored in the literature. To remedy this, we learn a meta model that predicts the $G^{LWCG}$ or $G^{INST}$ based on the collection of predicted graphs $\{\hat{G}_1, \ldots, \hat{G}_M\}$ from $M$ individual base CD methods. Specifically, we investigate a linear combination (Ensemble$_{\text{Linear}}$), a MLP (Ensemble$_{\text{MLP}}$), and a Transformer Vaswani et al. (2017) (Ensemble$_{\text{Transformer}}$). To train these meta-learners, we generate an additional, independent training dataset containing time series samples for all violations and data regimes. The exact training procedure is contained in **Apx. C.5**. Additionally, we report the performance of a simple unweighted

Table 1: Average robustness under wrongly specified $L$ ($\downarrow L$ denotes too low, $\uparrow L$ denotes too high) for $G^{\text{LWCG}}$ and $G^{\text{LSG}}$. In parentheses, we include the change from a correctly specified $L$. Further, we report the average hyperparameter performance. As CausalPretraining (*) does not require the specification of a max lag, its performance for the $\downarrow L$ regime is superior. As Cross Corr. has no hyperparameters, it has no standard deviation (†). We mark superior performance with **green** ● .

| Method | $G^{\text{LWCG}}$ | | $G^{\text{LSG}}$ | | HP |
|---|---|---|---|---|---|
| | $\downarrow L$ | $\uparrow L$ | $\downarrow L$ | $\uparrow L$ | Avg. |
| CrossCorrelation | .814(-.23) | .622(-.04) | .679(-.23) | .472(-.02) | .582± † |
| CausalPretraining | **.530(-.00)*** | **.530(-.00)*** | **.440(-.00)*** | **.440(-.00)*** | .532±.00 |
| GVAR | **.782(-.36)** | .467(-.04) | **.636(-.31)** | .358(-.03) | .515±.13 |
| Varlingam | .784(-.38) | **.429(-.02)** | .640(-.31) | **.346(-.01)** | **.420±.02** |
| PCMCI | .865(-.26) | .680(-.08) | .710(-.26) | .472(-.02) | .610±.01 |
| PCMCI+ | .836(-.30) | .592(-.05) | .690(-.29) | .420(-.01) | .555±.01 |
| SVAR-RFCI | .862(-.34) | .562(-.04) | .706(-.30) | .425(-.02) | .525±.01 |
| F-PCMCI | .826(-.33) | .528(-.03) | .742(-.31) | .453(-.02) | .504±.01 |
| Dynotears | .789(-.34) | .468(-.02) | .664(-.30) | .378(-.01) | .507±.06 |
| Nts-Notears | .804(-.36) | .483(-.04) | .673(-.31) | .386(-.03) | .634±.18 |

averaging of all $\hat{G}_m$ (Ensemble$_{\text{Avg.}}$) and an oracle strategy that comprises the Pareto front, which we call Ensemble$_{\text{Pareto}}$. For any given assumption violation, Ensemble$_{\text{Pareto}}$ selects the output from the CD method that achieves the highest measured robustness on that specific violation. While not practically attainable, it serves as a baseline, indicating the maximum potential performance gain achievable by perfectly selecting among the outputs of the base methods. All ensembling strategies are evaluated on the datasets used for all other experiments in this paper. We report the average performance of these ensembling approaches in **Fig. 3a** and provide additional analysis of performance gains in **Apx. D.4**. We find that simpler ensembling approaches (Ensemble$_{\text{Linear}}$ and Ensemble$_{\text{MLP}}$) notably improve robustness over any individual method for $G^{LWCG}$ and $G^{LSG}$, while the parameter-intensive approaches show no extrapolation capabilities from the training set (Ensemble$_{\text{MLP}}$ and Ensemble$_{\text{Transformer}}$). Specifically, Ensemble$_{\text{Linear}}$ leads to notable increases and outperforms Ensemble$_{\text{Pareto}}$, suggesting that the learned strategy is truly recombining predictions instead of selecting from them. Surprisingly, while the oracle Ensemble$_{\text{Pareto}}$ achieves a higher performance for $G^{INST}$, we find no superior Ensembles in this case. We present these results as a theoretical proof of concept, suggesting that simple ensembling strategies are promising for enhancing the robustness and reliability of CD methods for complex data. Especially when considering that the here-presented ensembles have no direct access to $\boldsymbol{X}$. As we are aware that applying these approaches to real-world scenarios will require addressing challenges such as domain adaptation and distributional shifts, we provide additional experiments on real-world data in **Apx. D.5**.

## 5 CONCLUSION

This study presents the first extensive empirical investigation into the robustness of Causal Discovery (CD) methods to assumption violations in time series data. We implemented 33 distinct assumption-violation scenarios inspired by real-world data complexities and evaluated 10 CD algorithms. In particular, we first quantify general robustness over all violations, then analyze how model misspecification affects performance, and finally investigate general hyperparameter sensitivities. Motivated by observed differences between CD methods, we investigate ensembles of CD methods and conclude that they can improve general robustness. Our study is supported by TCD-Arena, an empirical framework and testing kit for time series CD that we developed to conduct all our experiments. Given the vast landscape of potential data-generating processes, we are releasing TCD-Arena[1] as an open-source, modular package to facilitate future extensions and foster long-term comparability. We also provide a reproducibility statement in **Apx. C.6**. With this, we aim to foster a deeper understanding of causal discovery methods, including their strengths and weaknesses, across diverse synthetic and semi-synthetic conditions, thereby paving the way for more robust real-world applications.

---

[1]https://github.com/TCD-Arena

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

# Contents

# A APPENDIX — HIGH LEVEL OVERVIEW

| Violation | Short Description |
|---|---|
| $V_{obs,add}$ | Additive measurement noise. |
| $V_{obs,mul}$ | Signal dependent measurement noise. |
| $V_{obs,time}$ | Time-varying measurement noise. |
| $V_{obs,auto}$ | Autoregressive measurement noise. |
| $V_{obs,com}$ | Common source measurement noise. |
| $V_{obs,shock}$ | Spike measurement noise. |
| $V_{obs,real}$ | Measurement noise extracted from real world TS. |
| $V_{conf,inst}$ | Unseen internal common cause. |
| $V_{conf,lag}$ | Unseen external common causes. |
| $V_{faith,inst}$ | Instantaneous effects cancel out. |
| $V_{faith,lag}$ | Lagged effects cancel out. |
| $V_{faith,zero}$ | Effects become extremely small. |
| $V_{nl,mono}$ | Monotonic functions. |
| $V_{nl,trend}$ | B-spline functions with a linear trend. |
| $V_{nl,rbf}$ | GP-RBF functions. |
| $V_{nl,comp}$ | Composite functions. |
| $V_{inno,mul}$ | Signal dependent innovation noise. |
| $V_{inno,time}$ | Time-dependent innovation noise. |
| $V_{inno,auto}$ | Autoregressive innovation noise. |
| $V_{inno,com}$ | Common source innovation noise. |
| $V_{inno,shock}$ | Spike innovation noise. |
| $V_{inno,real}$ | Using a real-world ts as innovation noise. |
| $V_{inno,uni}$ | Uniform additive innovation noise. |
| $V_{inno,weib}$ | Weibull additive innovation noise. |
| $V_{inno,var}$ | Unequal variances in innovation noise. |
| $V_{coef}$ | Causal link strengths change over time. |
| $V_{stat}$ | Causal relationships are completely resampled over time. |
| $V_{length}$ | Reduced time series length. |
| $V_{mcar}$ | TS with values missing data completely at random |
| $V_{mar}$ | TS with values missing data at random (depending on a real-world TS) |
| $V_{mnat}$ | TS with values missing not data at random (depending on TS itself) |
| $V_{empty}$ | Temporary complete loss of causal signal. |
| $V_{scale}$ | Data standardization. |

Table 2: List of all 33 violations contained in TCD-Arena with corresponding short descriptions.

## A.1 VIOLATION LIST

Table 2 contains a brief overview of all 33 violations contained in TCD-Arena and investigated empirically in this work. Further details on the implementation and evaluated severity levels can be found in the following chapters.

## A.2 VIOLATION STEPS

| Violation | Increasing Intensity via | Parameter Values |
|---|---|---|
| $V_{obs,add}$ | Reducing the SNR | $\{1.1, 0.8375, 0.575, 0.3125, 0.05\}$ |
| $V_{obs,mul}$ | Reducing the SNR | $\{1.1, 0.8375, 0.575, 0.3125, 0.05\}$ |
| $V_{obs,time}$ | Reducing the SNR | $\{1.1, 0.8375, 0.575, 0.3125, 0.05\}$ |
| $V_{obs,auto}$ | Reducing the SNR | $\{1.1, 0.8375, 0.575, 0.3125, 0.05\}$ |
| $V_{obs,com}$ | Reducing the SNR | $\{1.1, 0.8375, 0.575, 0.3125, 0.05\}$ |
| $V_{obs,shock}$ | Reducing the SNR | $\{1.1, 0.8375, 0.575, 0.3125, 0.05\}$ |
| $V_{obs,real}$ | Reducing the SNR | $\{1.1, 0.8375, 0.575, 0.3125, 0.05\}$ |
| $V_{conf,inst}$ | Increased link probability from hidden confounders $\mathcal{Z}$. | $\{0.135, 0.27625, 0.4175, 0.55875, 0.7\}$ |
| $V_{conf,lag}$ | Increased link probability to/from hidden confounder $X_c$. | $\{0.135, 0.27625, 0.4175, 0.55875, 0.7\}$ |
| $V_{faith,lag}$ | Lagged effects increasingly cancel out. | $\{0.2, 0.15, 0.1, 0.05, 0.0\}$ |
| $V_{faith,inst}$ | Instantaneous effects increasingly cancel out. | $\{0.2, 0.15, 0.1, 0.05, 0.0\}$ |
| $V_{faith,zero}$ | Decrease the range from which elements in $A$ are drawn. | $Max \in \{0.24, , 0.1925, 0.145, 0.0975, 0.05\}$ |
| $V_{nl,mono}$ | Increasing the nonlinearity of sampled monotonic functions. | See Eq. (22) in Apx. B.4 |
| $V_{nl,trend}$ | Reducing number of interpolation points for B-spline functions. | $\{12, 10, 8, 6, 4\}$ |
| $V_{nl,rbf}$ | Higher probability of nonlinear links in the SCM (GP-RBF functions). | $\{0.425, 0.56875, 0.7125, 0.85625, 1\}$ |
| $V_{nl,comp}$ | Higher probability of nonlinear links in the SCM (composite functions). | $\{0.05, 0.2875, 0.525, 0.7625, 1\}$ |
| $V_{inno,mul}$ | Signal dependent innovation noise portion. | $\{0.91, 0.93, 0.95, 0.97, 0.99\}$ |
| $V_{inno,time}$ | Time-dependent innovation noise portion. | $\{0.8, 0.85, 0.9, 0.95, 1\}$ |
| $V_{inno,auto}$ | Autoregressive innovation noise portion. | $\{0.35, 0.5, 0.65, 0.8, 0.95\}$ |
| $V_{inno,com}$ | Common source innovation noise portion. | $\{0.475, 0.60625, 0.7375, 0.86875, 1\}$ |
| $V_{inno,shock}$ | Spike innovation noise portion. | $\{0.8, 0.85, 0.9, 0.95, 1\}$ |
| $V_{inno,real}$ | Noise consisting of a Real-world time series . | $\{0.8, 0.85, 0.9, 0.95, 1\}$ |
| $V_{inno,uni}$ | Uniform additive innovation noise scale. | $\{0.05, 0.25, 0.5, 0.75, 1.0\}$ |
| $V_{inno,weib}$ | Weibull additive innovation noise scale. | $\{0.2, 0.4, 0.6, 0.8, 1.0\}$ |
| $V_{inno,var}$ | Outer boundaries of the interval from which $\sigma_i^2$ are sampled. | $\{[0.551.45], [0.4125, 1.5875], [0.275, 1.725], [0.1375, 1.8625], [0., 2]\}$ |
| $V_{coef}$ | Increase the maximum allowed distances for changes in $A$. | $\{0.275, 0.33125, 0.3875, 0.44375, 0.5\}$ |
| $V_{stat}$ | Increase the number of resamples of $A$. | $\{1, 2, 3, 4, 5\}$ |
| $V_{length}$ | Reducing number of observed steps $T$ | $\{76, 58, 41, 23, 6\}$ |
| $V_{mcar}$ | Increasing probability of missing data points. | $\{0.2, 0.3375, 0.475, 0.6125, 0.75\}$ |
| $V_{mar}$ | Increasing probability of missing data points. | $\{0.2, 0.3375, 0.475, 0.6125, 0.75\}$ |
| $V_{mnar}$ | Increasing probability of missing data points. | $\{0.2, 0.3375, 0.475, 0.6125, 0.75\}$ |
| $V_{empty}$ | Lengthening % of timesteps with temporary loss of causal signal. | $\{0.52, 0.6, 0.7005, 0.824, 0.928\}$ |
| $V_{scale}$ | Mixing factor of the standardization. | $\{0.2, 0.4, 0.6, 0.8, 1\}$ |

Table 3: A short description of how we scale all 33 violations contained in TCD-Arena. We also include a list of the specific parameter values to reproduce our empirical study.

Table 3 contains a list of parameter values used to intensify all 33 violations contained in TCD-Arena and included in our study. Additionally, we note a short description of how each violation is scaled. We refer to Apx. B for more in-depth descriptions. The experimental violations were individually configured to establish a range that challenges the performance of Causal Discovery (CD) methods. Recognizing that the disruptive impact of each violation type varies considerably, a standardized approach was not employed. Instead, for each violation, the parameters were calibrated according to a three-step procedure. First, a parameter configuration was identified that induced a minor performance degradation for the baseline Cross-Correlation (CC) method. Second, where the violation type allowed, the maximum intensity was set to reduce the CC's performance to an Area Under the Receiver Operating Characteristic (AUROC) of approximately 0.5 (random performance). Third, discrete levels of the violation were established by equally spacing them between a minimal-effect level and the determined maximum. This methodology ensures that various CD methods can be effectively compared within a challenging, relevant operational range for each specific violation. Notably, though, it precludes direct performance comparisons across different violation types. We fully document this process in **TCD-Arena** for maximum clarity. Finally, as we only evaluate five

violation levels in this study, it is worth discussing when our robustness metric might fail. We discuss this in Apx. A.5.

## A.3 EXPERIMENTAL PROTOCOL DEPICTION

To clarify the protocol we used to assess the robustness of various CD methods against assumption violations, we depict the process in Fig. 4. The process can be divided into three aspects. Data generation, CD method evaluation, and the extraction of robustness profiles.

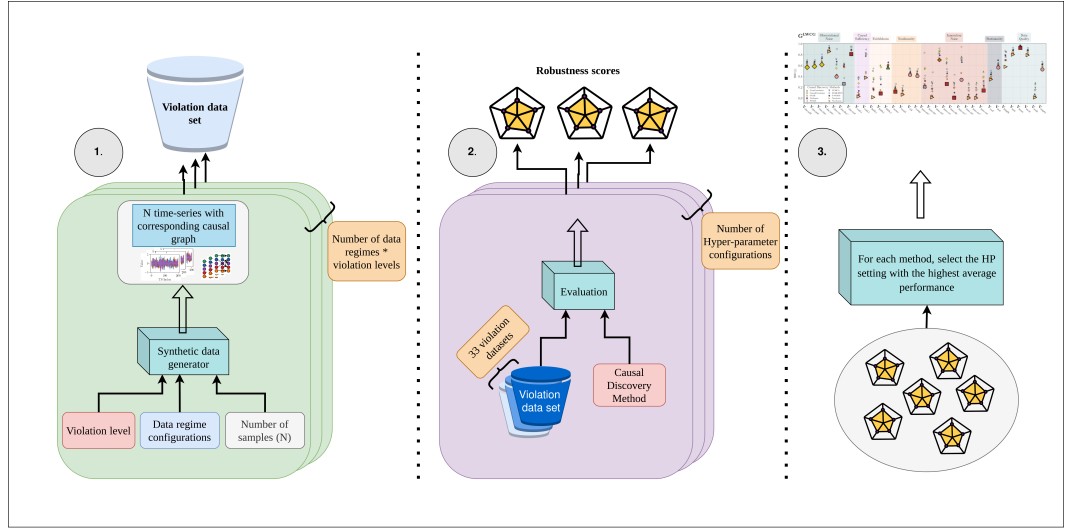

Figure 4: The experimental protocol was used to create robustness profiles for various Causal Discovery methods. The process can be divided into three steps: **1.** Data generation, **2.** CD method evaluation and **3.** Extraction of results.

## A.4 VIOLATION DEPICTIONS

We provide a graphical illustration for each violation type. Although the visualization methods vary, each figure isolates a specific data property that evolves with the severity of the violation. This is intended to enhance the intuitive understanding of the specific mechanism (Fig. 5 – Fig. 10).

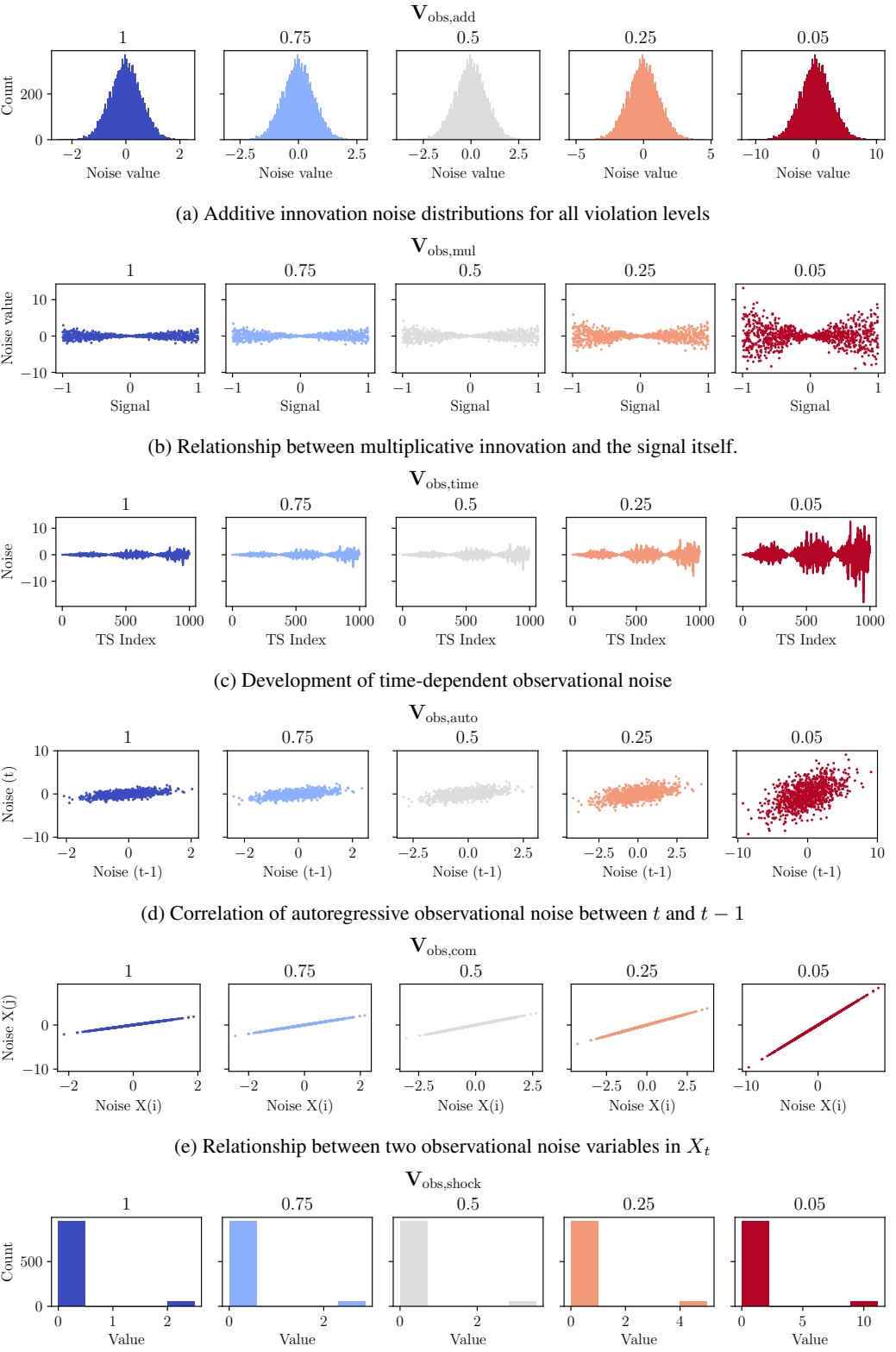

(a) Additive innovation noise distributions for all violation levels

(b) Relationship between multiplicative innovation and the signal itself.

(c) Development of time-dependent observational noise

(d) Correlation of autoregressive observational noise between $t$ and $t-1$

(e) Relationship between two observational noise variables in $X_t$

(f) Distributions over noise values when sampled from a shock distribution.

Figure 5: Various depictions of different violations of observational noise. We depict the severity of the violation from left to right and denote the SNR above the figure.

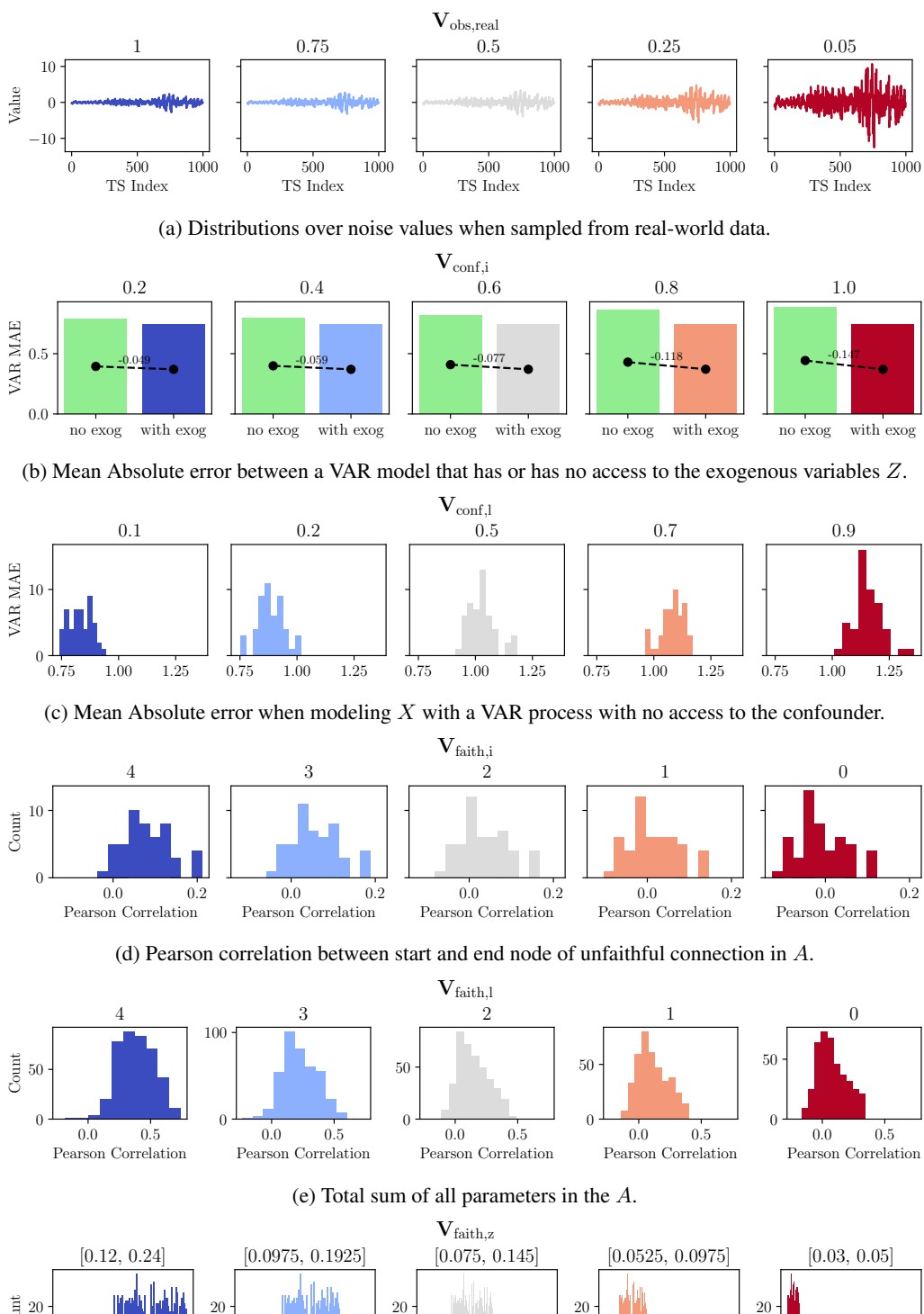

(a) Distributions over noise values when sampled from real-world data.

(b) Mean Absolute error between a VAR model that has or has no access to the exogenous variables $Z$.

(c) Mean Absolute error when modeling $X$ with a VAR process with no access to the confounder.

(d) Pearson correlation between start and end node of unfaithful connection in $A$.

(e) Total sum of all parameters in the $A$.

(f) Total sum of all parameters in the $A$.

Figure 6: Graphical depictions of different violations and their intensities.

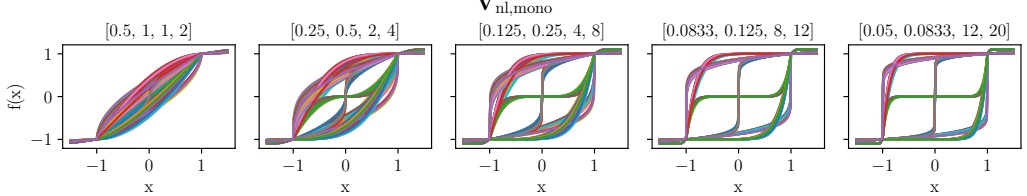

(a) Distributions from which we draw the nonlinear functions of the monotonic family.

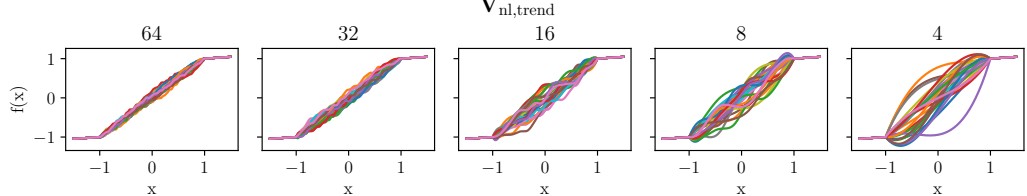

(b) Distributions from which we draw the nonlinear functions of the trend family.

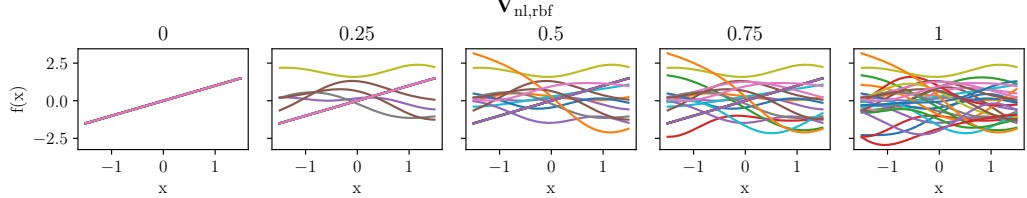

(c) Distributions from which we draw the nonlinear functions of the RBF family.

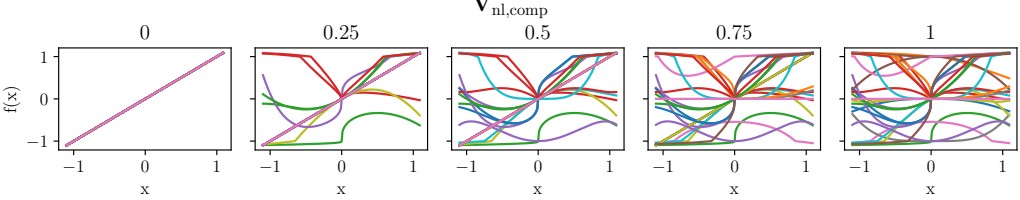

(d) Distributions from which we draw the nonlinear functions of the composite family.

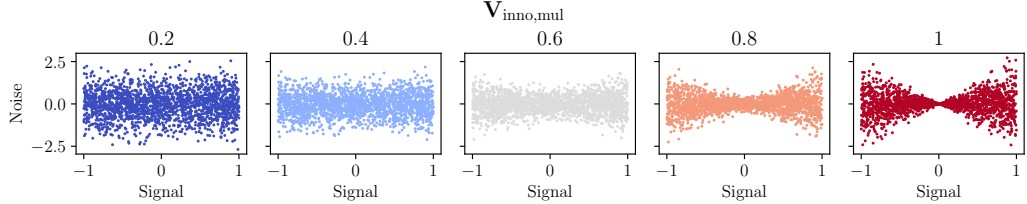

(e) Relationship between signal strength and innovation noise.

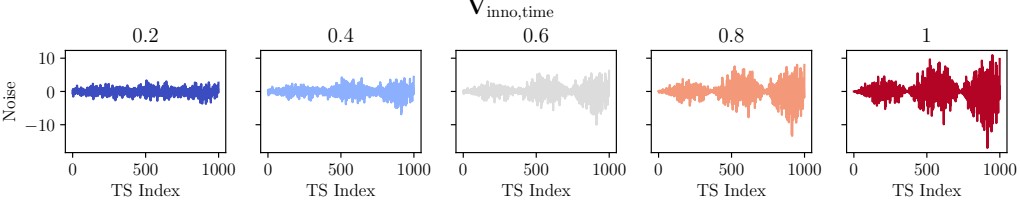

(f) Innovation noise over time.

Figure 7: Graphical depictions of different violations and their intensities.

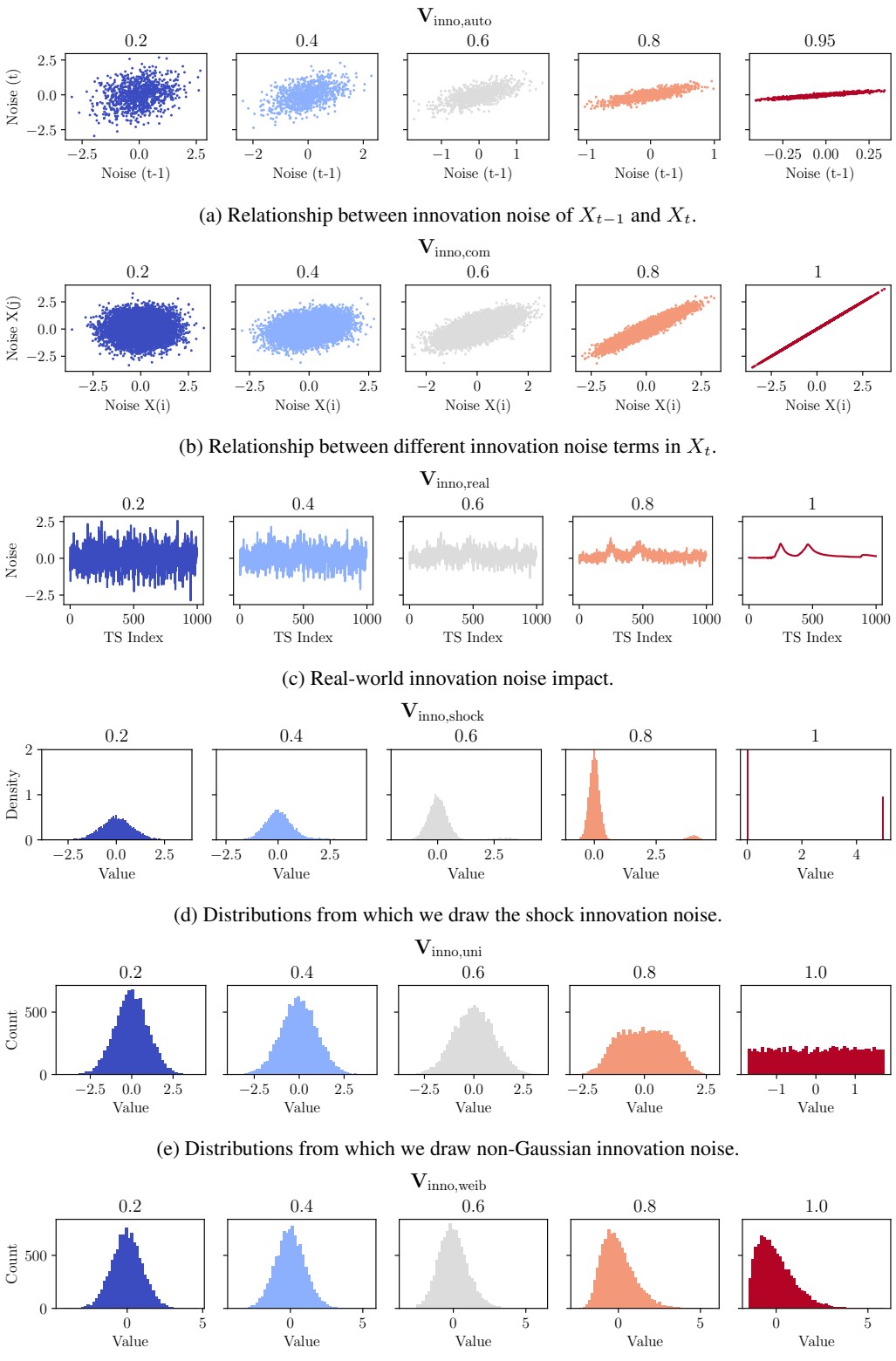

(a) Relationship between innovation noise of $X_{t-1}$ and $X_t$.

(b) Relationship between different innovation noise terms in $X_t$.

(c) Real-world innovation noise impact.

(d) Distributions from which we draw the shock innovation noise.

(e) Distributions from which we draw non-Gaussian innovation noise.

(f) Distributions from which we draw non-Gaussian innovation noise.

Figure 8: Graphical depictions of different violations and their intensities.

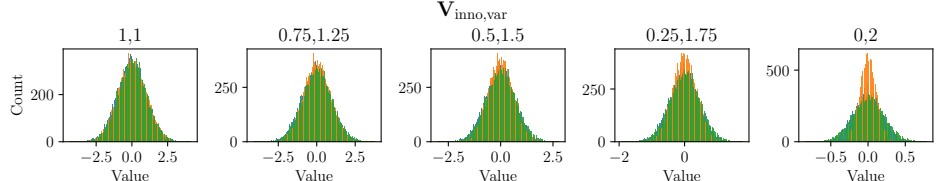

(a) Distributions from which we draw innovation noise with unequal variance for each variable in $X$.

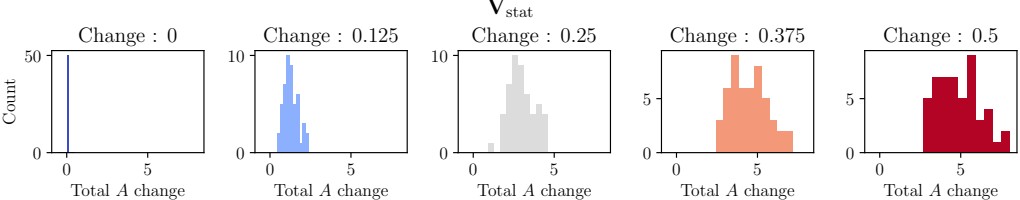

(b) Causal Coefficient absolute changes (elements in $A$) between first and last section of $X$.

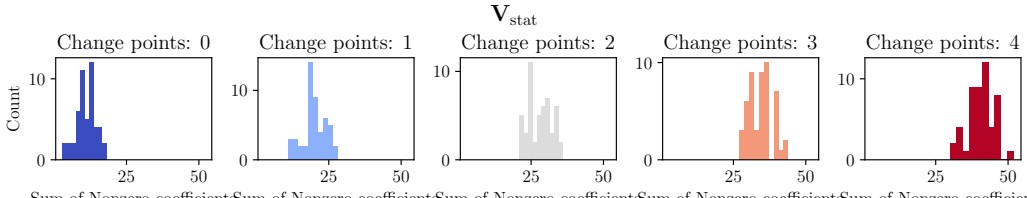

(c) Number of temporarily non-zero elements in $A$.

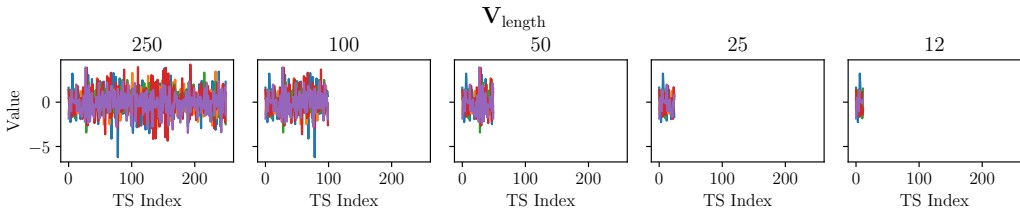

(d) Length of the time generated time series.

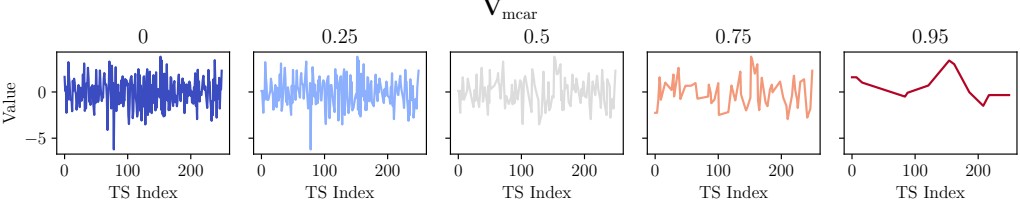

(e) Time series with increasingly missing (mcar) and afterwards interpolated values.

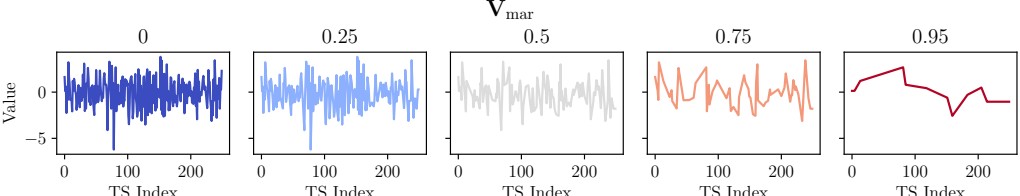

(f) Time series with increasingly missing (mar) and afterwards interpolated values.

Figure 9: Graphical depictions of different violations and their intensities.

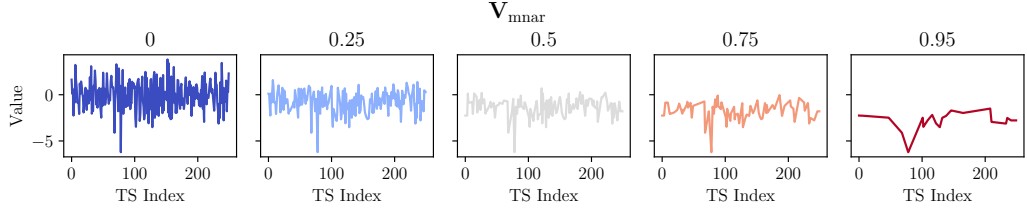

(a) Time series with increasingly missing (mnar) and afterwards interpolated values.

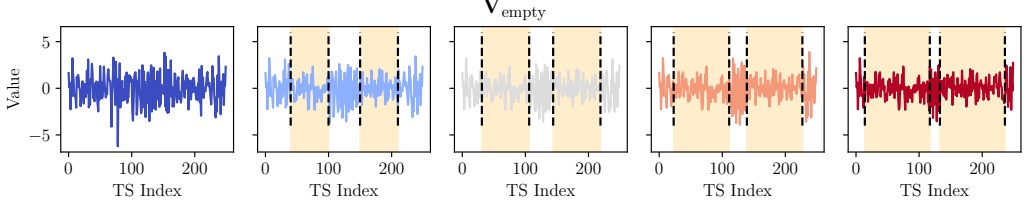

(b) Timeseries with partially empty $A$. Sections where $A$ is empty are marked as yellow areas.

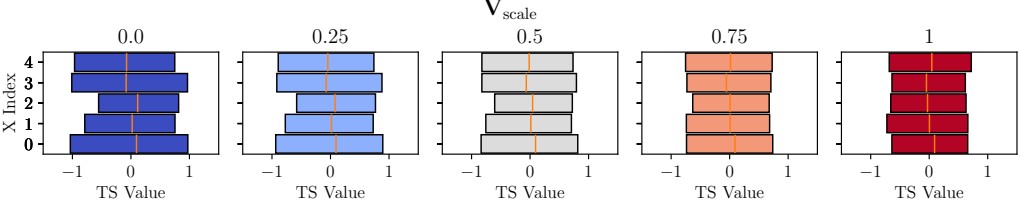

(c) Mean and standard deviations of individual variables in $X$.

Figure 10: Graphical depictions of different violations and their intensities.

## A.5 DISCUSSION ON METRIC FAILURE CASES

In this study, we fundamentally quantify the robustness of a method to an assumption violation using a limited number of samples (five violation intensities). As the underlying robustness is often continuous, we deem it reasonable to discuss under what conditions our methodological approach may yield potentially misleading results. To this end, we depict three simplified scenarios in Fig. 11.

In the first scenario (green box), we observe a consistent separation in robustness across methods. In this ideal scenario, the green curve consistently outperforms the blue curve across the entire range of violations. The discrete measurements (marked by stars) accurately capture this relationship, yielding a robust, faithful score.

In the second scenario (blue box), the curves cross. The discrete sampling points suggest that the performance of both curves is roughly equal across the measured violation levels, resulting in a very similar robustness score. However, this discrete evaluation fails to account for the precise dynamics of the robustness curves. Notably, depending on the application, either the blue or the green curve could be preferable.

In the third scenario, a non-monotonic and highly volatile curve highlights the most critical risk. At the discrete evaluation points, the blue curve is preferred even though its robustness is disputable.

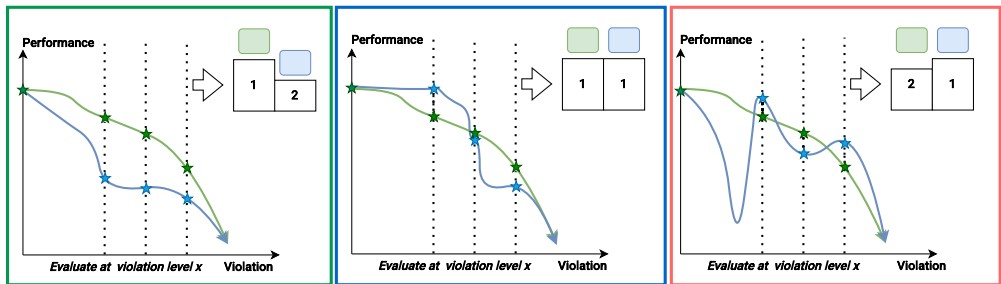

Figure 11: Depiction of problematic relationships between violation property and robustness measured as any performance score for a single data regime and a single method configuration. Left: Optimal case in which all performance curves are monotonically decreasing, and one curve is Pareto superior. Middle: While both curves are monotonically decreasing, our metric does not directly distinguish between them. Right: If any performance curve is highly non-monotonic, the comparison can be misleading.

# B    APPENDIX — VIOLATION DETAILS

Fundamentally, the base SCM that we use for each violation is a linear causal process with additive Gaussian noise:

$$X_{i,t} = \sum_{d=1}^{D} \sum_{l=0}^{L} A_{i,d,l} \cdot f_{i,d,l}(X_{d,t-l}) + \epsilon_{t,i}, \tag{4}$$

where $f$ is the identify function, $A$ a coefficient tensor, $X$ the time series and $\epsilon$ additive gaussian noise. To bring this into a more compact form, omitting $f$:

$$X_{i,t} = \sum_{d=1}^{D} \sum_{l=0}^{L} A_{i,d,l} \cdot X_{d,t-l} + \epsilon_{t,i}, \tag{5}$$

Further, we can bring this into matrix notation:

$$X_t = AX_{t...t-L} + \mathcal{E} \tag{6}$$

where $\mathcal{E}$ is a vector of independent innovation noise variables. Crucially, the non-zero entries in $A$ correspond to links in the causal graph $G$. While sampling $A$ in the base linear process, we control the density of links using a corresponding probability that determines whether entries $A_{i,d,l}$ are equal to zero.

Finally, we can separate instantaneous effects as they are not always present and are implemented in a different manner:

$$X_t = BX_t + AX_{t-1...t-L} + \mathcal{E} \tag{7}$$

where $A$ is a coefficient matrix and $B$ is a coefficient vector.

To implement all our violations, we alter this basic linear additive process.

## B.1    ADDITIONAL DETAILS $\mathbf{V}_{\text{OBS}}$

To briefly recap, when we violate the no observational noise assumption, we do not directly observe the measurements $X_{i,t}$. Instead, we measure noisy versions $\hat{X}_{i,t} = X_{i,t} + \zeta_{i,t}$. We define a concrete list of observational noise variables and structures in Fig. 2a. In this section, we provide additional details for the specific design choices of the various implemented $\zeta_{i,t}$. Afterward, we present the concrete formula we use to control the signal-to-noise ratio as we increase the respective observational noise violations.

First, we consider independent additive noise ($\mathbf{V}_{\text{obs,add}}$), where we model the noise as standard normal $\zeta_{i,t} \sim \mathcal{N}(0,1)$.

Second, we consider multiplicative noise ($\mathbf{V}_{\text{obs,mul}}$), which in the signal-dependent noise model is an additive noise scaled by a function of the signal strength Torricelli et al. (2002); Liu et al. (2014). Here, we use $\zeta_{i,t} \sim \mathcal{N}(0, (X_{i,t})^2)$, i.e., a multiplication of standard normal noise with the signal $X_{i,t}$ (see Fig. 2a). Real-world examples include temperature sensors whose precision degrades at high signal values Bentley (1984) and speckle noise in image processing Liu et al. (2014).

Third, $\mathbf{V}_{\text{obs,time}}$, specifies noise with distribution characteristics changing over time. Real-world examples of such noise sources include sensor drift and interference from periodic environmental factors. We model this by scaling the variance by a periodic signal, i.e., $\zeta_{i,t} \sim \mathcal{N}(0, ((1 + \alpha t) \cdot \sin(2\pi t/\beta))^2)$, where $\alpha$ and $\beta$ are hyperparameters. In our experiments, we fix them to simulate an annual cycle and a small linear trend to simulate sensor degradation. Specifically, we use $\alpha = 0.01$ and $\beta = 2 \cdot 365 = 730$.

Fourth, $\mathbf{V}_{\text{obs,auto}}$ indicates an autoregressive noise structure ($\zeta_{i,t} \not\perp\!\!\!\perp \zeta_{i,t-1}$) that can be found when disturbances of the measurement process persist over multiple timesteps. Here, one could imagine a sensor that is overshadowed by a cloud for multiple consecutive time steps. We model this term as $\zeta_{i,t} \sim \mathcal{N}(\alpha\zeta_{i,t-1} + (1-\alpha)\mathcal{N}(0,1))$ where $\alpha$ is the weighting coefficient which is a

hyperparameter. Intuitively, the mean of the distribution depends on the last sampled noise, similarly to a random walk. In our implementation, we equally mix the previous step with the random source, i.e., $\alpha = 1/2$. Notably, keeping $\alpha$ below 1 ensures that the overall process, while dependent on prior noise, remains nondeterministic.

Fifth, noise sources across different variables can be dependent ($\mathbf{V}_{\text{obs,com}}$), i.e., $\zeta_{i,t} \not\perp\!\!\!\perp \zeta_{j,t}$ for $i \neq j$. Such a scenario can occur when multiple sensors are affected by a shared, unmeasured environmental factor (e.g., temperature or power fluctuations). We model this by sampling from a single noise source $\zeta_t \sim \mathcal{N}(0,1)$ that is shared for all variables in $\boldsymbol{X}$, i.e., $\forall i \in \{1, \ldots, D\} : \zeta_{i,t} = \zeta_t$ for a timestep $t$.

Sixth, observed data might be subject to infrequent, large disturbances or measurement failures ($\mathbf{V}_{\text{obs,shock}}$). Using a shock probability $p_{\text{shock}}$, we model $\zeta_{i,t} = \begin{cases} S & \text{with probability } p_{\text{shock}} \\ 0 & \text{else} \end{cases}$, where $S$ is a fixed scalar. In our experiments, we set $= 5$ and $p_{\text{shock}} = 0.05$.

Seventh, we directly extract noise from real-world time series to generate semi-synthetic measurement noise ($\mathbf{V}_{\text{obs,real}}$). We deploy a High-Pass Butterworth filter Butterworth (1930) to extract high-frequency components from real-world stock price time series. We then randomly sample appropriate-length sequences as observational noise components.

As specified in our main paper, we isolate the influence of the noise structure by using five discrete, decreasing SNR levels. Inparticular, we are using $\{10, 5, 1, 1/2, 1/10\}$ for all observational noise violations in Fig. 2a.

To rescale the noise vector, to achieve a desired Signal-to-Noise Ratio ($\text{SNR}_{\text{target}}$) with respect to $X$, we first compute the average power of the signal and the unscaled base noise $\zeta_{base}$.

The signal power, $P_X$, is defined as:

$$P_X = \frac{1}{T * D} \sum_{d=1}^{D} \sum_{t=1}^{T} X_{d,t}^2$$

The power of $\zeta_{base}$, is:

$$P_{\zeta,base} = \frac{1}{T * D} \sum_{d=1}^{D} \sum_{t=1}^{T} \zeta_{\text{base,d},t}^2$$

Given a target $\text{SNR}_{\text{target}}$, the desired power for the final noise, $P_{N,\text{target}}$, is calculated as:

$$P_{N,\text{target}} = \frac{P_X}{\text{SNR}_{\text{target}}}$$

We find a scaling factor, $\alpha$, that transforms the base noise power to the target noise power.

$$\alpha = \sqrt{\frac{P_X / \text{SNR}_{\text{target}}}{P_{\zeta,base}}}$$

The final noise vector $\zeta$ is then obtained by scaling the base noise:

$$\zeta = \alpha \cdot \zeta_{base}.$$

## B.2 ADDITIONAL DETAILS $\mathbf{V}_{\text{CONF}}$

In our main paper, we specify two possible methods for introducing confounding into a sampled time series. Specifically, we separate instantaneous effects ($\mathbf{V}_{\text{conf,inst}}$) and and lagged confounding ($\mathbf{V}_{\text{conf,lag}}$). We model the former by generating a set of $N$ independent potential parent variables $Z_1, ..., Z_N$. In all time steps $t$, the $Z_n$ are standard normally distributed and can act as common causes for any variable in $\boldsymbol{X}$. Hence, the causal assignment (Eq. (1)) for an observed variable $X_{i,t}$ becomes: $X_{i,t} = f_i\left(\text{Pa}_{\boldsymbol{X}}(X_{i,t}) \cup \text{Pa}_{\mathcal{Z}}(X_{i,t}), \epsilon_{i,t}\right)$. We scale $\mathbf{V}_{\text{conf,inst}}$ by increasing the probability of links from $\mathcal{Z}$ to variables in $\boldsymbol{X}$. In particular, we use the probabilities $\{0.2, 0.4, 0.6, 0.8, 1.0\}$. Finally, to make the more pronounced, links from $Z$ to $X$ are parameterized slightly higher ($\{0.8, 0.9\}$ than elements in $A$ ($\{0.3, 0.5\}$).

To model $\mathbf{V}_{\text{conf,lag}}$, generate the time series with an additional confounding variable $X_C$. This variable is part of the normal SCM and can influence any variable $X_i$. Further, it can also be influenced by all other variables. After generating the complete time series, we create the observed data $\boldsymbol{X}$ by removing $X_C$, rendering it a hidden confounder. To scale $\mathbf{V}_{\text{conf,lag}}$, we again increase the probability of links from and to $X_C$. Specifically, we use the probabilities $\{0.1, 0.2, 0.5, 0.7, 0.9\}$. As before, links from $Z$ to $X$ are parameterized slightly higher ($\{0.8, 0.9\}$ than elements in $A$ ($\{0.3, 0.5\}$)) to render the effect more pronounced.

## B.3 ADDITIONAL DETAILS $\mathbf{V}_{\text{FAITH}}$

To violate faithfulness via path cancellation ($\mathbf{V}_{\text{faith,i}}$, $\mathbf{V}_{\text{faith,l}}$), we have to ensure that there are variables with a connection in the causal graph $G$, which have no measurable dependency, i.e., cancel each other out. We separate instantaneous effects ($\mathbf{X}_{\text{faith,inst}}$) and lagged effects ($\mathbf{X}_{\text{faith,lag}}$) and visualize the structures we implement in Fig. 12.

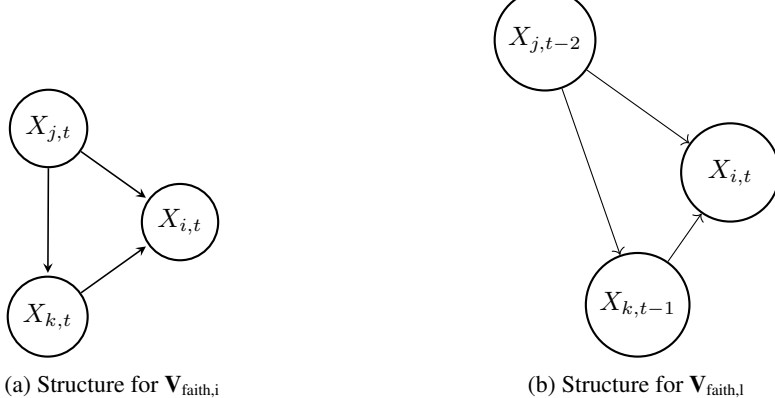

(a) Structure for $\mathbf{V}_{\text{faith,i}}$        (b) Structure for $\mathbf{V}_{\text{faith,l}}$

Figure 12: The two structures, we enforce to violate faithfulness. Note that in Fig. 12b, some of the variables are lagged. In both cases, the connection from $X_j$ to $X_i$ is partially canceled by the connection over $X_k$, so $X_i$ and $X_j$ become closer to independent during the violations.

Now, to scale the violations of the faithfulness assumption, we adapt the parameters of the causal graph to cancel out information of $X_j$ from $X_i$, using the path over $X_k$ (see Fig. 12). Specifically, we implement the following assignments for $\mathbf{V}_{\text{faith,inst}}$ and $\mathbf{V}_{\text{faith,lag}}$ respectively:

$$v \sim \text{Uniform}(0.3, 0.5)$$
$$X_{k,t} := 2vX_{j,t} + \epsilon_{i,t}$$
$$X_{i,t} := (-v + d)X_{j,t} + 0.5X_{k,t} + \epsilon_{i,t}$$

$$v \sim \text{Uniform}(0.3, 0.5)$$
$$X_{k,t-1} := 2vX_{j,t-2} + \epsilon_{i,t}$$
$$X_{i,t} := (-v + d)X_{j,t-2} + 0.5X_{k,t-1} + \epsilon_{i,t}$$

Here, $d$ denotes the distortion parameter, which we decrease as the violation severity increases. We choose the levels $\{0.2, 0.15, 0.1, 0.05, 0.0\}$ for the violation severity in both cases. Further, we note that this implementation is only one way, arguably the most straightforward, to generate Unfaithfulness. As an alternative, we introduce $\mathbf{V}_{\text{faith,zero}}$ where we stepwise reduce the range of elements in $A$ to be closer to $0$, making it harder to detect statistical dependencies. During our experiments, the elements of $A$ are sampled uniformly from the intervals $[0.12, 0.24], [0.0975, 0.1925], [0.075, 0.145],$

$[0.0525, 0.0975]$, and $[0.03, 0.05]$, respectively. Finally, as other ways of generating Unfaithfulness, such as deterministic relationships, conditional links (XOR), or specific mixtures of causal models, are possible, we plan on extending TCD-Arena in these directions in the future to gain additional insights.

## B.4  ADDITIONAL DETAILS $\mathbf{V}_{\text{NL}}$

To introduce nonlinearities, studies on CD robustness sample functions from Gaussian Processes (GPs) with Radial Basis Function (RBF) kernels Yi et al. (2025); Montagna et al. (2023a). The smooth and oscillatory nature of these functions provides a difficult test case for algorithms that assume linear relationships, motivating the development of more robust nonlinear methods.

However, they may not fully represent the diverse spectrum of nonlinearities encountered in real-world applications. In many domains, such as those governed by physical constraints, nonlinearities often adhere to specific characteristics like monotonicity or saturation, rather than arbitrary nonlinearity (e.g., SIR-models Kermack et al. (1997) or the Michaelis-Menten Kinetics Ainsworth (1977)).

In this section, we detail our four specific choices in nonlinear function distributions and the respective design paradigms. Further, a critical consideration in generating synthetic time series from such structural equations is ensuring the stability of the process (i.e., preventing divergence) because the functions are applied iteratively. This often requires constraining the output range or characteristics of the sampled functions $f_{i,d,l}$. As the specific constraints depend on the functional class, we first detail the normalization or bounding procedures possible during the generation process. Then we note the respective choices for the four families of functions we investigate. We also formalize what we mean by increasing the nonlinearity of the structural causal model (Eq. (2)).

### B.4.1  DETAILS ON BOUNDARY CONDITIONS

For sampling nonlinear functions used to iteratively generate observations $\boldsymbol{X}$ using Eq. (7), it is important to consider the boundary behavior when inputs are either very large $x \gg 0$ or very small $x \ll 0$, as it is possible for any of the $D$ time series to diverge towards positive or negative infinity. This behavior could lead to numerically unstable values even in our finite simulated time steps $T$. While we will discuss specific checks to test for such conditions in Apx. C.1, we consider it here explicitly for the set of violations $\mathbf{V}_{\text{nl}}$ concerning the functional relationships $f_{i,d,l}$.

In our implementation, we focus on the input interval $x \in [-1, 1]$. Again, this specific setup is motivated by the time series sampling process, in which we apply these functions iteratively as described in our main paper. To enforce saturation for $x \to \pm\infty$, we wrap sampled univariate functions $f$ using hyperbolic tangents to roughly enforce value ranges of $[-1, 1]$. Consequently, values contained in the generated time series $\boldsymbol{X}$ stay close to $[-1, 1]$.

Specifically, we either wrap a function $f$ with respect to the input $x$ or the output $f(x)$ to ensure saturation. In particular, we employ

$$s_x(f(x)) = \begin{cases} f(x) & \text{if } x \in [-\alpha, \alpha] \\ \tanh(x) & \text{else} \end{cases}, \tag{8}$$

and

$$s_y(f(x)) = \begin{cases} f(x) & \text{if } |f(x)| <= \alpha \\ \tanh(f(x)) & \text{else} \end{cases}, \tag{9}$$

respectively. In Eq. (8) and Eq. (9), $\alpha$ defines the symmetric intervals around zero, which in our experiments is set to one. When detailing the specific functional families used for the violations $\mathbf{V}_{\text{nl,mono}}$, $\mathbf{V}_{\text{nl,trend}}$, $\mathbf{V}_{\text{nl,rbf}}$, and $\mathbf{V}_{\text{nl,comp}}$, we will specify which concrete wrapper $s_x$ or $s_y$, we use to ensure saturation of the sampled $f_{i,d,l}$.

### B.4.2  QUANTIFYING AND INCREASING NONLINEARITY

To formalize the concept of nonlinearity of a univariate function $f$, multiple scores were proposed in the literature. For instance, roughness penalties for spline smoothing, e.g., (Ramsay & Silverman, 2005, Sec. 5.2.2), quantify it using the squared curvature of a function over a specified interval. In

particular, they calculate the deviation from a linear function as

$$\mathscr{D}_{\text{curv}}(f) = \int_{-\alpha}^{\alpha} (f''(x))^2 dx, \tag{10}$$

where $f''$ denotes the second derivative. If and only if the second derivative is zero over the complete interval $[-\alpha, \alpha]$, then $f$ is linear in said interval. Further, given the squared integrand, $\mathscr{D}_{\text{curv}}$ is strictly nonnegative with minima exactly when $f$ is a linear function. Intuitively, this score quantifies changes in the derivative of $f$, which is constant only for linear functions.

In contrast, in Emancipator & Kroll (1993), the authors measure the minimum possible mean squared error of $f$ to any linear function in the interval of interest. Specifically, for a given $f$, we follow their approach and define the nonlinearity $\mathscr{D}_{\text{MSE}}$ in an interval $[-\alpha, \alpha]$ as

$$\mathscr{D}_{\text{MSE}}(f) = \min_{a,b \in \mathbb{R}} \left( \frac{1}{2\alpha} \int_{-\alpha}^{\alpha} \big(f(x) - (ax+b)\big)^2 dx \right). \tag{11}$$

Intuitively, $\mathscr{D}_{\text{MSE}}$ measures the minimum possible mean squared error to any linear function in $[-\alpha, \alpha]$ and is greater than or equal to zero for any $f$. Further, if $f$ can be expressed as a linear function, then $\mathscr{D}_{\text{MSE}}$ is exactly zero. To compute $\mathscr{D}_{\text{MSE}}$, we have to consider the optimal $a^*, b^* \in \mathbb{R}$, which are necessary to minimize the mean squared error. In Emancipator & Kroll (1993), the authors give general solutions for arbitrary interval boundaries. In our case of a boundary symmetric around $x = 0$ of $[-\alpha, \alpha]$, the optimal solutions that minimize the error for a function $f$ are given by

$$a^* = \frac{3}{2\alpha^3} \int_{-\alpha}^{\alpha} x f(x) dx, \quad \text{and}$$
$$b^* = \frac{1}{2\alpha} \int_{-\alpha}^{\alpha} f(x) dx. \tag{12}$$

Hence, the measure for nonlinearity becomes

$$\mathscr{D}_{\text{MSE}}(f) = \frac{1}{2} \int_{-1}^{1} \big(f(x) - (a^*x + b^*)\big)^2 dx. \tag{13}$$

Both $\mathscr{D}_{\text{curv}}$ and $\mathscr{D}_{\text{MSE}}$ behave differently and are not always aligned. Specifically, $\mathscr{D}_{\text{MSE}}$ considers the absolute distance to a line, meaning it can change if we multiply $f$ with a constant factor, while $\mathscr{D}_{\text{curv}}$ would not change. However, $\mathscr{D}_{\text{curv}}$ necessitates that the function is twice differentiable, i.e., in $C^2$, in the interval of interest $[-\alpha, \alpha]$. Hence, it cannot distinguish nonlinearity between step functions or absolute values, even if they closely follow a linear function in absolute deviation. In contrast, $\mathscr{D}_{\text{MSE}}$ is finite in such cases but includes an optimization process. However, using linear regression, we can empirically estimate $\mathscr{D}_{\text{MSE}}$ for any given function.

In our work, we generate time series at random. Hence, we are interested in the approximate behavior of the resulting processes. We formalize this by describing families of distributions $\mathcal{F}$ for a function $f$ with stepwise-varying nonlinearity. Specifically, we ensure that sampled functions $f \sim \mathcal{F}$ have controllable expected nonlinearity

$$\mathbb{E}_{f \sim \mathcal{F}}[\mathscr{D}(f))], \tag{14}$$

where $\mathscr{D}$ is a measure of nonlinearity. Thus, to increase the nonlinearity of sampled processes, we stepwise change $\mathcal{F}$ from which we sample the functions $f_{i,d,l}$ in Eq. (2).

Lastly, consider that in our formulation of the general structural causal model (Eq. (2)), the functions in the causal graph $G$ are univariate and connect two, possibly lagged variables. Hence, another approach to increase nonlinearity in a stepwise manner is to sample only a subset of the $f_{i,d,l}$ from a distribution of nonlinear functions, while keeping the rest linear. This leaves us with a third possibility to increase the overall SCM's nonlinearity.

In the following, we specify four families of distributions of nonlinear functions and describe how we can adjust the nonlinearity of the resulting sampled time series processes.

### B.4.3   1. MONOTONIC FAMILY:

For the first family, we sample uniformly from three univariate functions, where a parameter $\beta > 0$ determines the nonlinearity. Before we formally analyze the influence of this parameter, we detail

our specific functions and motivate our design choices. We use

$$f_1(x; \beta) = \text{sgn}(x) \cdot |x|^\beta,$$

$$f_2(x; \beta) = c \left| \frac{(x+1)}{c} \right|^\beta - 1, \quad \text{and} \tag{15}$$

$$f_3(x; \beta) = -c \left| \frac{(x-1)}{c} \right|^\beta + 1,$$

where $c$ is a hyperparameter which we set to 2 in our experiments. The specific choice of the added and subtracted 1 in $f_2$ and $f_3$ depends on our interval of interest $[-\alpha, \alpha]$, where we want to change the non-linearity using $\beta$.

Our intention in designing this family of functions is to ensure monotonicity in the specified interval, hypothesizing that this property is important for CD methods. We investigate this hypothesis empirically in our main paper. As a first step, we now show that all three of our functions are monotonically increasing in $[-1, 1]$ and for $\beta > 0$.

To prove this statement, it is enough to show that the first derivative is greater than or equal to zero for all $x \in [-1, 1]$. Note that given Eq. (2) in our main paper, monotonically decreasing functions are possible because the coefficient matrix $A$ can have negative entries. Hence, without loss of generality, we focus in the following on the monotonically increasing nature. Specifically, consider our three functions (Eq. (15)) only in the interval of interest $[-1, 1]$

$$f_1(x; \beta) = \text{sgn}(x) \cdot |x|^\beta$$

$$f_2(x; \beta) = c \left( \frac{(x+1)}{c} \right)^\beta - 1, \tag{16}$$

$$f_3(x; \beta) = -c \left( -\frac{(x-1)}{c} \right)^\beta + 1,$$

where the absolute value can be removed from $f_2$ and $f_3$ because $x \pm 1$ is always positive or negative, respectively.

We start our analysis with the derivative of $f_1$, which has three cases, i.e., $x < 0$, $x > 0$, and $x = 0$. We start with the first two:

**Case 1,** $-1 \leq x < 0$**:** In this case, $\text{sgn}(x) = -1$ and $|x| = -x$ apply, leading to $f_1(x; \beta) = (-1)(-x)^\beta$. Using the chain rule, we find

$$f_1'(x; \beta) = (-1)\beta(-x)^{\beta-1} \cdot (-1) = \beta(-x)^{\beta-1} \tag{17}$$

$$= \beta|x|^{\beta-1}. \tag{18}$$

**Case 2,** $0 < x \leq 1$**:** Here, the absolute value becomes the identity and $\text{sgn}(x) = 1$. Thus, we have $f_1(x; \beta) =$, and

$$f_1'(x; \beta) = \beta \cdot x^{\beta-1} = \beta|x|^{\beta-1}. \tag{19}$$

In both cases, we can see that the derivative is equal to $f_1'(x; \beta) = \beta|x|^{\beta-1}$. For all $x \in [-1, 1] \setminus \{0\}$ and $\beta > 0$ this is strictly nonnegative. Lastly, to prove that $f_1$ is monotonically increasing in the interval of interest, it is left to show that the derivative is also larger than or equal to zero for $x = 0$. Here, the value depends on the specific setting of $\beta$. For $\beta > 0$, we have three cases and we study the limits of $f_1'$ from both directions

**Case 3.1,** $\beta = 1$**:** In the linear case, we find

$$\lim_{x \to 0^+} f_1'(x; 1) = \lim_{x \to 0^+} 1 \cdot |x|^0 = \lim_{x \to 0^+} 1 = 1, \text{ and}$$

$$\lim_{x \to 0^-} f_1'(x; 1) = \lim_{x \to 0^-} 1 \cdot |x|^0 = \lim_{x \to 0^-} 1 = 1.$$

In other words, the derivative is constant and larger than zero.

**Case 3.2, $\beta > 1$:** Here we find the exponent $\beta - 1 > 0$ leading to the following two limits

$$\lim_{x \to 0^+} f_1'(x; 1) = \lim_{x \to 0^+} \beta \cdot |x|^{\beta - 1} = \beta \cdot |0|^{\beta - 1} = 0, \text{ and}$$

$$\lim_{x \to 0^-} f_1'(x; 1) = \lim_{x \to 0^-} \beta \cdot |x|^{\beta - 1} = \beta \cdot |0|^{\beta - 1} = 0.$$

Hence, we find a saddle point, where the rate of change is exactly zero when $x = 0$.

**Case 3.3, $0 < \beta < 1$:** In this case, the exponent $\beta - 1$ becomes negative meaning $|x|^{\beta - 1} = {}^1/_{|x|^{1 - \beta}}$. Consequently, limits from both sides diverge towards

$$\lim_{x \to 0^+} f_1'(x; 1) = \lim_{x \to 0^+} \beta \cdot \frac{1}{|x|^{1 - \beta}} = +\infty, \text{ and}$$

$$\lim_{x \to 0^-} f_1'(x; 1) = \lim_{x \to 0^-} \beta \cdot \frac{1}{|x|^{1 - \beta}} = +\infty.$$

Crucially, in all three cases, the limits of the derivative from both sides are equal and strictly nonnegative. Hence, $f_1$ is monotonically increasing in $[-1, 1]$ for all $\beta > 0$.

Next, we investigate the derivatives of $f_2$ and $f_3$. Following the observation that for $x \in [-1, 1]$ the absolute values can be rewritten as in Eq. (16), we calculate $f_2'$ and $f_3'$ using the chain rule as

$$f_2'(x; \beta) = \beta \left( \frac{x + 1}{c} \right)^{\beta - 1}, \quad \text{and}$$

$$f_3'(x; \beta) = \beta \left( \frac{1 - x}{c} \right)^{\beta - 1}. \tag{20}$$

For both functions, we have a strictly positive number ($\beta > 0$) which is multiplied by a base raised to a real power. Remember that in our experiments, we set $c = 2$, meaning both ${}^{(x-1)}/_2$ and ${}^{(1-x)}/_2$ vary in $[0, 1]$, i.e., are strictly nonnegative. Therefore, raising it to the real power ($\beta - 1$) leads, for both $f_2'$ and $f_3'$, to a positive factor times a nonnegative factor. Hence, for all $x \in [-1, 1]$ and $\beta > 0$, we find $f_2'(x; \beta) \geq 0$ and $f_3'(x; \beta) \geq 0$. Note that in both cases, when $0 < \beta < 1$, we again find limits for both derivatives, where they become infinite. Specifically, for $f_2$, we observe a vertical tangent when $x = -1$, and for $f_3$, we similarly observe one for $x = 1$ (compare to case 3.3 of $f_1$). Nevertheless, the derivatives of all three functions $f_1$, $f_2$, and $f_3$ are strictly nonnegative in the specified interval. Hence, the functions themselves are monotonically increasing in $[-1, 1]$ for all $\beta > 0$. Next, we discuss how we can increase the nonlinearity of all three functions.

**Monotonic Family Increasing Nonlinearity** As specified above, for a given distribution of functions, we can quantify the linearity by considering the corresponding expectation. For the monotonic family of functions, the distribution we consider is a uniform choice of $\{f_i, f_2, f_3\}$. Hence, for a fixed $\beta$, we are interested in

$$\mathbb{E}_{f_j \sim \text{Uniform}\{f_1, f_2, f_3\}} [\mathscr{D}_{\text{MSE}}(f_j(\,\cdot\,; \beta))]. \tag{21}$$

In this uniform distribution, all three cases are equally likely. Thus, the expectation for a fixed $\beta$ is equal to the average of $\mathscr{D}_{\text{MSE}}$ for the three functions. We specifically choose $\mathscr{D}_{\text{MSE}}$ because the integral over the squared second derivative ($\mathscr{D}_{\text{curv}}$) of $f_1$ diverges for $1 < \beta < 1.5$. Further, we are interested in measuring the squared deviation from any possible line in $[-1, 1]$.

Consider that the parameter $\beta$ directly controls the nonlinearity of $f_1$, $f_2$, and $f_3$. In particular, all three functions are equal and linear in $[-1, 1]$ if $\beta = 1$

$$f_1(x; 1) = \text{sgn}(x) \cdot |x| = x,$$

$$f_2(x; 1) = c \left( \frac{x + 1}{c} \right) - 1 = x,$$

$$f_3(x; 1) = -c \left( -\frac{x - 1}{c} \right) + 1 = x.$$

Hence, the expectation in Eq. (21) becomes zero for $\beta = 1$.

Now, by changing $\beta$ away from 1, all three functions become nonlinear in the sense of $\mathscr{D}_{\text{MSE}}$. Specifically, we construct five discrete levels $\ell \in \{1, 2, 3, 4, 5\}$ to scale $\mathbf{V}_{\text{nl,mono}}$ and sample $\beta$ with an equal chance from either of the following intervals

$$
\begin{aligned}
\ell = 1 &\to \beta \in [1/2, 1] \text{ or } \beta \in [1, 2], \\
\ell = 2 &\to \beta \in [1/4, 1/2] \text{ or } \beta \in [2, 4], \\
\ell = 3 &\to \beta \in [1/8, 1/4] \text{ or } \beta \in [4, 8], \\
\ell = 4 &\to \beta \in [1/12, 1/8] \text{ or } \beta \in [8, 12], \\
\ell = 5 &\to \beta \in [1/20, 1/12] \text{ or } \beta \in [12, 20].
\end{aligned}
\tag{22}
$$

For a concrete level $\ell$, we denote the lower and upper boundaries of the two intervals with $[\beta_L^{(\ell\downarrow)}, \beta_U^{(\ell\downarrow)}]$ and $[\beta_L^{(\ell\uparrow)}, \beta_U^{(\ell\uparrow)}]$, respectively. Fig. 13 visualizes examples of functions drawn from the five levels of the resulting distributions. Crucially, the intervals of the distinct levels only overlap at a maximum of two concrete boundary points with any of the other intervals.

To analyze the non-linearity of our functions $f_j(\cdot; \beta)$ in the interval $x \in [-1, 1]$, we consider a second expectation over $\beta$ distributed uniformly from either of the two intervals given by a level $\ell$, i.e.,

$$
\mathbb{E}_\beta[\mathbb{E}_{f_j}[\mathscr{D}_{\text{MSE}}(f_j(\cdot; \beta))]],
\tag{23}
$$

where we omit the specific distributions for brevity.

Here, both the function $f_j$ and $\beta$ are sampled independently. Hence, Eq. (23) is equal to

$$
\begin{aligned}
&\int_{\beta_L^{(\ell\downarrow)}}^{\beta_U^{(\ell\downarrow)}} \frac{1}{2(\beta_U^{(\ell\downarrow)} - \beta_L^{(\ell\downarrow)})} \sum_{j=1}^{3} \frac{1}{3} \mathscr{D}_{\text{MSE}}(f_j(\cdot; \beta)) d\beta \\
&+ \int_{\beta_L^{(\ell\uparrow)}}^{\beta_U^{(\ell\uparrow)}} \frac{1}{2(\beta_U^{(\ell\uparrow)} - \beta_L^{(\ell\uparrow)})} \sum_{j=1}^{3} \frac{1}{3} \mathscr{D}_{\text{MSE}}(f_j(\cdot; \beta)) d\beta,
\end{aligned}
\tag{24}
$$

where both of the integrals describe one of the two equally likely and symmetrical intervals from which $\beta$ is sampled, respectively. Further, for a fixed level $\ell$, the factors contained in the intervals are a fixed normalization given by the probability density of the corresponding uniform distributions over the intervals $[\beta_L^{(\ell\downarrow)}, \beta_U^{(\ell\downarrow)}]$ and $[\beta_L^{(\ell\uparrow)}, \beta_U^{(\ell\uparrow)}]$.

By linearity of expectation, we can reorder Eq. (24) into

$$
\begin{aligned}
\frac{1}{3} \sum_{j=1}^{3} \Bigg( &\int_{\beta_L^{(\ell\downarrow)}}^{\beta_U^{(\ell\downarrow)}} \frac{\mathscr{D}_{\text{MSE}}(f_j(\cdot; \beta))}{2(\beta_U^{(\ell\downarrow)} - \beta_L^{(\ell\downarrow)})} d\beta \\
&+ \int_{\beta_L^{(\ell\uparrow)}}^{\beta_U^{(\ell\uparrow)}} \frac{\mathscr{D}_{\text{MSE}}(f_j(\cdot; \beta))}{2(\beta_U^{(\ell\uparrow)} - \beta_L^{(\ell\uparrow)})} d\beta \Bigg).
\end{aligned}
\tag{25}
$$

As stated above, the only point in all intervals we consider where the $f_j$ are linear is for $\beta = 1$. In any other case, $\mathscr{D}_{\text{MSE}}(f_j(\cdot; \beta)) > 0$ applies.

To now show that the expectation increases with the level $\ell$, consider that the integrals in Eq. (25) calculate averages over all values of $\mathscr{D}_{\text{MSE}}$ for $\beta$ in the corresponding intervals. Hence, it is enough to show that for $\beta > 0$, $\mathscr{D}_{\text{MSE}}$ is smooth and increases when moving away from the global minimum at $\beta = 1$ in our specified intervals. In all cases, we focus our analysis on the set of functions $\{f_1, f_2, f_3\}$ we defined above. Consider that for the interval $[-1, 1]$, the optimal parameter for the MSE minimizing line (Eq. (12)), become

$$
\begin{aligned}
a^* &= \frac{3}{2} \int_{-1}^{1} x f(x) dx, \quad \text{and} \\
b^* &= \frac{1}{2} \int_{-1}^{1} f(x) dx.
\end{aligned}
\tag{26}
$$

We visualize examples for $f_1$, $f_2$, and $f_3$ and the corresponding optimal lines in Fig. 13.

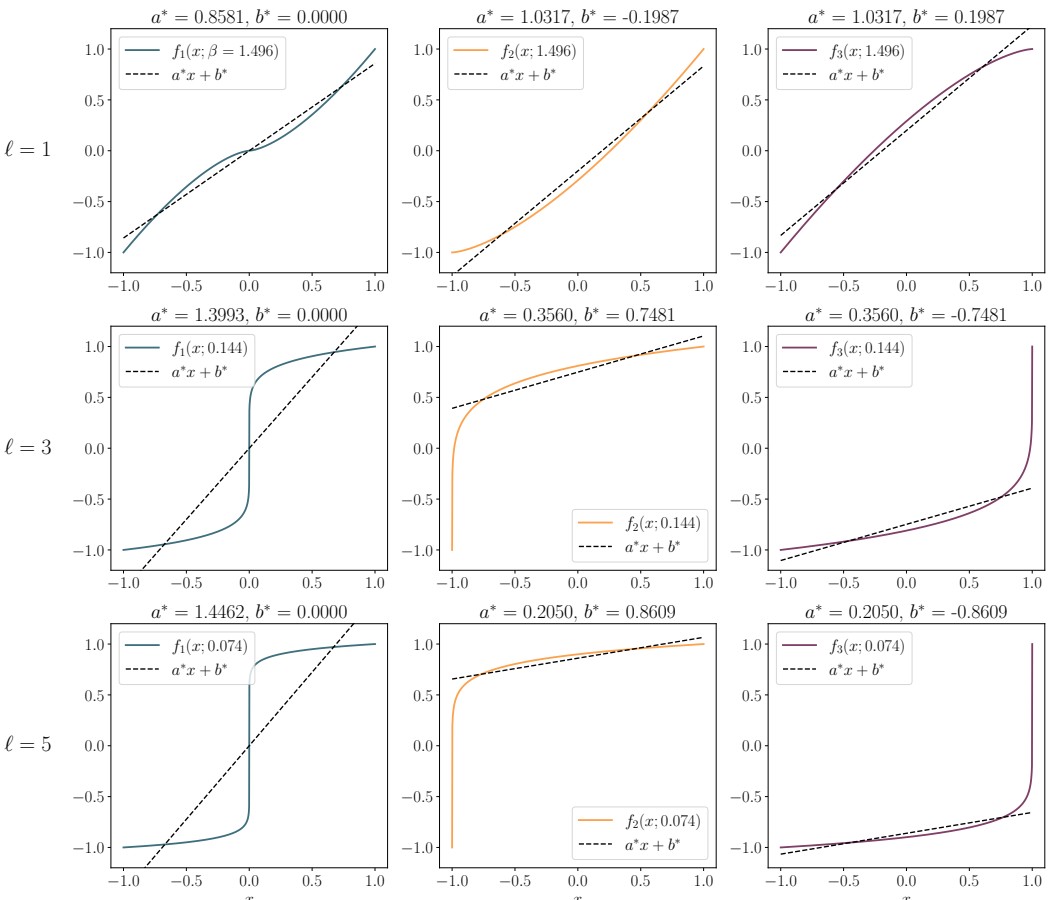

Figure 13: Examples of $f_1$, $f_2$, and $f_3$ with randomly sampled $\beta$ from the respective level $\ell$. We also visualize the optimal line and denote the corresponding parameters.

Now to determine whether $\mathscr{D}_{\text{MSE}}$ is smooth with respect to changes in $\beta$, we have to consider the three terms in Eq. (13) that are functions of $\beta$: $f_j$, $a^*$, and $b^*$, where the last two also depend on the specific function $f_j$ (Eq. (12), Eq. (26)).

For all three of our functions $f_j$, the critical part is of the form $|g(x)|^\beta$, where $g(\cdot)$ is defined in Eq. (15). Hence, the following equality holds

$$|g(x)|^\beta = e^{\beta \ln |g(x)|}. \tag{27}$$

In particular, the function has an exponential form, which is infinitely differentiable ($C^\infty$) with respect to $\beta$ for any fixed $x$ (where $g(x) \neq 0$). In other words, $f_1$, $f_2$, and $f_3$ are smooth with respect to $\beta$ in $[-1, 1]$. Further, as a direct consequence of the Leipnitz integral rule, e.g., (Protter & Morrey, 1985, Chap. 8), integrals of the form $\int_{-1}^{1} f_j(x; \beta) dx$ and $\int_{-1}^{1} x f_j(x; \beta) dx$ are also smooth functions with respect to $\beta$. Finally, consider that $\mathscr{D}_{\text{MSE}}$ is again an integral with respect to $x$ of a square of the sum of three functions that are smooth with respect to $\beta$. Hence, using the Leipnitz integral rule and the chain rule for differentiation, we can conclude that $\mathscr{D}_{\text{MSE}}$ is also smooth in $\beta$ for the functions $f_1$, $f_2$, and $f_3$.

To show that the nonlinearity increases if we shift $\beta$ away from the linear case of $\beta = 1$, we now study $\frac{\partial}{\partial\beta}\mathscr{D}_{\text{MSE}}$. In particular, using the Leibniz rule and the chain rule, we have

$$\frac{\partial\mathscr{D}_{\text{MSE}}(f_j)}{\partial\beta} = \int_{-1}^{1} \left( f_j(x;\beta) - (a^*x + b^*) \right) \\ \cdot \left( \frac{\partial f_j(x;\beta)}{\partial\beta} - x\frac{\partial a^*}{\partial\beta} - \frac{\partial b^*}{\partial\beta} \right) dx. \tag{28}$$

Expanding the product in the integral leaves us with three separate terms

$$\frac{\partial\mathscr{D}_{\text{MSE}}(f_j)}{\partial\beta} = \int_{-1}^{1} \left( f_j(x;\beta) - (a^*x + b^*) \right)\frac{\partial f_j(x;\beta)}{\partial\beta}dx \\ -\frac{\partial a^*}{\partial\beta} \int_{-1}^{1} x(f_j(x;\beta) - a^*x - b^*)dx \\ -\frac{\partial b^*}{\partial\beta} \int_{-1}^{1} (f_j(x;\beta) - a^*x - b^*)dx. \tag{29}$$

Crucially, note that it is possible to move the partial derivatives $\frac{\partial a^*}{\partial\beta}$ and $\frac{\partial b^*}{\partial\beta}$ outside of the integral because they are independent of $x$. This is important because the integrals in the second and third terms are exactly the first-order optimality conditions of $a^*$ and $b^*$, respectively Emancipator & Kroll (1993). Hence, both of these integrals vanish, and we are left with

$$\frac{\partial\mathscr{D}_{\text{MSE}}(f_j)}{\partial\beta} = \int_{-1}^{1} \left( f_j(x;\beta) - (a^*x + b^*) \right)\frac{\partial f_j(x;\beta)}{\partial\beta}dx. \tag{30}$$

In Eq. (30), we have two factors: the residual error to the MSE optimal line and the sensitivity of $f_j$ with respect to changes in $\beta$. Given that the residual is a constant zero at $\beta = 1$ when our $f_j$ become linear, we again confirm that this is a minimum of $\mathscr{D}_{\text{MSE}}$. Hence, $\frac{\partial\mathscr{D}_{\text{MSE}}(f_j)}{\partial\beta} = 0$ if $\beta = 1$. Consider now that for all values $\beta > 0$ which are not $\beta = 1$, all our functions $f_1$, $f_2$, and $f_3$ are nonlinear, we know that $\mathscr{D}_{\text{MSE}}$ has to be strictly larger than zero. This implies that $\beta = 1$ is a unique global minimum. Given this observation and the previous insight that $\mathscr{D}_{\text{MSE}}$ is smooth with respect to $\beta$, we conclude that the averages over $\mathscr{D}_{\text{MSE}}$ have to increase locally in the neighborhood of the global minimum at $\beta = 1$. However, this does not necessarily imply that the only critical point is at $\beta = 1$. To test whether our claim that the expected nonlinearity increases for an increase in level $\ell$, we use Eq. (12) and Eq. (13) to simulate the nonlinearity.

We visualize $\mathscr{D}_{\text{MSE}}$ in Fig. 14 and confirm that the nonlinearity increases when we move away from the global minimum $\beta = 1$. However, we observe local maxima in the interval $(0, 20]$. Hence, it is unclear how the expected nonlinearity for randomly sampled $f_j$ and $\beta$ according to the defined levels $\ell$ behaves.

| $\ell$ | $f_1$ | $f_2$ | $f_3$ | $\mathbb{E}[\mathscr{D}_{\text{MSE}}(f_j)]$ |
|---|---|---|---|---|
| 1 | 0.005178 | 0.006049 | 0.005783 | 0.005670 |
| 2 | 0.032103 | 0.030878 | 0.029993 | 0.030991 |
| 3 | 0.066364 | 0.046946 | 0.048134 | 0.053815 |
| 4 | 0.087101 | 0.046346 | 0.044872 | 0.059440 |
| 5 | 0.103752 | 0.039548 | 0.037819 | 0.060373 |

Table 4: Approximated nonlinearity scores for the three functions $f_1$, $f_2$, and $f_3$ and different levels $\ell$. The last column contains the accumulated $\mathscr{D}_{\text{MSE}}$ over all $f_j$ for all $\beta$ sampled in the respective level.

Thus, we estimate the expected nonlinearity (Eq. (25)) per level $\ell$. Specifically, we sample 1000 $\beta$ values for each level and use the theoretically optimal line parameters $a^*$ and $b^*$. We list the approximated expected nonlinearity in Table 4. We find that the expected $\mathscr{D}_{\text{MSE}}$ does stepwise increase for $f_1$ while it decreases slightly for $f_2$ and $f_3$ again after $\ell = 3$. However, the accumulated expectation over all functions (Eq. (23)) does grow for $\ell = 1, ..., 5$. Therefore, we conclude that the empirical nonlinearity does increase stepwise for our defined violation levels.

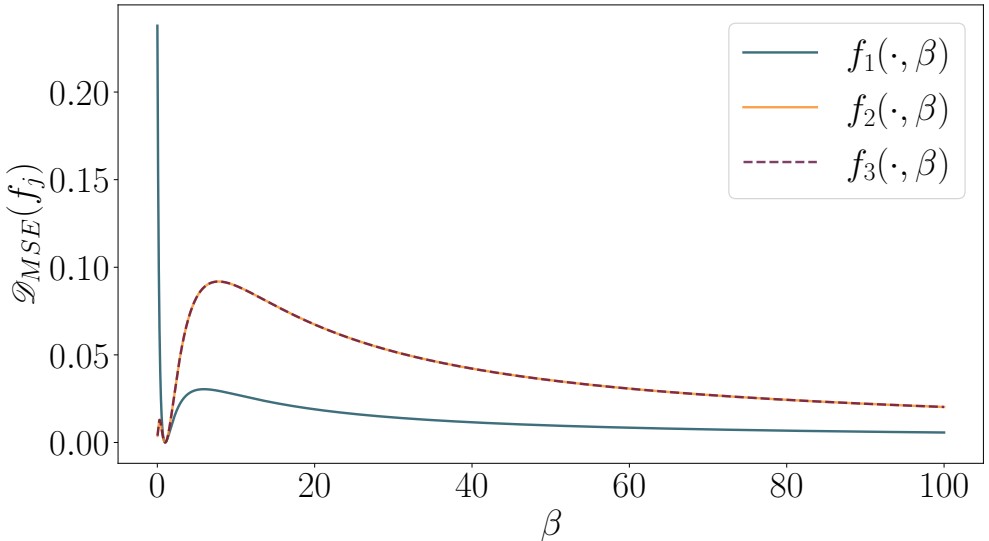

Figure 14: Nonlinearity measured with $\mathscr{D}_{\text{MSE}}$ for the three functions $f_1$, $f_2$, and $f_3$ for increasing values of $\beta > 0$.

### B.4.4    2. B-SPLINES FOLLOWING A TREND:

Next, we investigate univariate functions $f$ that exhibit an overall increasing trend but are not necessarily monotonic. To do this, we rely on B-spline interpolations, e.g, de Boor (2001). Specifically, we sample sample $N_P$ scalar values (interpolation points) $\{v_1, v_2, ..., v_{N_P}\}$ from a uniform distribution Uniform$(-1, 1)$. Next, we sort the values $v_j$ and set them as targets for $f(x)$ at equidistant abscissae in the range $x \in [-1, 1]$. Consequently, a B-spline $f(x) = \sum_{j=1}^{N_P} c_k B_{j,k}(x)$ of degree $k = 3$ is constructed to smoothly interpolate these points. The corresponding B-spline basis elements are given by

$$B_{j,0}(x) = \begin{cases} 1 & \text{if } \tau_j \leq x < \tau_{j+1}, \\ 0 & \text{else} \end{cases}$$

$$B_{j,k}(x) = \frac{x - \tau_j}{\tau_{j+k} - \tau_j} B_{j,k-1}(x)$$
$$+ \frac{\tau_{j+k+1} - x}{\tau_{j+k+1} - \tau_{j+1}} B_{j+1,k-1}(x),$$

where we determine the entries of the knot vector $\tau$ as described in de Boor (2001) using the implementation provided in Virtanen et al. (2020). Given that the basis functions are piecewise cubic, the fitted curve is not monotonic, as visualized in Fig. 7b. To ensure saturation, we use the procedure described in Eq. (8). For $\mathbf{V}_{\text{nl,trend}}$, we sample different $f_{i,d,l}$ for all non-zero interactions in Eq. (2) by sampling different $v_j$. Next, we discuss how we stepwise increase in nonlinearity of the sampled functions.

**B-Splines Increasing Nonlinearity**    To stepwise scale the nonlinearity of the sampled functions $f_{i,d,l}$, we decrease the number $N_P$ of interpolation points. Specifically, we use the following values: $\{25, 15, 10, 6, 4\}$. Intuitively, a higher number of interpolation points indicates more values $v_j$ that have to be interpolated and which are strictly increasing. To empirically show that a larger $N_P$ leads to more linear functions in the sense of $\mathscr{D}_{\text{MSE}}$ (Eq. (13)), we use Eq. (12) to approximate the average nonlinearity in Fig. 15. Further, following the uniform distribution of independently sampled $v_j$, we approximate the expected $\mathscr{D}_{\text{MSE}}$ of our concrete $\mathbf{V}_{\text{nl,trend}}$ in Table 5. Similar to the monotonic family of functions, we observe that the nonlinearity of sampled $f_{i,d,l}$ increases on average.

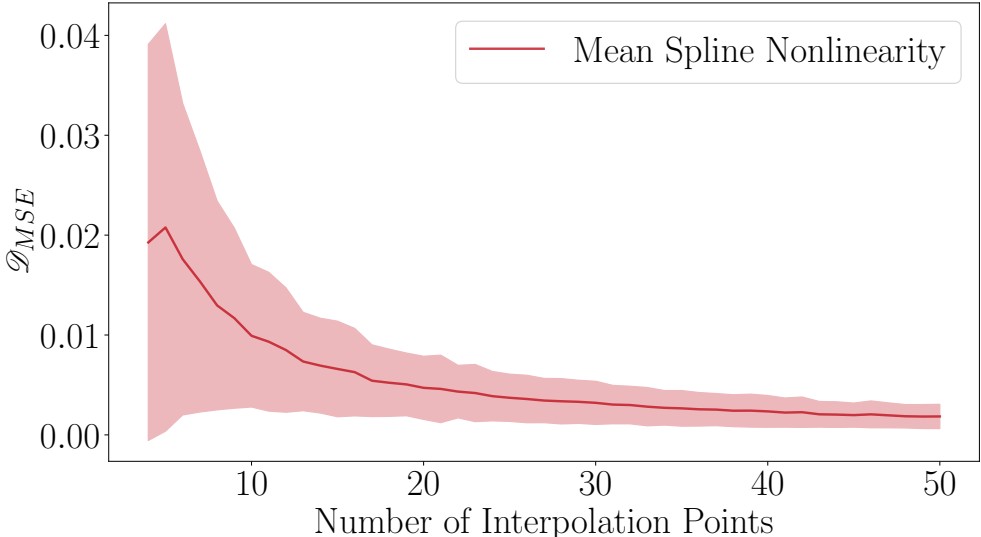

Figure 15: Nonlinearity measured with $\mathscr{D}_{\mathrm{MSE}}$ for the spline functions for an increasing number of interpolation points. We report the average and the standard deviations.

| Number of Inter-polation Points | $\mathbb{E}_{f_{\mathrm{spline}}}[\mathscr{D}_{\mathrm{MSE}}(f_{\mathrm{spline}})]$ |
|---|---|
| 25 | 0.003792 |
| 15 | 0.006517 |
| 10 | 0.010198 |
| 6 | 0.017728 |
| 4 | 0.019475 |

Table 5: Estimated expected $\mathscr{D}_{\mathrm{MSE}}$ for the spline functions sampled for the five decreasing number of interpolation points.

### B.4.5   3. GAUSSIAN PROCESSES WITH RBF KERNELS:

Related studies focusing on the robustness of CD methods for i.i.d. sample data Montagna et al. (2023a); Yi et al. (2025) primarily replace standard linear relationships with interactions modeled by Gaussian processes using Radial Basis Function (RBF) kernels. In our third family of functions for the violation $\mathbf{V}_{\mathrm{nl,rbf}}$, we follow the same approach. Specifically, we use a Gaussian process prior $f(x) \sim \mathcal{GP}(0, \kappa_{\mathrm{RBF}}(x, x'))$ with the RBF kernel $\kappa_{\mathrm{RBF}}(x, x') = \exp\left(-\frac{(x-x')^2}{2\lambda^2}\right)$, where $\lambda$ is the length scale which we set to one. In particular, we use the implementation of Pedregosa et al. (2011).

Further, we do not employ wrapping in this case, as the Gaussian process's mean zero ensures that the sampled functions do not diverge. Next, we describe how we stepwise increase the nonlinearity of the resulting SCM.

**Stepwise Increasing Nonlinearity**   To stepwise increase the nonlinearity, we use a different approach for $\mathbf{V}_{\mathrm{nl,rbf}}$. Specifically, we sample nonlinear links $f_{i,d,l}$ with an increasing probability from the Gaussian process. Here, we use the following probabilities: $\{0.2, 0.4, 0.6, 0.8, 1.0\}$. All remaining links in the causal graph $G$ are linear and use the identity function in Eq. (2). Hence, we increase the overall SCM's nonlinearity by increasing the likelihood of sampling nonlinear links. To be specific, for our two empirical scenarios $(D, L) = (5, 3)$ and $(D, L) = (7, 4)$, we have a maximum of $5 \times 5 \times (3 + 1)$ and $7 \times 7 \times (4 + 1)$ links, respectively, where $D$ is the number of variables and $L$ the number of lags (+1 for instantaneous). For lagged links, we employ the link probabilities

$p_{\text{lag}} = 0.075$ or $p_{\text{lag}} = 0.15$, while for instantaneous links, we use $p_{\text{inst}} = 0.0$ and $p_{\text{inst}} = 0.1$ (see Eq. (7)).

We list the number of expected links for the eight combinations in Table 9. For each of these eight combinations, we can calculate the number of expected nonlinear links for the increasing intensities $\{0.2, 0.4, 0.6, 0.8, 1.0\}$ via a standard multiplication because they are sampled independently. This leads to the following increasing numbers of expected nonlinear links for the smallest and sparsest scenario ($D = 5, L = 3, p_{\text{lag}} = 0.075, p_{\text{inst}} = 0.0$): $\{1.125, 2.25, 3.375, 4.5, 5.625\}$. Conversely, we get the following expected nonlinear links in the largest scenario ($D = 7, L = 4, p_{\text{lag}} = 0.15, p_{\text{inst}} = 0.1$): $\{6.86, 13.72, 20.58, 27.44, 34.3\}$. Hence, we conclude that sampling the SCM in this stepwise manner progressively increases the amount of nonlinear interactions.

### B.4.6 4. COMPOSITE FUNCTIONS

Lastly, and inspired by symbolic regression, we sample the $f_{i,d,l}$ through a random hierarchical composition. First, we define a set of base functions $\mathfrak{B}$. Specifically, we implement $\{x^{1/3}, \tanh(x), \sinh^{-1}(x), \max(x, 0), x, x^2, |x|, \cosh(x), \sin(x), cos(x)\}$. Then, $m$ independent chains $h^{(j)}(x)$ are formed, each by randomly selecting and sequentially composing $N_\beta$ functions from $\mathfrak{B}$, i.e., $h^{(j)}(x) = b_{N_\beta}(\ldots b_2(b_1(x)) \ldots)$. Finally, the results of the independent chains get multiplied by $-1$ with a probability of $1/2$, i.e., $c_{\text{flip}}^{(j)} \sim \text{Uniform}\{-1, 1\}$, before all chains get summed up to

$$f(x) = \sum_{j=1}^{m} c_{\text{flip}}^{(j)} \cdot h^{(j)}(x).$$

This construction allows for a wide range of possible, potentially highly nonlinear functions. Hence, we apply Eq. (9) to enforce stable behavior. In our empirical evaluation, we use two chains, each composed of two base functions uniformly sampled from $\mathfrak{B}$ to model $\mathbf{V}_{\text{nl,comp}}$. Next, we describe how we stepwise increase the nonlinearity of the resulting SCM.

**Stepwise Increasing Nonlinearity** For $\mathbf{V}_{\text{nl,comp}}$, we strictly follow the procedure also employed for $\mathbf{V}_{\text{nl,rbf}}$. Specifically, we increase the probability of sampling nonlinear interactions when generating the SCM. Again, we use the following probabilities: $\{0.2, 0.4, 0.6, 0.8, 1.0\}$ to scale the violation intensity. Table 9 summarizes the number of links for the various scenarios in our experiments, and the same calculations as for $\mathbf{V}_{\text{nl,rbf}}$ apply to estimate the expected number of nonlinear links. Hence, we conclude that sampling the SCM in this stepwise manner progressively increases the amount of nonlinear interactions.

### B.5 ADDITIONAL DETAILS $\mathbf{V}_{\text{INNO}}$

The standard assumption of independent additive innovation noise is often violated in practice. Here, we detail our setup for evaluating the impact of such violations. Specifically, we consider three different paradigms:

First, we discuss direct changes to the noise structures by introducing dependencies. Here, we also deploy real-world time series as direct innovation noise components, relying on "Rivers - flood," a set of river discharge time series from Stein et al. (2024a). Second, we shift the distribution from Gaussian to non-Gaussian variations. Third, we consider widely different variances, leading to stronger variation among the variables $X_i$, which can be problematic Peters & Bühlmann (2014). In the first set of violations, we test robustness to six alternative noise structures, as discussed in Apx. B.1. To be specific, implement $\mathbf{V}_{\text{inno,mul}}, \mathbf{V}_{\text{inno,auto}}, \mathbf{V}_{\text{inno,com}}, \mathbf{V}_{\text{inno,time}}, \mathbf{V}_{\text{inno,shock}}$ and $\mathbf{V}_{\text{inno,real}}$.

All of these structures change the distributions of the independent additive noise $\epsilon_{i,t}$ in Eq. (2). We list the specific distributions in Table 6. Regarding the corresponding hyperparameters, i.e., $\alpha$, $\beta$, $S$, and $p_{\text{shock}}$, we use the same setting as for the observational variants (see Apx. B.1).

In contrast to scaling the SNR, we blend the different $\epsilon_{i,t}$ distributions with a decreasing amount of standard normal noise to intensify the five violations. This is important because innovation noise is part of the signal (Eq. (2)) and relying solely on the specified noise can lead to arbitrarily low signal

| Violation | Definition of $\epsilon_{i,t}$ | Depends On |
|---|---|---|
| $\mathbf{V}_{\text{inno,mul}}$ | $\epsilon_{i,t} = X_{i,t} \cdot \eta_{i,t}$ | the signal $X_{i,t}$ |
| $\mathbf{V}_{\text{inno,time}}$ | $\epsilon_{i,t} = \eta_{i,t} \cdot (1 + \alpha t) \cdot \sin(2\pi t/\beta)$ | time step $t$ |
| $\mathbf{V}_{\text{inno,auto}}$ | $\epsilon_{i,t} = \alpha \cdot \epsilon_{i,t-1} + (1 - \alpha) \cdot \eta_{i,t}$ | autoregressive |
| $\mathbf{V}_{\text{inno,com}}$ | $\forall i : \epsilon_{i,t} = \eta_t$ | — |
| $\mathbf{V}_{\text{inno,shock}}$ | $\epsilon_{i,t} \sim \begin{cases} S & \text{with prob. } p_{\text{shock}}, \\ 0 & \text{else.} \end{cases}$ | fixed scalar $S$, shock prob. $p_{\text{shock}}$ |
| $\mathbf{V}_{\text{inno,real}}$ | $z_{i,t}$ | exog. real-world TS $Z^{I,T}$ |

Table 6: First set of innovation noise violations. In our experiments, the sources of randomness for the variables $X_i$ are routed in the innovation noise $\epsilon_{i,t}$, which are typically additive and standard normal. Here, we include various dependencies by using random variables $\eta_{i,t}$ and $\eta_t$ $\big($standard normal $\mathcal{N}(0,1)\big)$, which are subsequently influenced by various factors, e.g., the signal strength ($\mathbf{V}_{\text{inno,mul}}$). Both $\alpha$ and $\beta$ denote hyperparameters.

variances. In particular, we use alpha blending, i.e.,

$$\alpha\epsilon_{i,t} + (1 - \alpha)\varepsilon_{i,t}, \tag{31}$$

where $\varepsilon_{i,t} \sim \mathcal{N}(0,1)$. We specify the $\alpha$ values that we use for all innovation noise violation types in Table 3.

For the second set of innovation noise violations, i.e., $\mathbf{V}_{\text{inno,uni}}$ and $\mathbf{V}_{\text{inno,weib}}$, we progressively blend non-Gaussian distributed noise with standard normal noise. Let $\omega \sim \Omega_{ng}$ be a non-Gaussian random variable and let $\psi \sim \mathcal{N}(0,1)$ be standard normal noise. Then, we define $\epsilon_{i,t}$ as

$$\epsilon_{i,t} = \frac{(1 - \alpha)(\omega - \mathbb{E}[\omega]) + \alpha\psi}{\sqrt{\text{var}(\omega)(1 - 2\alpha) + \alpha^2(\text{var}(\omega) + 1)}}, \tag{32}$$

where $\alpha$ is a blending parameter. To stepwise scale the intensity, we use the following blending values for $\alpha$: $\{0.2, 0.4, 0.6, 0.8, 1.0\}$.

The denominator in Eq. (32) and substracting $\mathbb{E}[\omega]$ in the numerator ensure that $\mathbb{E}[\epsilon_{i,t}] = 0$ and $\text{var}(\epsilon_{i,t}) = 1$. To verify this, note that the expectation is linear and the denominator is a constant factor for a fixed $\omega$. Hence, it is enough to analyze the numerator. Here, $\mathbb{E}[\omega] - \mathbb{E}[\mathbb{E}[\omega]] = \mathbb{E}[\omega] - \mathbb{E}[\omega] = 0$ and $\mathbb{E}[\psi]$, which directly implies $\mathbb{E}[\epsilon_{i,t}] = 0$. Further, to show that the denominator scales the mixture to unit variance, we have to show that it is equivalent to the standard deviation of the numerator. Given that the standard deviation is the square root of the variance, it is enough to show that the variance of the numerator is equal to the squared denominator. Crucially, both $\omega$ and $\psi$ are independent random variables, which means their covariance is zero. Hence, $\text{var}((1 - \alpha)(\omega - \mathbb{E}[\omega]) + \alpha\psi)$

$$= (1 - \alpha)^2\text{var}(\omega) + \alpha^2\text{var}(\psi)$$
$$= (1 - 2\alpha + \alpha^2)\text{var}(\omega) + \alpha^2 1$$
$$= \text{var}(\omega)(1 - 2\alpha) + \alpha^2(\text{var}(\omega) + 1),$$

i.e., the squared denominator in Eq. (32). A direct consequence of this is that $\text{var}(\epsilon_{i,t}) = 1$.

In our experiments, we use two different non-Gaussian distributions to model the violations $\mathbf{V}_{\text{inno,uni}}$ and $\mathbf{V}_{\text{inno,weib}}$. In the first case, for $\mathbf{V}_{\text{inno,uni}}$, we use a uniform distribution over the interval $[-2, 2]$ with the corresponding density

$$p_{\text{Uniform}}(x) = \mathbf{1}_{[-2,2]}(x) \cdot \frac{1}{4}, \tag{33}$$

where $\mathbf{1}_{[-2,2]}$ is a unit function that is equal to one iff $x \in [-2, 2]$. In the second case, for $\mathbf{V}_{\text{inno,weib}}$, we employ a Weibull distribution Weibull (1939). which is described by two parameters: a scale $\lambda$, which we set to one, and shape $a$, where we use 1.5. The corresponding density is defined as

$$p_{\text{Weibull}}(x) = \frac{a}{\lambda} \left(\frac{x}{\lambda}\right)^{a-1} e^{-(x/\lambda)^a}. \tag{34}$$

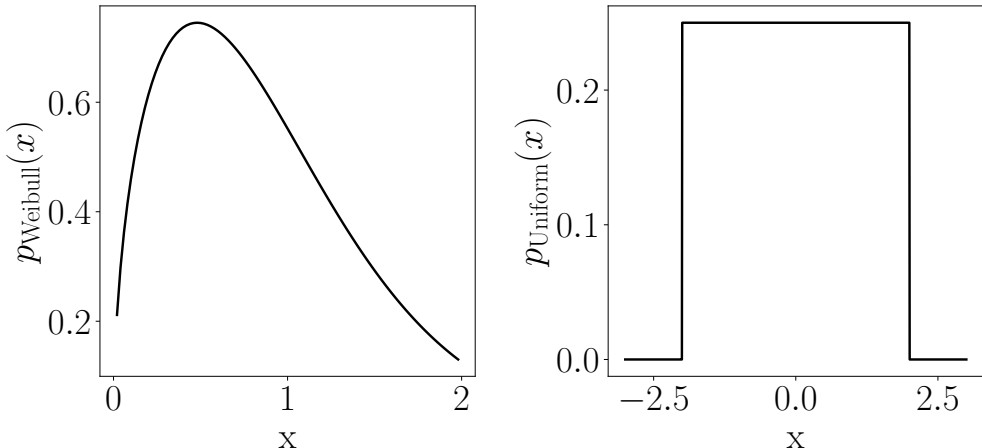

Figure 16: The densities of the two non-Gaussian distributions, we employ to violate standard normal Gaussian innovation noise. Specifically, we employ a Weibull distribution Weibull (1939) with scale $\lambda = 1$ and shape $a = 1.5$ and a uniform distribution over the interval $[-2, 2]$.

We visualize the densities of both non-Gaussian distributions in Fig. 16

Lastly, as a final violation concerning innovation noise, which is known to be potentially problematic Peters & Bühlmann (2014), we introduce the equal variance violation ($\mathbf{V}_{\text{inno,var}}$). We implement this violation by sampling a variance $\sigma_i^2$ for each variable $X_i$ at the initialization of the sampling process. This value is used in every time step to draw innovations $\epsilon_{i,t} \sim \mathcal{N}(0, \sigma_i^2)$. In particular, we sample $\sigma_i^2$ uniformly from the union of two disjoint intervals. To stepwise intensify $\mathbf{V}_{\text{inno,var}}$, we increase the separation distance between these intervals. Specifically, we use the following order of interval unions to model the widening distributions:

- $[0.55, 0.75] \cup [1.25, 1.45]$,
- $[0.4125, 0.6125] \cup [1.3875, 1.5875]$,
- $[0.275, 0.475] \cup [1.525, 1.725]$,
- $[0.1375, 0.3375] \cup [1.6625, 1.8625]$, and
- $[0, 0.2] \cup [1.8, 2]$.

### B.6 ADDITIONAL DETAILS $\mathbf{V}_{\text{COEF}}$, $\mathbf{V}_{\text{STAT}}$

Stationarity assumes that the structural assignments in Eq. (7) do not change during the generation/measurement process. First, for $\mathbf{V}_{\text{coef}}$, we keep the nonzero entries in $A$ fixed, but resample the coefficients during the generation of the time series. To increase the severity of the violation, we stepwise allow larger changes ($\delta$) to the current coefficients and sample the actual changes uniformly from $(-\delta, \delta)$. In total, we resample the coefficients 4 times at $S_{\text{change}} \in \{50, 100, 150, 200\}$ for $T = 250$ and $S_{\text{change}} \in \{200, 400, 600, 800\}$ for $T = 1000$. Second, for $\mathbf{V}_{\text{stat}}$, we fully resample $A$ at certain change points in the generation process. To scale this violation, we introduce more change points during the time series generation. Notably, we treat the union of all existing $A$ as the ground truth for the process (If there is an effect for some limited amount of time, there is a causal effect). As we set the number of time steps $T$ to either 250 or 1000 in our experiments, we introduce $N_{\text{change}} \in \{1, 2, 3, 4, 5\}$ change points, using the time steps denoted in Table 7.

### B.7 ADDITIONAL DETAILS $\mathbf{V}_{\text{LENGTH}}$

To reliably estimate relationships and identify patterns, CD algorithms need a sufficient number of samples. To violate this, often implicit, assumption ($\mathbf{V}_{\text{length}}$), we reduce the number of time

| $N_{\text{change}}$ | Selected Time Steps for change points | |
|---|---|---|
| | $T = 250$ | $T = 1000$ |
| 1 | 125 | 500 |
| 2 | 83,166 | 333,666 |
| 3 | 63,126,187 | 250,500,750 |
| 4 | 50,100,150,200 | 200,400,600,800 |
| 5 | 41,82,122,163,205 | 166,333,500,666,833 |

Table 7: The specific time steps we use for the respective number of change points to violate stationarity ($\mathbf{V}_{\text{stat}}$). We separate the time steps for the two settings $T = 250$ and $T = 1000$ in our experiments.

steps $T$ we sample from Eq. (7). Specifically, we employ the following five discrete levels $T \in \{76, 58, 41, 23, 6\}$.

## B.8 ADDITIONAL DETAILS $\mathbf{V}_{\text{MCAR}}$, $\mathbf{V}_{\text{MAR}}$, $\mathbf{V}_{\text{MNAR}}$, $\mathbf{V}_{\text{EMPTY}}$

| Empty Periods | $T = 250$ | $T = 1000$ |
|---|---|---|
| $2\times$ 0.24 | (40, 100) and (150, 210) | (160, 400) and (600, 840) |
| $2\times$ 0.30 | (31, 106) and (144, 219) | (124, 424) and (576, 876) |
| $2\times$ 0.352 | (23, 111) and (139, 227) | (92, 444) and (556, 908) |
| $2\times$ 0.412 | (14, 117) and (133, 236) | (56, 468) and (532, 944) |
| $2\times$ 0.464 | (6, 122) and (128, 244) | (24, 488) and (512, 976) |

Table 8: The specific time periods, denoted by (start, end), where we set the parent sets to $\varnothing$ during generation of $\boldsymbol{X}$, $\mathbf{V}_{\text{empty}}$. We separate the time steps for the two settings $T = 250$ and $T = 1000$ in our experiments. In all cases, we introduce two periods with no causal signal and scale the length to increase intensity. We denote the average ratio of empty periods for sampled time series in the first column. Note that this ratio, i.e., the violation intensity, increases with each row.

In practical applications, data quality cannot always be controlled, leading to various forms of degradation beyond observational noise. We investigate four quality violations that are common across domains, namely missing measurements ($\mathbf{V}_{\text{mcar}}$, $\mathbf{V}_{\text{mar}}$, $\mathbf{V}_{\text{mnar}}$) and sensor failures $\mathbf{V}_{\text{empty}}$. Concerning missing measurements, we define a binary mask matrix $\mathbf{M} \in \{0, 1\}^{N \times T}$, where $m_{i,t} = 1$ indicates a value is missing, and $m_{i,t} = 0$ indicates it is observed. Further, we set a target missingness rate $\rho$. For $\mathbf{V}_{\text{mcar}}$, each value in the time series has the same probability of missing $m_{i,t} \sim \text{Bernoulli}(\rho)$. For $\mathbf{V}_{\text{mar}}$, we let $\mathbf{M}$ depend on an auxiliary observed, in our case, real-world (Stein et al. (2024a)), time series. Let $\mathbf{Z} \in \mathbb{R}^{N \times T}$ be a set of external, real-world time series sampled from a reference dataset. The missingness probability is determined by $P(m_{i,t} = 1 \mid z_{i,t}) = \sigma(\alpha + \beta \cdot z_{i,t})$, where $\beta$ specifies the sensitivity to $Z$. To ensure the empirical missingness rate matches the target $\rho$, the intercept $\alpha$ is numerically determined. For $\mathbf{V}_{\text{mnar}}$, we let $M$ depend on the time series itself. Here, the probability of a value missing is determined by $P(m_{i,t} = 1 \mid x_{i,t}) = \sigma(\alpha + \beta \cdot x_{i,t})$ where $\alpha$ and $\beta$ fulfill the same roles as described beforehand. In our experiments, we rely on $\beta = 3.5$. To increase the severity of these violations, we stepwise increase the target missingness rate $\rho \in \{0.2, 0.3375, 0.475, 0.6125, 0.75\}$. Regarding sensor failure ($\mathbf{V}_{\text{empty}}$), we periodically set the parent set of variables to $\varnothing$ during the generation of $\boldsymbol{X}$, simulating temporary unnoticed sensor failures (zero-information measurements). Specifically, we introduce two such periods for each sampled time series and list the corresponding intervals in Table 8. Crucially, the average ratio of each of the empty periods during the $T = 250$ or $T = 1000$ time steps increases as follows: $\{0.25, 0.345, 0.4, 0.425, 0.455\}$.

## B.9 ADDITIONAL DETAILS $\mathbf{V}_{\text{SCALE}}$

Related work suggests that synthetically generated data introduces artifacts that are beneficial for identifying causal order, e.g., Ormaniec et al. (2025). This phenomenon is problematic because

it can lead to an overestimation of a CD method's efficacy. Because it can be partly remedied using standardization to mean zero and variance one Reisach et al. (2021); Kaiser & Sipos (2021), we violate this condition by mixing the original observations $\boldsymbol{X}$ with its standardized version $\overline{\boldsymbol{X}}$ after generating the time series. In particular, we alpha-blend both versions of the time series: $\hat{\boldsymbol{X}} = \alpha \overline{\boldsymbol{X}} + (1 - \alpha)\boldsymbol{X}$, where $\alpha$ determines the intensity of $\mathbf{V}_{\text{scale}}$. Specifically, in our investigation, we use $\alpha \in \{0.2, 0.4, 0.6, 0.8, 1\}$.

## C  Appendix — Experimental Setup

### C.1  Sampling Details

|  | (D, L) | |
|---|---|---|
|  | (5,3) | (7,4) |
| $(p_{\text{lag}}, p_{\text{inst}}) = (0.075, 0.0)$ | 5.625 | 14.7 |
| $(p_{\text{lag}}, p_{\text{inst}}) = (0.075, 0.1)$ | 8.125 | 19.6 |
| $(p_{\text{lag}}, p_{\text{inst}}) = (0.15, 0.0)$ | 11.25 | 29.4 |
| $(p_{\text{lag}}, p_{\text{inst}}) = (0.15, 0.1)$ | 13.75 | 34.3 |

Table 9: Expected number of links in sampled SCMs. Here, $D$ denotes the number of variables in $X$ and $L$ denotes the number of lags. The probabilities $(p_{\text{lag}}, p_{\text{inst}})$, correspond to the likelihood of lagged and instantaneous connections in the causal graphs $G$ (i.e., nonzero elements in $A$ and $B$, Eq. (7)).

In this section, we describe the data-generation process we use throughout our experiments and for all violations. Generally, we base all of our experiments on Eq. (7) and alter it according to the violations described in Apx. B. We employ two combinations of the number of variables $D$ and the maximum lags $L$, resulting in a small and a big scenario. Specifically, we set $(D, L) = (5, 3)$ or $(D, L) = (7, 4)$, respectively. Then, with a probability $p_{\text{lag}} \in \{0.075, 0.15\}$ and probability $p_{\text{inst}} \in \{0.0, 0.1\}$ links in $A$ and $B$ are being selected to be nonzero. Finally we generate either 250 or 1000 samples for each time series. This leads to 16 "data regime" combinations, and we list the expected number of links in Table 9. Next, each nonzero element in $A$ and $B$ receives a value that is uniformly sampled from the joint interval $[-0.5, -0.3] \cup [0.3, 0.5]$. Notably, we explicitly exclude coefficients close to 0 to render causal relationships detectable. If $f$ is not the identity function (i.e., for $\mathbf{V}_{\text{nl}}$), a univariate function is drawn from the corresponding distribution. We then generate $X$ iteratively using Eq. (7). To initialize, we sample every variable from $(\mathcal{N}(0, 1))$.

Further, before we start to generate $X$, we need to evaluate the following two conditions: First, concerning instantaneous coefficients $B$, we guarantee the sample graph to be acyclic by checking the following necessary and sufficient condition for acyclicity:

$$\text{tr}(e^B) \overset{!}{=} D, \tag{35}$$

where $tr$ is the trace operator. If this condition is not met, we resample $B$ until it passes. See Zheng et al. (2018) for details of this condition. To account for potential divergence of the SCM, we test the VAR stability of $A$. In particular, we investigate whether it is stationary by evaluating the eigenvalues of its companion matrix $F$:

$$F = \begin{bmatrix} A_{t-1} & A_{t-2} & \dots & A_{t-3} & A_{t-4} \\ I_D & 0 & \dots & 0 & 0 \\ 0 & I_D & \dots & 0 & 0 \\ \vdots & \vdots & \ddots & \vdots & \vdots \\ 0 & 0 & \dots & I_D & 0 \end{bmatrix} \tag{36}$$

where $I_D$ is a $D \times D$ identity matrix. To guarantee the stationarity of the corresponding process, all eigenvalues of $F$ have to lie in the complex unit circle. This condition applies if

$$\max_i |\lambda_i(F)| < 1. \tag{37}$$

If it does not hold for the sampled $A$, we resample the coefficients.

However, if the $f_{i,d,l}$ are nonlinear, then the VAR stability test does not apply. Hence, we additionally check for divergence with the following two tests: First, while generating $X$ we continuously check whether any variable in $X$ is monotonically increasing over the last $\mathcal{T}$ time steps by testing:

$$\exists i \in \{1, \dots, d\}, \exists t \quad \text{s.t.} \quad \forall k \in \{0, \dots, \mathcal{T}\},$$
$$|X_{i,t-k-1}| < |X_{i,t-k}| \tag{38}$$

If this condition is met, we halt the generation process and resample a new SCM. In our experiments, we set $\mathcal{T} = 10$.

Second, we test whether any time series in $X$ exceed a maximum value (likely indicating divergent processes). In our experiments, this value is set to $\pm 25$. Again, if this condition is met, we halt the generation process and resample a new SCM.

For each violation intensity and data regime, we sample 100 random SCMs along with a corresponding $X$. As discussed before, a "data regime" is a combination of $D$, $L$, $p_{lag}$, and $p_{inst}$ (compare Table 9) as well as the length of the time series $T \in \{250, 1000\}$. In summary, this results in $2 \times 2 \times 2 \times 2 \times 100 = 1600$ SCMs per violation intensity level. Considering that we evaluate 5 stepwise violation intensities, we typically sample 8000 SCMs for each violation contained in Table 2. Notably, for violations that are concerned with instantanous links ($\mathbf{V}_{faith,i}$, $\mathbf{V}_{conf,i}$), we do not generate samples with $p_{inst} = 0$, effectively halving the amount of SCMs to 4000. For $\mathbf{V}_{length}$, the time series length becomes the violation property, also halving the data-regime space. In total, the results on the robustness of time series causal discovery in this paper are based on: $30 \times 8000 + 3 \times 4000 = $ **252000 pairs of SCM and time series.**

## C.2 CAUSAL DISCOVERY METHODS DETAILS

| | CrossCorrelation | CausalPretraining | GVAR | Varlingam | PCMCI | PCMCI+ | SVAR-RFCI | F-PCMCI | Dynotears | NTS-NOTears |
|---|---|---|---|---|---|---|---|---|---|---|
| *Temporal Dynamics* | | | | | | | | | | |
| Lagged Effects | ✓ | ✓ | ✓ | ✓ | ✓ | ✓ | ✓ | ✓ | ✓ | ✓ |
| Instantaneous Effects | ✗ | ✗ | ✗ | ✓ | ✗ | ✓ | ✓ | ✓ | ✓ | ✓ |
| *Observational noise* | ✗ | ✗ | ✗ | ✗ | ✗ | ✗ | ✗ | ✗ | ✗ | ✗ |
| *Hidden confounding* | ✗ | ✗ | ✗ | ✗ | ✗ | ✗ | ✗ | ✗ | ✗ | ✗ |
| *Unfaithfulness* | ✗ | † | ✗ | ✗ | ✗ | ✗ | ✗ | ✗ | ✓ | ✓ |
| *Nonlinearity* | ✗ | ✓ | ✗ | ✗ | ✓* | ✓* | ✓* | ✓* | ✗ | ✓ |
| *Innovation noise* | | | | | | | | | | |
| Non-Additive noise | ✗ | ✗ | ✗ | ✗ | ✗ | ✗ | ✗ | ✗ | ✗ | ✗ |
| Gaussian additive noise | ✓ | ✓ | ✓ | ✗ | ✓ | ✓ | ✓ | ✓ | ✓ | ✓ |
| *NonStationarity* | ✗ | ✗ | ✗ | ✗ | ✗ | ✗ | ✗ | ✗ | ✗ | ✗ |

Table 10: Comparison of core assumptions and capabilities of selected causal discovery algorithms. Note, constraint-based approaches can handle both linearity and nonlinearity. We, however, only test these methods with a linear conditional independence test in this study (partial correlation and robust partial correlation). Therefore, we mark their ability to handle nonlinearity with a *. We include a small study on this matter in apx. D.6. We found no information on the faithfulness assumption for CausalPretraining and therefore mark it with †. Finally, we omit data quality issues such as $V_{length}$, $V_q$, and $V_{scale}$ from this table, as they are typically not explicitly mentioned as assumptions.

We include details on the method assumptions of all causal discovery methods involved in Table 10. Note, many of the data quality assumptions that we test, such as $\mathbf{V}_{length}$, $\mathbf{V}_q$, or $\mathbf{V}_{scale}$, are not explicitly assumed by most methods, however they are nonetheless often implicitly modeled in the synthetic data that is used for testing algorithm performance.

## C.3 HYPERPARAMETER SEARCH SPACES

In Table 11, we include a list of the full hyperparameter space that we evaluated for each causal discovery method used throughout this paper. Note, as we do not consider nonlinear conditional independence tests in our experiments due to computational overhead, we include a small study on this matter in apx. D.6. In total, we evaluate 143 hyperparameter configurations. For our main experiment, we therefore perform **252000 × 143 = 36.036000 causal discovery attempts**.

## C.4 CD METHOD SELECTION

As a wide variety of time series causal discovery algorithms exist, many of which are partially integrated into TCD-Arena, we had to narrow the set of included CD algorithms. Our method selection for this study was guided by a threefold rationale:

1. **Established Baselines:** We prioritized the most well-established methods, typically focusing on original method formulations without domain-specific extensions.

| Method (Combos) | Parameters | Values |
|---|---|---|
| Cross Correlation (3) | $L_{\mathrm{model}}$ | $L-2, L, L+2$ |
| CausalPretraining (2) | Architecture | TRF, GRU |
| Varlingam (6) | $L_{\mathrm{model}}$ | $L-2, L, L+2$ |
|  | Prune | True, False |
| GVAR (6) | $L_{\mathrm{model}}$ | $L-2, L, L+2$ |
|  | Use | coeff, p-val |
| PCMCI (6) | $L_{\mathrm{model}}$ | $L-2, L, L+2$ |
|  | CI test | ParC, RParC |
| PCMCI+ (12) | $L_{\mathrm{model}}$ | $L-2, L, L+2$ |
|  | CI test | ParC, RParC |
|  | RLL | True, False |
| SVAR-RFCI (6) | $L_{\mathrm{model}}$ | $L-2, L, L+2$ |
|  | CI test | ParC, RParC |
| F-PCMCI (6) | $L_{\mathrm{model}}$ | $L-2, L, L+2$ |
|  | CI test | ParC, RParC |
| Dynotears (48) | $L_{\mathrm{model}}$ | $L-2, L, L+2$ |
|  | Lambda-w | 0.1, 0.3 |
|  | Lambda-a | 0.1, 0.3 |
|  | Max iter | 100, 40 |
|  | H-tol | 1e-8, 1e-5 |
| NTS-NOTears (48) | $L_{\mathrm{model}}$ | $L-2, L, L+2$ |
|  | h-tol | 1e-60, 1e-10 |
|  | Rho-max | 1e+16, 1e+18 |
|  | Lambda1 | 0.005, 0.001 |
|  | Lambda2 | 0.01, 0.001 |

Table 11: Hyperparameter space and number of combinations in the hyperparameter grid. For NTS-NOTears and Dynotears, we use default parameters (first value) and an alternative value per HP. ParC and RParC denote the Partial Correlation conditional independence test and the Robust Partial Correlation conditional independence test, respectively. RLL denotes the resetting of the lagged links before calculating instantaneous effects. Importantly, for the main robustness results (Fig. 3b we only consider $L_{model} = L$. We refer to the implementations of all methods in **TCD-Arena** for further details.

2. **Implementation Reliability:** We selected algorithms with reliable, publicly available implementations to ensure reproducibility.

3. **Scope of Analysis:** We focused on the fundamental form of causal discovery: analyzing a single time series to produce a single window causal graph.

Consequently, we purposefully excluded specialized algorithms developed for specific data properties or extended settings that fall outside this scope. For instance, methods targeting extended summary graphs (e.g., PCGCE Assaad et al. (2022a)), irregular or gappy data (e.g., Cuts Cheng et al. (2022)), latent context (e.g., J-PCMCI Günther et al. (2023), multiple datasets (e.g., SpaceTime Mameche et al. (2025)), or interventional data (e.g., CANDOIT Castri et al. (2024)) were not included. While an analysis of these specialized methods would be valuable, this study aims to first establish a

comprehensive benchmark for the most common causal discovery setting. Despite this, we are committed to continually expanding the methods available in TCD-Arena to increase the range of experiments that can be conducted. We refer to our **TCD-Arena** for a continually growing list.

## C.5 ENSEMBLE TRAINING

| Category | Hyperparameter | Evaluated Values |
|---|---|---|
| *Common to all architectures* | | |
| General | Batch Size | {32, 128,256, 1024, 2048} |
| | Loss Function | {BCE (1,2.5,3), MSE, Focal} |
| | Weight Decay | {BCE (0.01,0.1), MSE, Focal} |
| AdamW | Learning Rate | {1e-4, 1e-2} |
| **Base Model: Linear** | | |
| **Base Model: MLP** | | |
| Architecture | Hidden Layers | {3528+1764} |
| Regularization | Dropout | {0,0.1} |
| **Base Model: Transformer** | | |
| Regularization | Dropout Rate | {0.0, 0.1} |
| Architecture | Model Dim | {256,512} |
| | Num Layers | {2} |

Table 12: Summary of hyperparameters that were evaluated for CD ensembling.

To train our examples, we generate a separate training dataset that holds SCMs and corresponding $X$ from all violations, respective intensities, and data regimes. These samples are combined into a single joint training dataset that we use to train ensembles with trainable parameters. In the most general sense, all our ensembles take a tensor of the shape $B \times M \times D \times D \times L_{model}$, where M denotes the number of individual CD methods, $D$ the number of variables, $L_{model}$ the model order, and $B$ the batch size. This tensor is then, if necessary, reshaped to match the first layer of the respective network architectures (Ensemble$_{Linear}$, Ensemble$_{MLP}$, and Ensemble$_{Transformer}$). All network architectures return a $B \times D \times D \times L_{model}$ tensor that is directly used as the final predicted graph $G$. Notably, for the Ensemble$_{Mean}$ and Ensemble$_{Pareto}$ we directly recombine elements in the input tensor by either taking the average over the model dimension $M$ or selecting the optimal element. Ensemble$_{Linear}$ is implemented as a single fully-connected layer without an activation function. For Ensemble$_{MLP}$, we use a 2-layer MLP with RELU activation functions and batch norm. For Ensemble$_{Transformer}$, we use a standard Transformer architecture Vaswani et al. (2017), with learnable embeddings and a learnable CLS token, from which we use the embedding vector as the final prediction. Further, we evaluate the hyperparameters specified in Table 12 to select the best model that we report in Fig. 3a:

## C.6 REPRODUCABILITY AND COMPUTATIONAL RESOURCES

To facilitate reproducing our results, we have made all code and required resources available in the TCD-Arena repository. This repository includes seeded functionality to generate the datasets used in this paper, along with hashing functions to verify their integrity. The code for all evaluated Causal Discovery methods and ensembling approaches, as well as the scripts to generate the figures presented, is also included. Experiments were conducted on a 7-node Slurm cluster using 14-24 CPU cores per node, with the exception of ensemble training, which was performed on a single Nvidia RTX 3090 GPU. While most individual experiments are not resource-intensive, reproducing the complete set of approximately 36 million causal discovery attempts will require a multi-day runtime on a comparable setup. However, as many scripts can be executed in parallel on a Slurm cluster, the total runtime may vary depending on the specific hardware and configuration. As the execution of various CD methods requires vastly different computational resources, we provide statistics on the average runtimes per tested hyperparameter configuration and per method in table 13.

| Method | Correct Lag | $\uparrow L$ | $\downarrow L$ |
|---|---|---|---|
| CrossCorrelation | $5.2\pm$ † | $7.7\pm$ † | $2.8\pm$ † |
| CausalPretraining | $24.3_{\pm.5}$ | $24.3_{\pm.5}$ | $24.3_{\pm.5}$ |
| GVAR | $.3_{\pm.1}$ | $.4_{\pm.1}$ | $.3_{\pm.1}$ |
| Varlingam | $9.2_{\pm6.4}$ | $11.5_{\pm9.0}$ | $7.2_{\pm4.3}$ |
| PCMCI | $245.0_{\pm89.3}$ | $444.9_{\pm173.8}$ | $86.2_{\pm27.2}$ |
| PCMCI+ | $357.0_{\pm145.3}$ | $47.7_{\pm222.8}$ | $194.3_{\pm67.7}$ |
| SVAR-RFCI | $671.3_{\pm258.0}$ | $1417.7_{\pm626.2}$ | $221.8_{\pm75.5}$ |
| F-PCMCI | $801.7_{\pm44.2}$ | $1155.3_{\pm25.3}$ | $413.5_{\pm21.3}$ |
| Dynotears | $18.3_{\pm8.6}$ | $22.2_{\pm1.8}$ | $16.5_{\pm7.8}$ |
| Nts-Notears | $441.6_{\pm162.2}$ | $505.3_{\pm217.8}$ | $402.8_{\pm12.6}$ |

Table 13: Average computational efforts (in seconds) per 100 samples and for a single hyperparameter configuration for each of the 10 CD strategies. The standard deviations denote differences between hyperparameter configurations. As Cross Corr has no Method hyperparameters, no standard deviation is provided. Note that we run 8,000 samples per HP combination and violation (See apx. C.1).

## C.7 EXTENSIONS OF THE EXPERIMENTAL PROTOCOL

While the experimental protocol for this study was fixed to ensure consistency, we designed TCD-Arena as an extendable dynamic benchmark Shirali et al. (2022). The framework is highly configurable: users can define novel combinations of violations, data regimes, and generation parameters via a single YAML configuration file or command-line arguments. Furthermore, the codebase features a modular architecture explicitly intended to facilitate community contributions. We encourage researchers to extend the benchmark by integrating additional assumption violations and Causal Discovery (CD) methods.

# D   APPENDIX — ADDITIONAL RESULTS

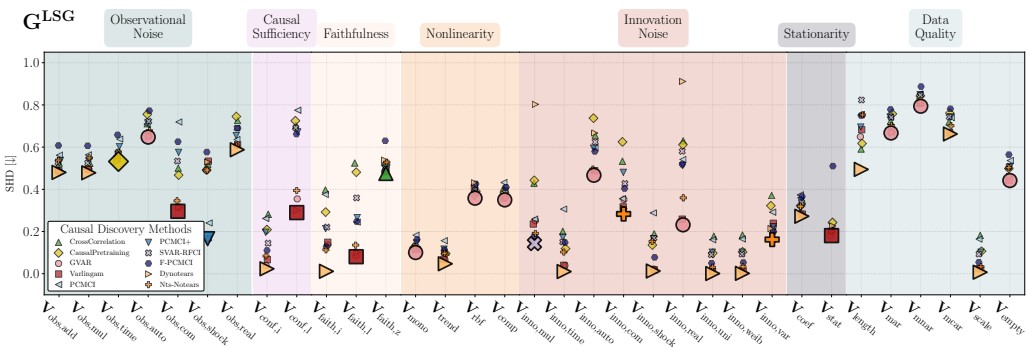

Figure 17: Robustness profile for $G^{\text{LSG}}$ and of ten Causal Discovery algorithms against a multitude of stepwise assumption violations measured as average normalized SHD over various data regimes.

## D.1   ADDITIONAL METRICS

To extend our empirical evaluation, we include alternative metrics and additional depictions. First, we include additional depictions of the main metric (normalized minimum SHD) in Fig. 18 and Fig. 19. Second, we include the same depiction for three alternative metrics (AUROC, Maximum F1, and Maximum Accuracy) in Fig. 20- Fig. 25. Generally, we find that most metrics show a similar picture (e.g., the ordering of methods when looking at performance metrics for $G^{\text{INST}}$, or the generally low performance of Cross Correlation and CausalPretraining). A notable exception to this is the difference between the normalized SHD and other metrics for $G^{INST}$. Since the normalized SHD cannot become smaller than 1 for undirected graphs (PCMCI+ and SVARRFCI) by definition, other metrics provide a more informative picture. Third, we include the same depictions of normalized minimum SHD when selecting the optimal hyperparameters per method for each violation individually (contrary to our protocol; see Fig. 4) in Fig. 26-Fig. 27. Notably, differences between these protocols are typically quite small, suggesting that a general, well-performing hyperparameter configuration exists. Finally, we also report the worst-case performance per violation for the selected hyperparameter configuration in the main results in Fig. 28- Fig. 29. Again, we find that method ordering is generally consistent with the results that were presented in the main section of the paper.

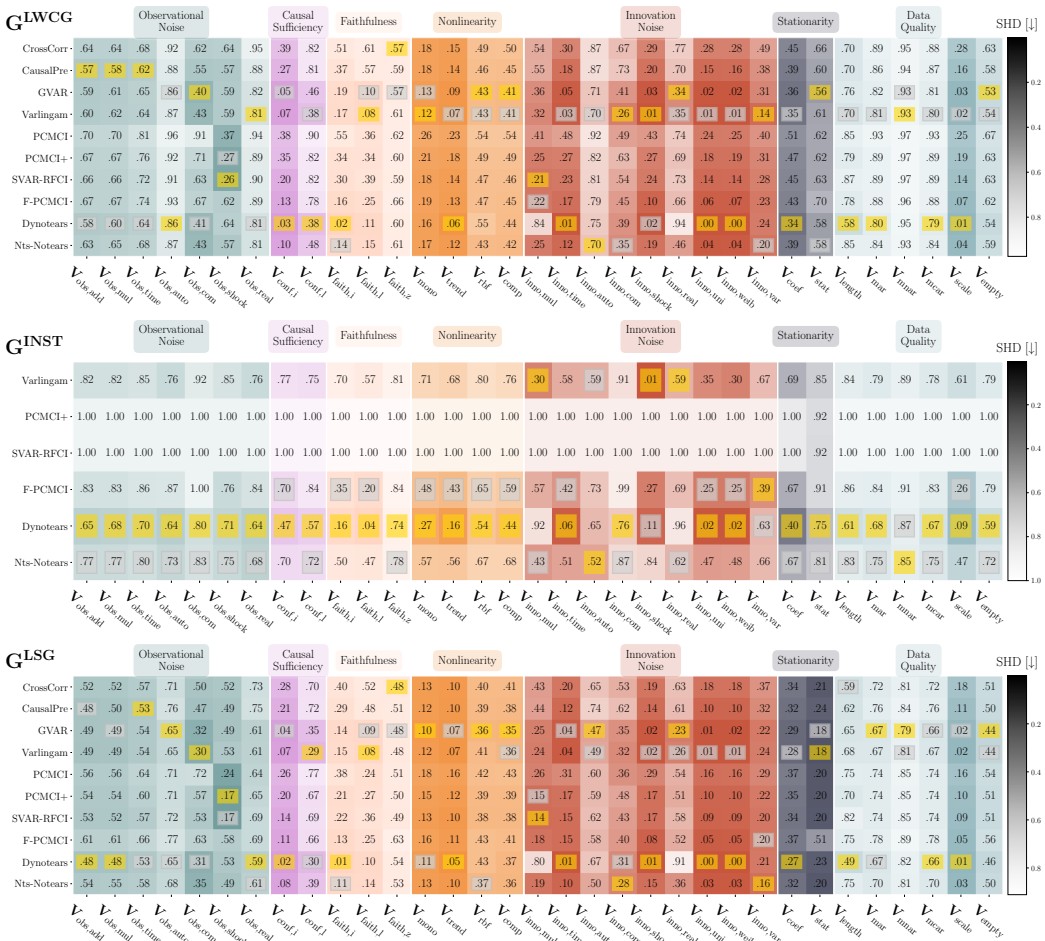

Figure 18: Alternative depiction 1 of robustness profiles of ten Causal Discovery algorithms against a multitude of stepwise assumption violations measured as **average SHD** over various data regimes. From top to bottom: results for $G^{\text{LWCG}}$, $G^{\text{INST}}$ and $G^{\text{LSG}}$.

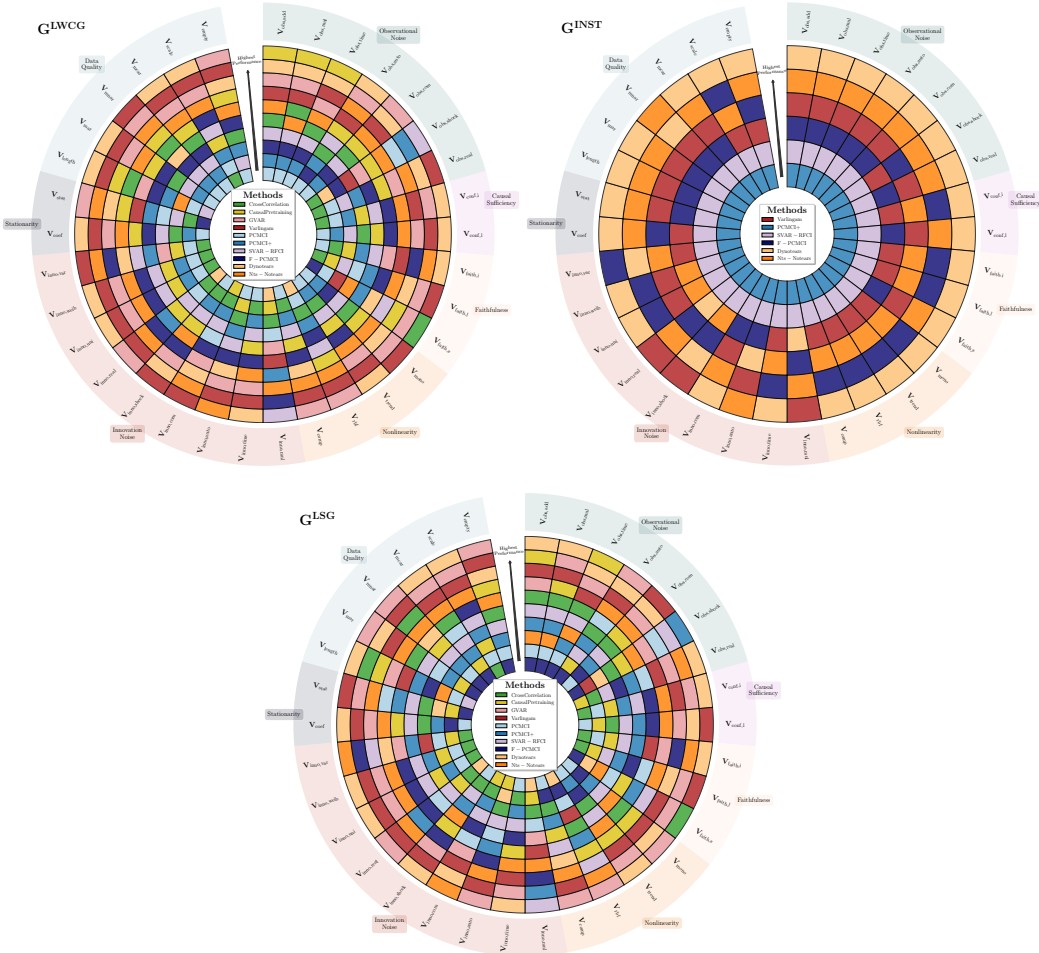

Figure 19: Alternative depiction 2 of robustness profiles of ten Causal Discovery algorithms against a multitude of stepwise assumption violations measured as **average SHD** over various data regimes. From top to bottom: results for $G^{\text{LWCG}}$, $G^{\text{INST}}$ and $G^{\text{LSG}}$.

### D.1.1 Alternative Metric - AUROC

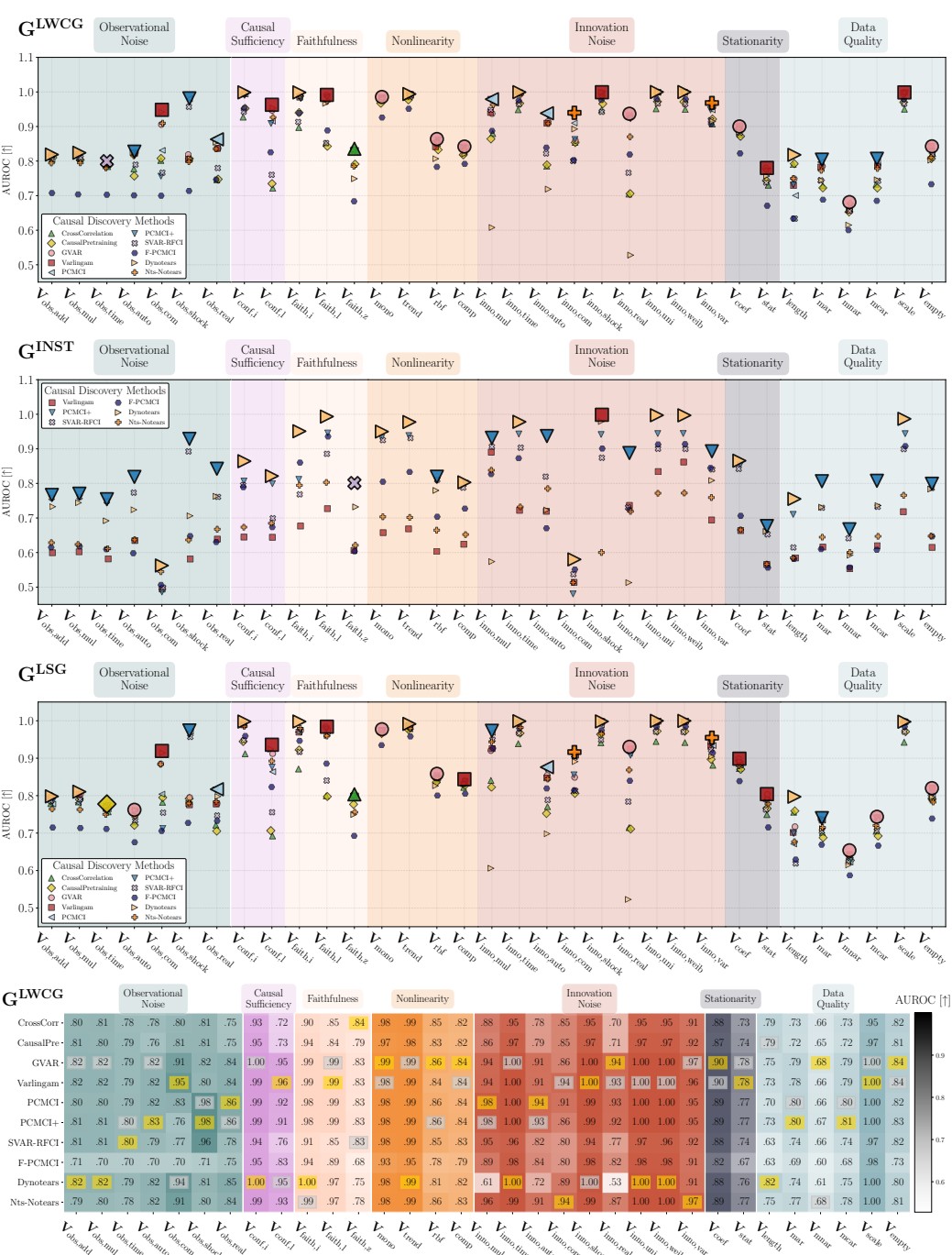

Figure 20: Robustness profiles of ten Causal Discovery algorithms against a multitude of stepwise assumption violations measured as **average AUROC** over various data regimes. From top to bottom: results for $G^{\text{LWCG}}, G^{\text{INST}}$ and $G^{\text{LSG}}$. Notably, we include three depictions for each graph to improve data comprehensibility. First, **a scatter plot (1-3)**, second, **a heatmap (4-6)**, and third, **a Vinylplot (7-9, see next page)** that specifies the ranking of the methods .

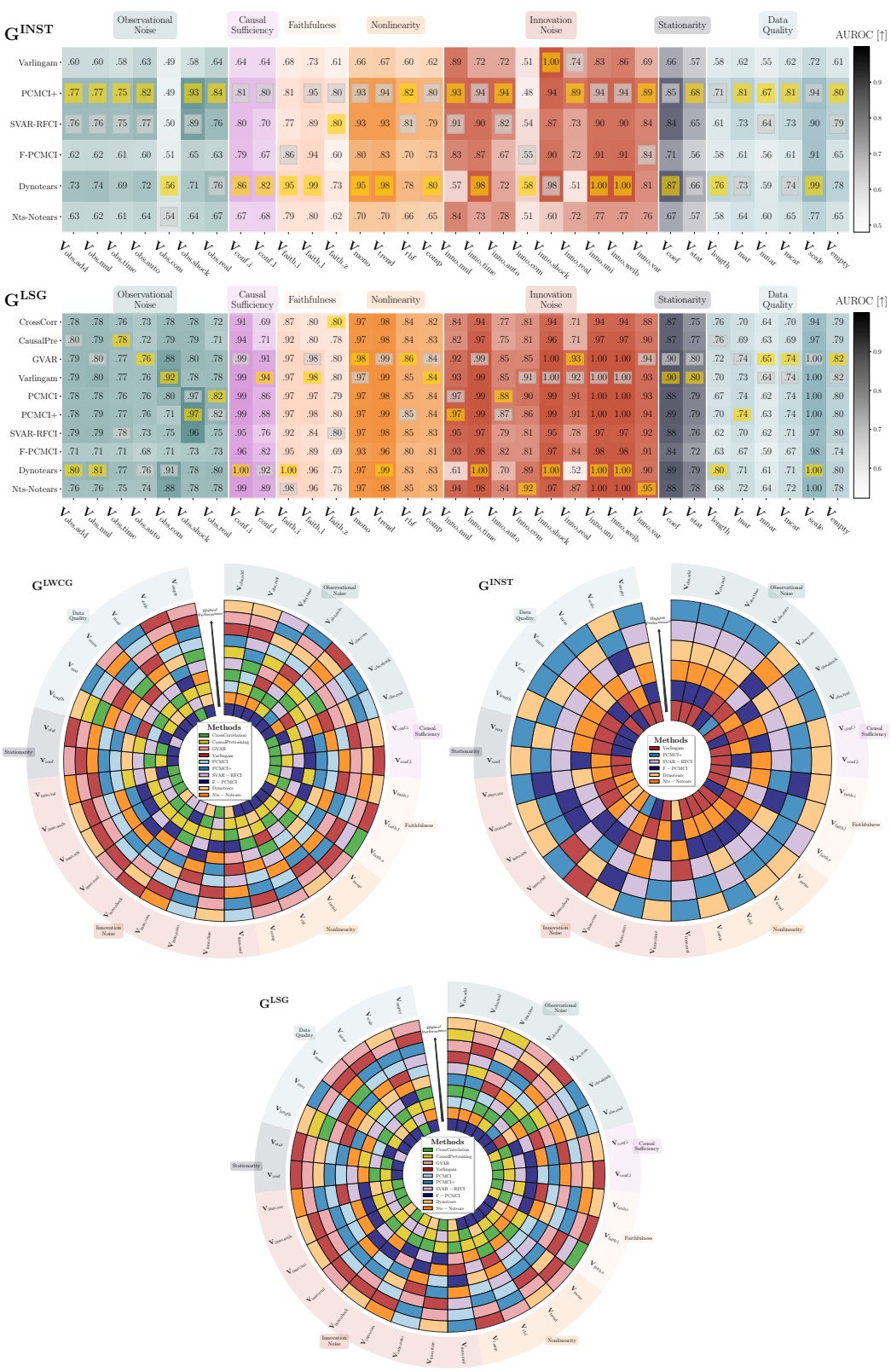

Figure 21: Additional depictions of the robustness profiles of ten Causal Discovery algorithms against assumption violations measured as **average AUROC**. Top: heatmaps for $G^{\text{INST}}, G^{\text{LSG}}$. Bottom: Method performance rankings. **Best viewed in conjunction with the figures on the previous page.**

## D.1.2 ALTERNATIVE METRIC - F1 MAX

Figure 22: Robustness profiles of ten Causal Discovery algorithms against a multitude of stepwise assumption violations measured as **average maximum F1** over various data regimes. From top to bottom: results for $G^{\text{LWCG}}, G^{\text{INST}}$ and $G^{\text{LSG}}$. Notably, we include three depictions for each graph to improve data comprehensibility. First, **a scatter plot (1-3)**, second, **a heatmap (4-6)**, and third, **a Vinylplot (7-9, see next page)** that specifies the ranking of the methods .

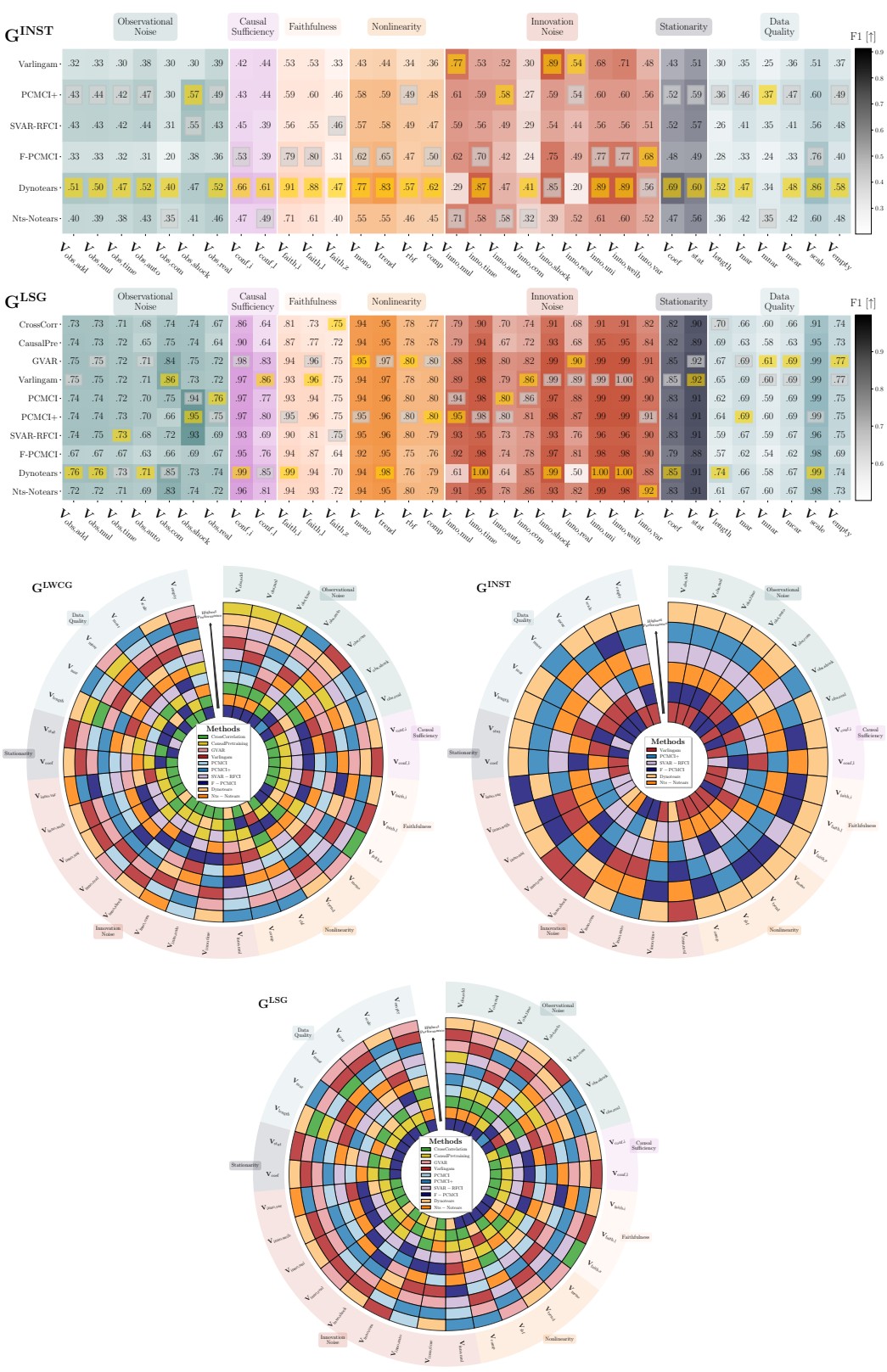

Figure 23: Additional depictions of the robustness profiles of ten Causal Discovery algorithms against assumption violations measured as **average maximum F1**. Top: heatmaps for $G^{\text{INST}}, G^{\text{LSG}}$. Bottom: Method performance rankings. **Best viewed in conjunction with the figures on the previous page.**

### D.1.3 ALTERNATIVE METRIC - MAXIMUM ACCURACY

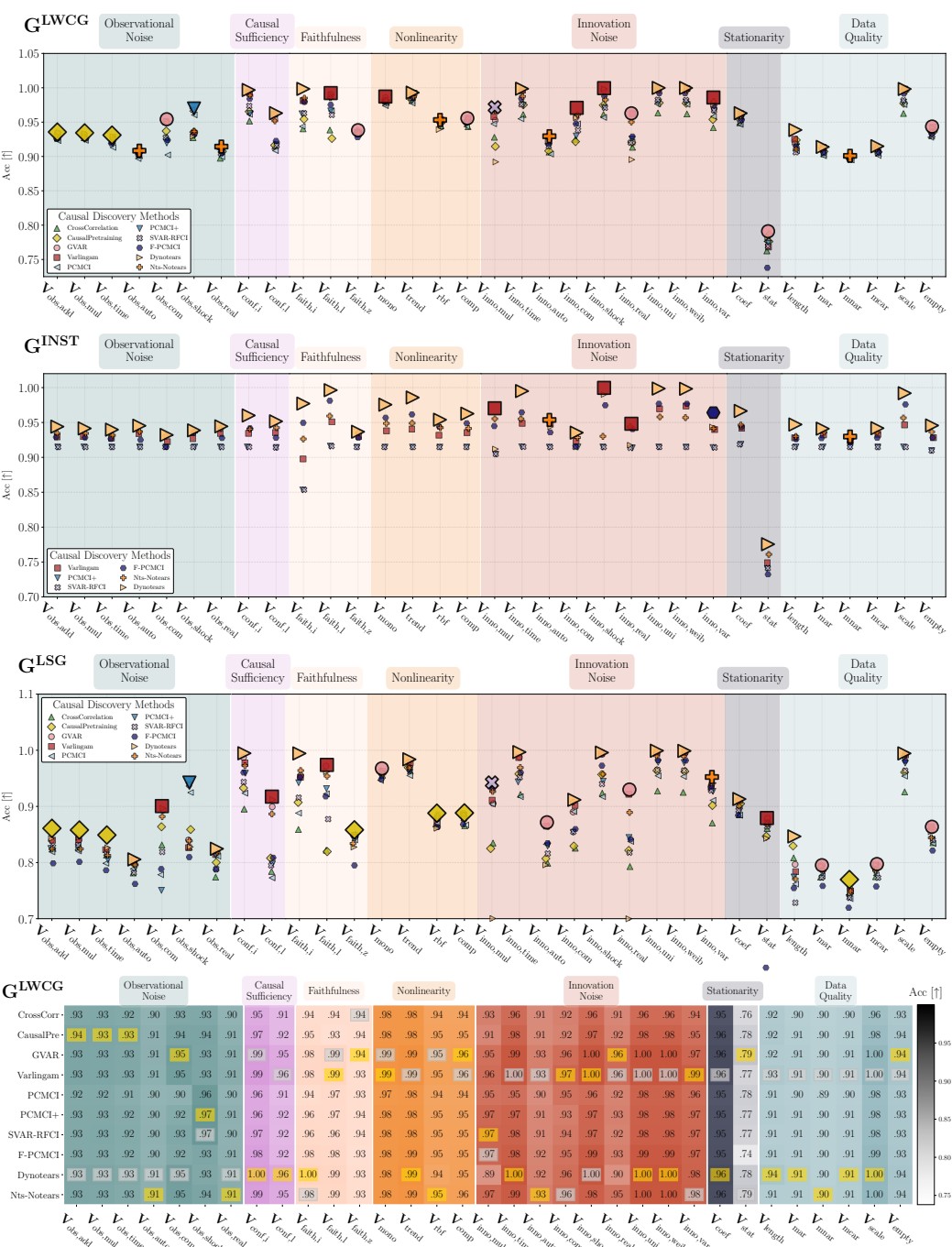

Figure 24: Robustness profiles of ten Causal Discovery algorithms against a multitude of stepwise assumption violations measured as **average maximum accuracy** over various data regimes. From top to bottom: results for $G^{\text{LWCG}}, G^{\text{INST}}$ and $G^{\text{LSG}}$. Notably, we include three depictions for each graph to improve data comprehensibility. First, **a scatter plot (1-3)**, second, **a heatmap (4-6)**, and third, **a Vinylplot (7-9, see next page)** that specifies the ranking of the methods .

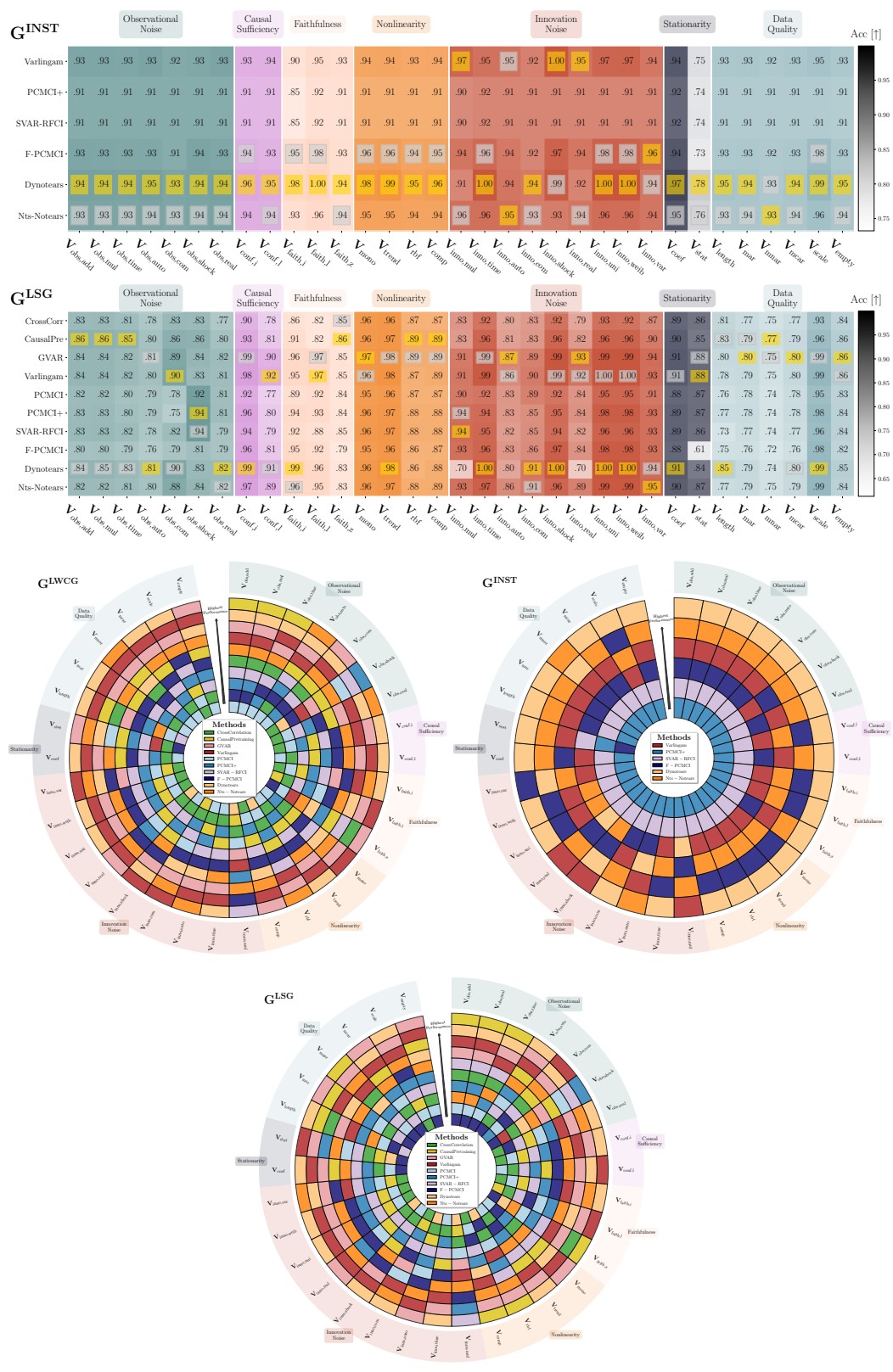

Figure 25: Additional depictions of the robustness profiles of ten Causal Discovery algorithms against assumption violations measured as **average maximum accuracy**. Top: heatmaps for $G^{\text{INST}}, G^{\text{LSG}}$. Bottom: Method performance rankings. **Best viewed in conjunction with the previous page.**

### D.1.4 Alternative Metric - Individual hyperparameter selection per violation

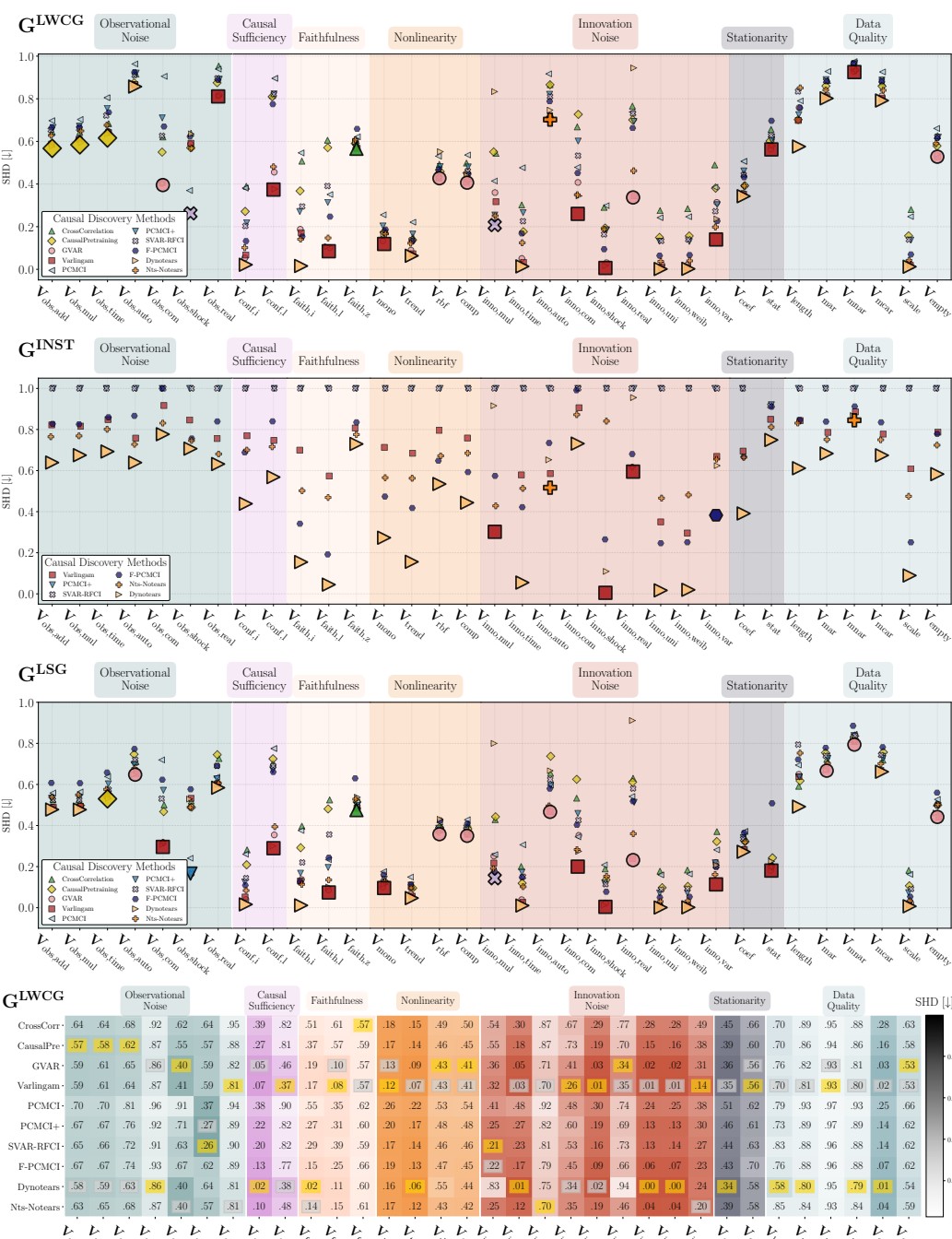

Figure 26: Robustness profiles of ten Causal Discovery algorithms against a multitude of stepwise assumption violations measured as **normalized SHD and per-violation hyperparameter selection over** various data regimes. From top to bottom: results for $G^{\text{LWCG}}, G^{\text{INST}}$ and $G^{\text{LSG}}$. Notably, we include three depictions for each graph to improve data comprehensibility. First, **a scatter plot (1-3)**, second, **a heatmap (4-6)**, and third, **a Vinylplot (7-9, see next page)** that specifies the ranking of the methods .

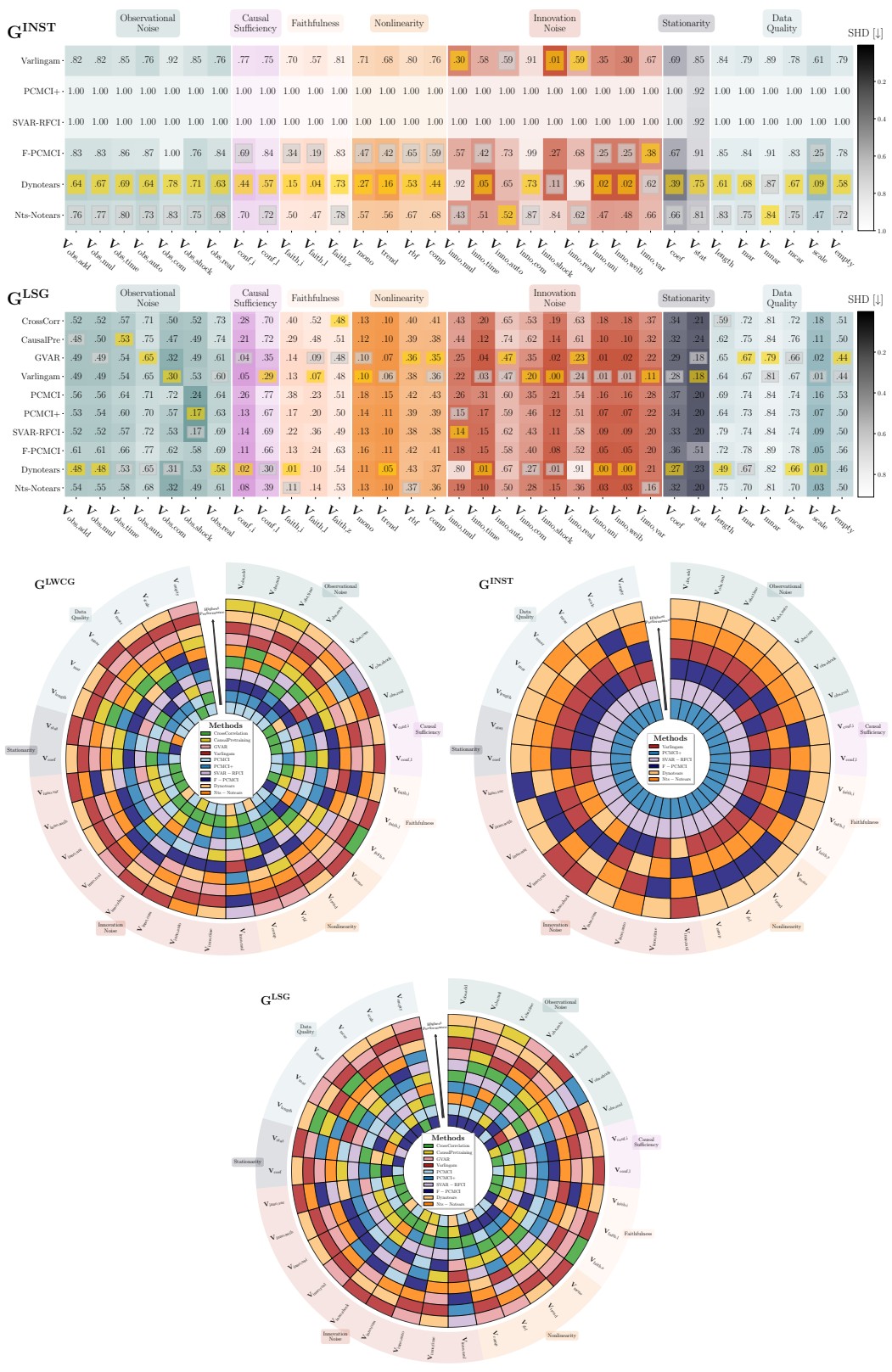

Figure 27: Depictions of robustness profiles of ten CD algorithms against assumption violations measured as **normalized SHD and per-violation hyperparameter selection**. Top: heatmaps for $G^{\text{INST}}, G^{\text{LSG}}$. Bottom: Method rankings. **Best viewed in conjunction with the previous page.**

### D.1.5 ALTERNATIVE METRIC - WORST CASE PERFORMANCE PER VIOLATION

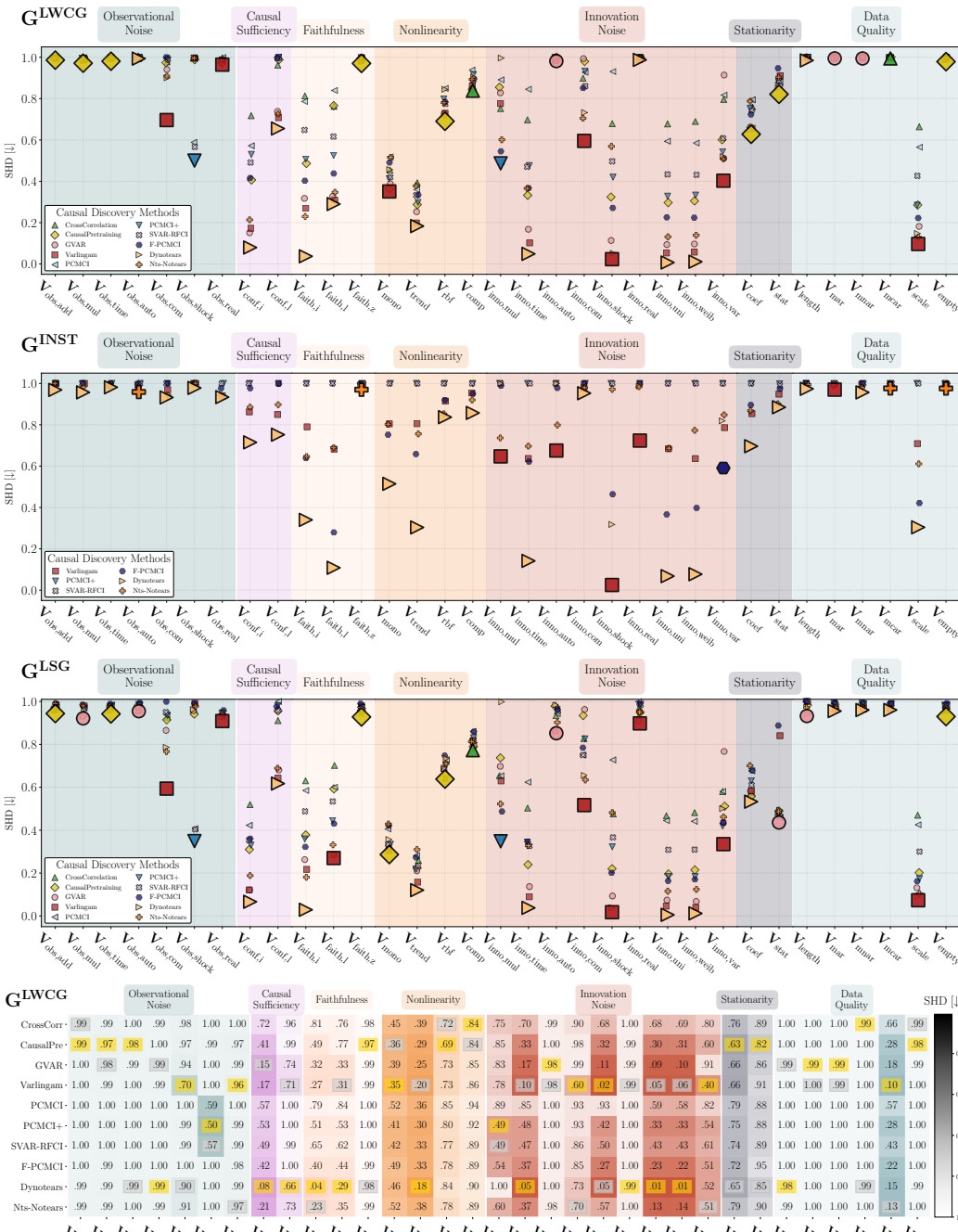

Figure 28: Robustness profiles of ten Causal Discovery algorithms against a multitude of stepwise assumption violations measured as **the highest normalized SHD that was achieved for any data regime and violation level**. From top to bottom: results for $G^{\text{LWCG}}$, $G^{\text{INST}}$ and $G^{\text{LSG}}$. Notably, we include three depictions for each graph to improve data comprehensibility. First, **a scatter plot (1-3)**, second, **a heatmap (4-6)**, and third, **a Vinylplot (7-9, see next page)** that specifies the ranking of the methods .

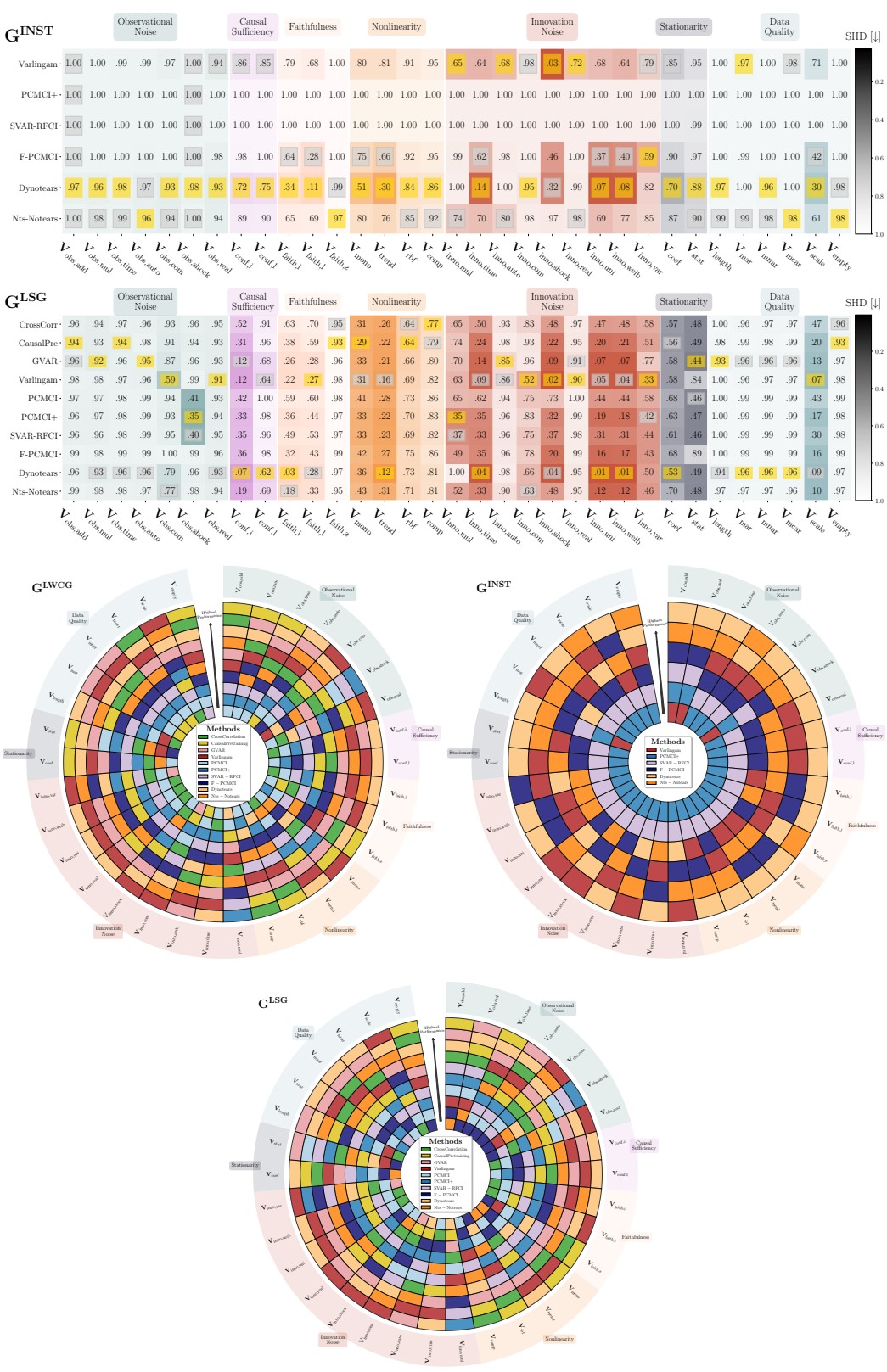

Figure 29: Depictions of robustness profiles of ten CD algorithms against assumption violations measured as **the highest normalized SHD for any data regime and violation level**. Top: heatmaps for $G^{\text{INST}}$, $G^{\text{LSG}}$. Bottom: Method rankings. **Best viewed in conjunction with the previous page.**

## D.2 VISUALIZATIONS OF MISSPECIFIED MODELS

In Table 1 we report robustness profiles for misspecified modelling parameters, i.e., $L_{model} \neq L$. To further support these results, we include more fine-grained depictions of this experiment in Fig. 30 - Fig. 33. Notably, because Causal Pretraining does not require specifying a max lag, its performance is superior in the low-lag regime. Further, concerning $G^{INST}$, Dynotears seems particularly robust.

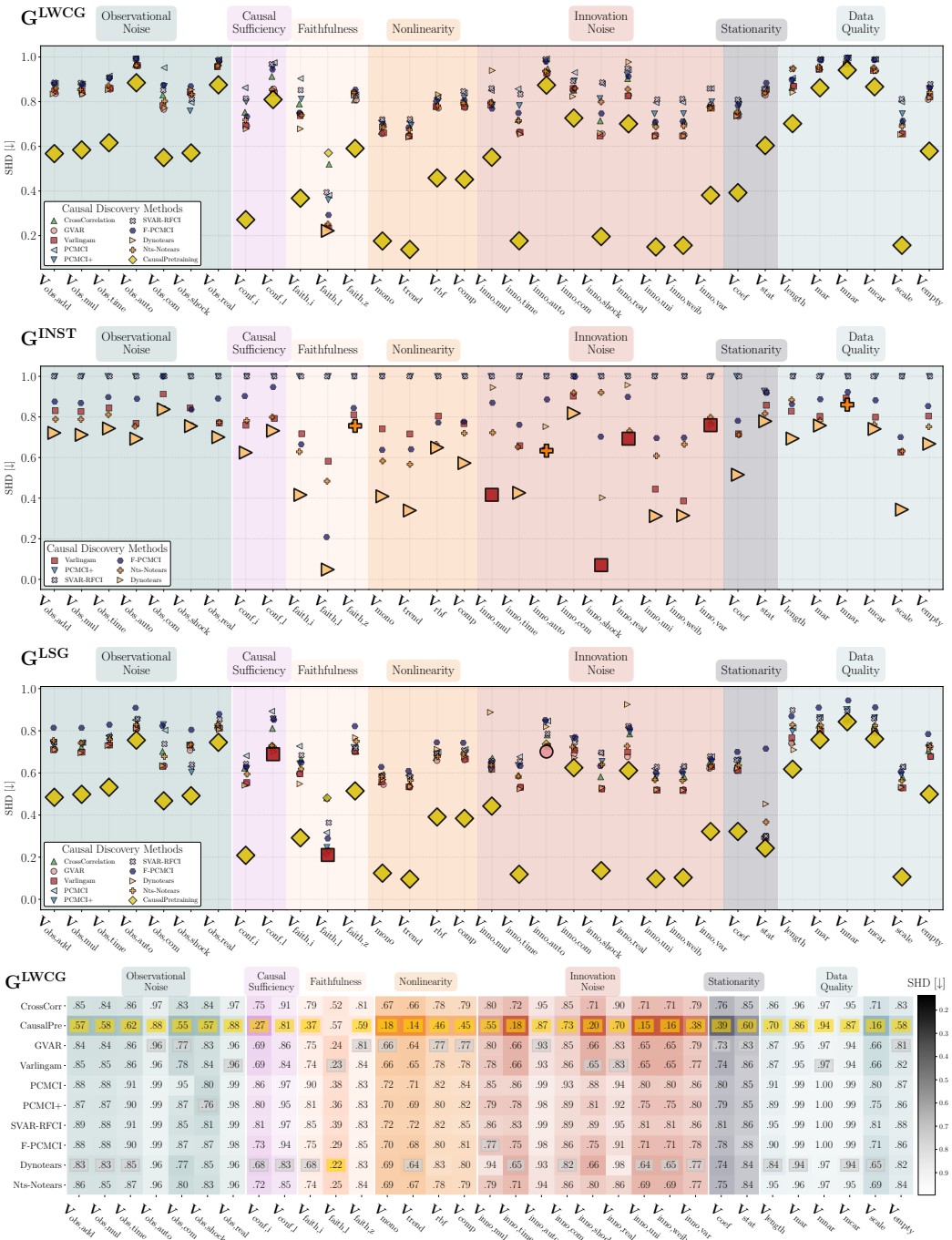

Figure 30: Robustness profiles of ten Causal Discovery algorithms against a multitude of stepwise assumption violations measured as **normalized SHD and under ↓ L**. From top to bottom: results for $G^{\text{LWCG}}$, $G^{\text{INST}}$ and $G^{\text{LSG}}$. Notably, we include three depictions (also see next page) for each graph to improve data comprehensibility.

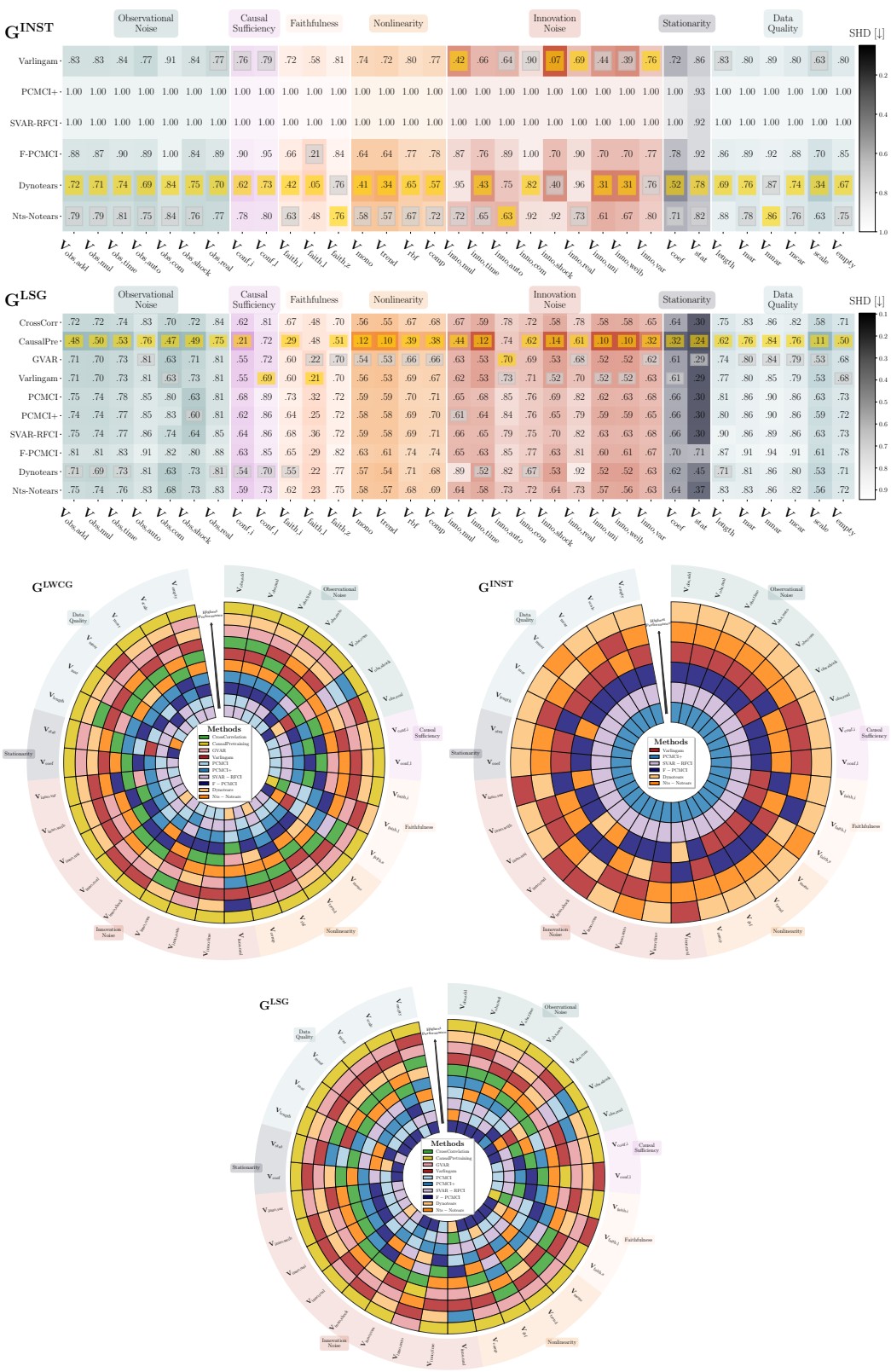

Figure 31: Depictions of robustness profiles of ten CD algorithms against assumption violations measured as **normalized SHD and under ↓ L**. Top: heatmaps for $G^{\text{INST}}, G^{\text{LSG}}$. Bottom: Method rankings.**Best viewed in conjunction with the previous page.**

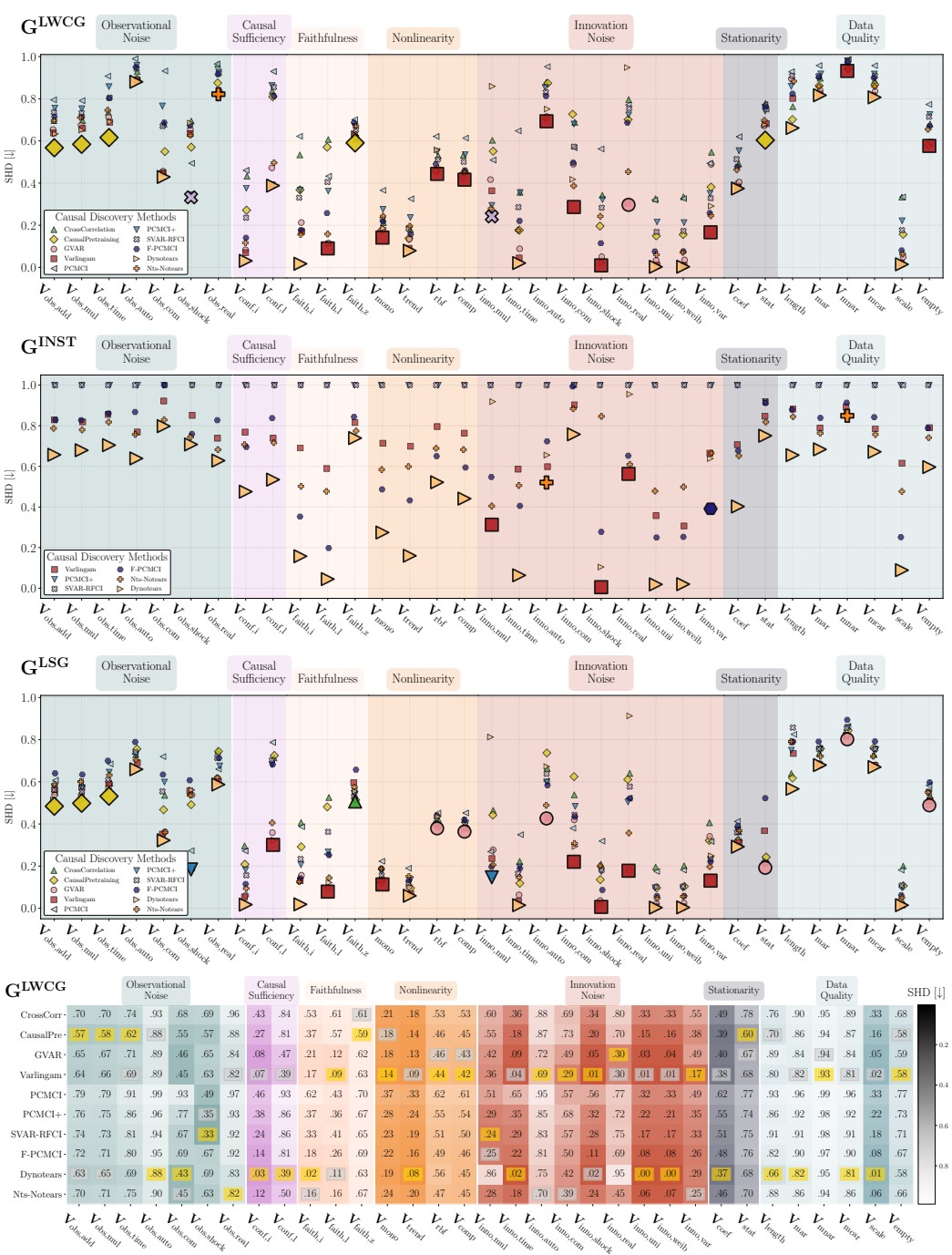

Figure 32: Robustness profiles of ten Causal Discovery algorithms against a multitude of stepwise assumption violations measured as **normalized SHD and under ↑ L**. From top to bottom: results for $G^{LWCG}, G^{INST}$ and $G^{LSG}$. Notably, we include three depictions (also see next page) for each graph to improve data comprehensibility.

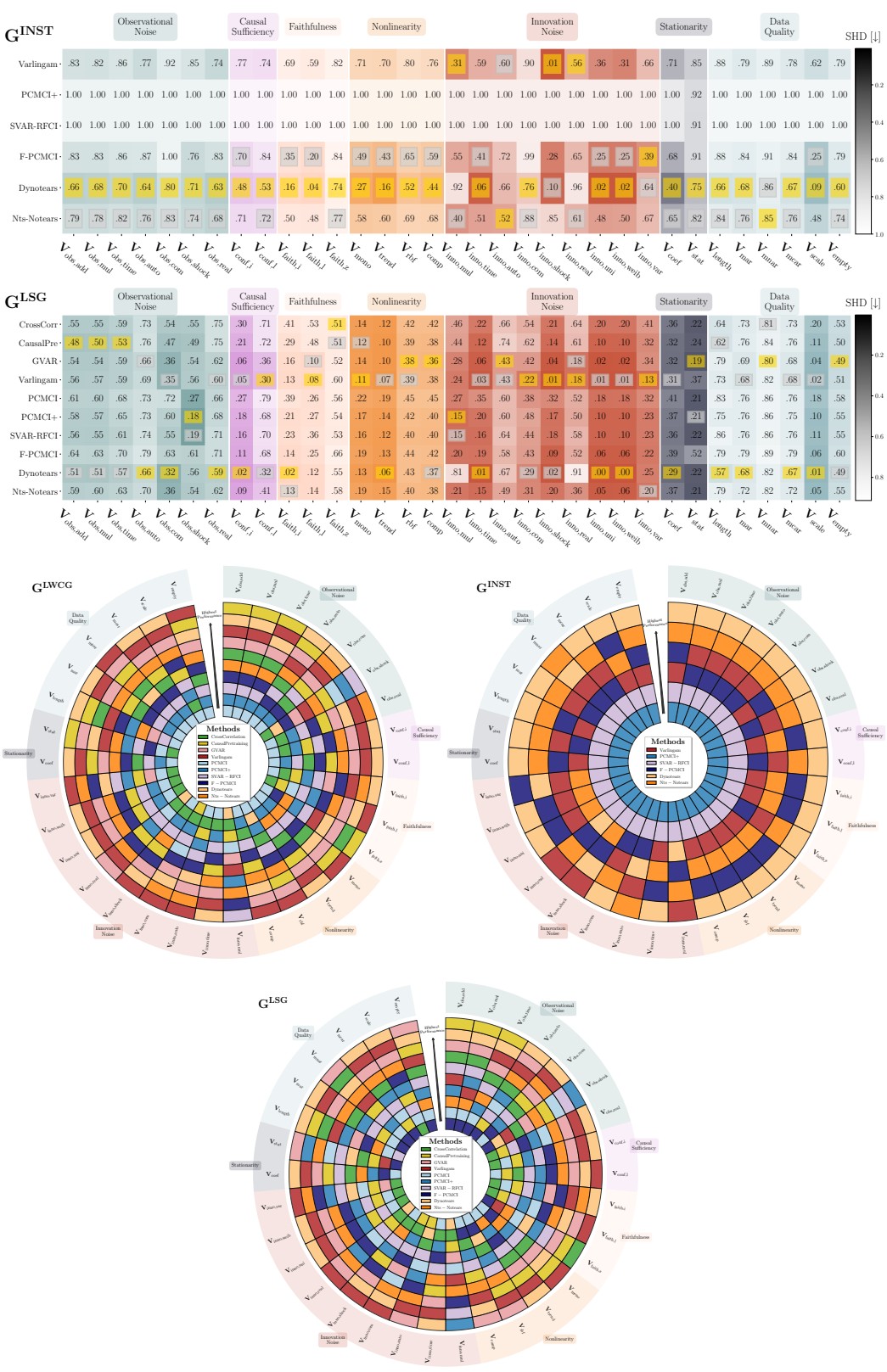

Figure 33: Depictions of robustness profiles of ten CD algorithms against assumption violations measured as **normalized SHD and under ↑ L**. Top: heatmaps for $G^{\text{INST}}, G^{\text{LSG}}$. Bottom: Method rankings.**Best viewed in conjunction with the previous page.**

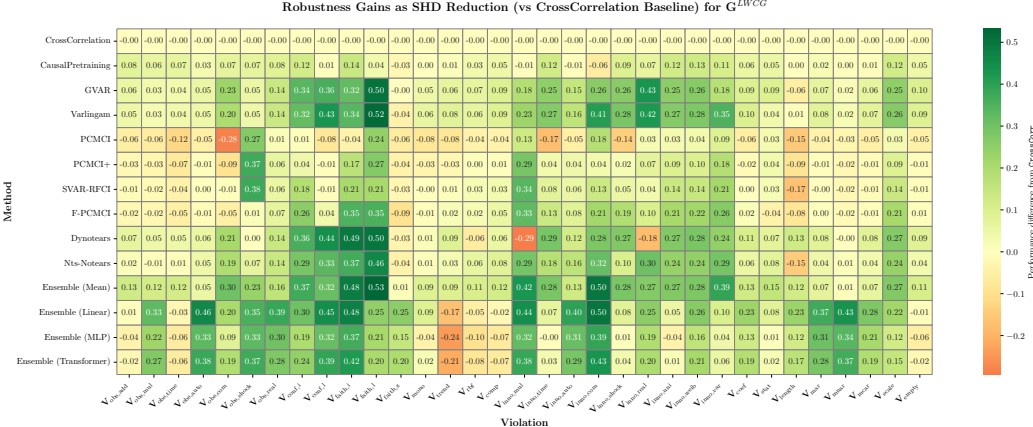

Figure 34: Robustness improvements in comparison to the performance of Cross Correlation per violation and for $G^{\text{LWCG}}$. Interestingly, the highest robustness gains of ensembles are achieved on missingness violations.

### D.3 HYPERPARAMETER SENSITIVITY

To provide a comprehensive view of the hyperparameter sensitivity of all CD methods, we include graphics that illustrate the relationship between violation severity and performance for all combinations of hyperparameters (excluding $L_{model} \neq L$)and data regimes per method. Fig. 37-Fig. 39 contain the sensitivities for lagged effects, while Fig. 40 and Fig. 41 depict methods estimating instantaneous links. Notably, NTS-Notears's performance is often highly hyperparameter-dependent, and a poor parameter selection can result in almost arbitrary performance. Furthermore, we observe that some algorithms occasionally exhibit unintuitive performance curves, where an increase in violation does not necessarily correspond to a decrease in performance (e.g., GVAR for $V_{\text{inno,com}}$). This is, however, typically only the case if the initial performance is already poor. As we only report the best-performing hyperparameter configuration per method in our main results, this has no influence on this evaluation

### D.4 ROBUSTNESS GAINS OF ENSEMBLES

To gain deeper insights into the efficacy of our ensemble strategies, Fig. 34-Fig. 36 illustrate their robustness gains relative to the Cross Correlation baseline and other CD methods across different violations. The most significant improvements occur under the missing data violations $\mathbf{V}_{\text{mcar}}$, $\mathbf{V}_{\text{mar}}$, and $\mathbf{V}_{\text{mnar}}$. Furthermore, we investigated that the ensemble yields the greatest robustness gains in **small** data regimes. For instance, on small datasets, Ensemble$_{\text{Linear}}$ achieves a score of 0.346, compared to the best individual method (Varlingam) with a score of 0.396. For big, their scores are 0.411 and 0.421, respectively. Finally, this strong performance in low-variable environments, combined with the ability to handle missing data better than individual CD methods, provides a plausible explanation for the substantial gains observed on the CausalRivers dataset Apx. D.5, which similarly consists of 5-variable samples.

### D.5 ENSEMBLE PERFORMANCE ON CAUSALRIVERS

As we train and evaluate our ensemble approaches exclusively on synthetic and semi-synthetic data in the main section of the paper, we were interested in how reliant ensembles are on real-world, out-of-domain data distributions. To test this, we evaluate all CD methods and all ensembles on CausalRivers Stein et al. (2024a). Here, we take the first 500 samples from the setting *Random-5*, as this matches the number of variables we use during training. Since CausalRivers provides labels only for $G^{\text{LSG}}$, we evaluate only the corresponding predictions. We find that all ensembles notably improve performance compared to individual CD methods. From this, we conclude that ensembles could be reliable in real-world applications despite being trained on synthetic data.

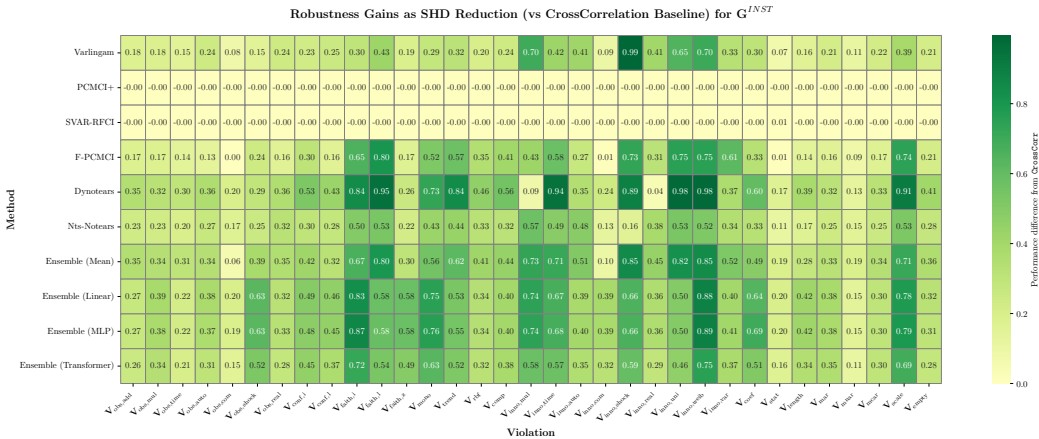

Figure 35: Robustness improvements in comparison to the performance of Cross Correlation per violation and for $G^{\text{INST}}$

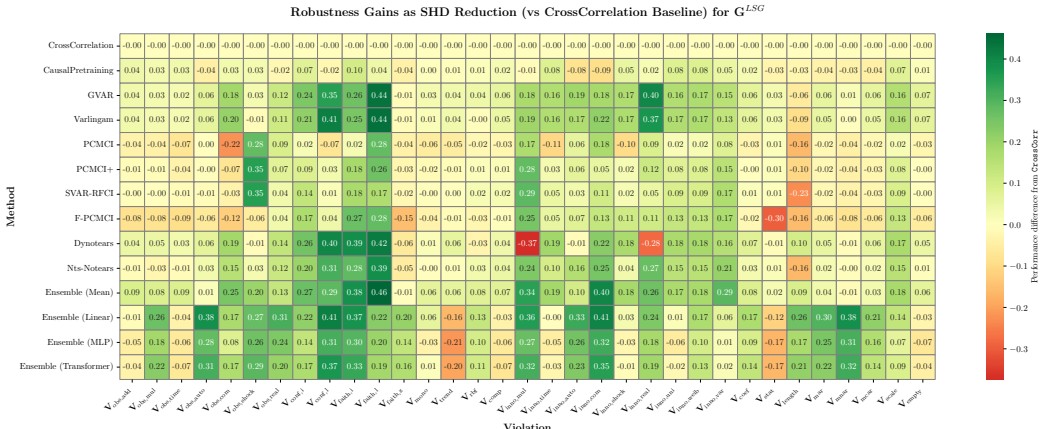

Figure 36: Robustness improvements in comparison to the performance of Cross Correlation per violation and for $G^{\text{LSG}}$. Interestingly, the highest robustness gains of ensembles are achieved on missingness violations.

| Method | CausalRivers (SHD) |
|---|---|
| CausalPretraining | 0.851 |
| PCMCI | 0.978 |
| Nts-Notears | 0.758 |
| PCMCI+ | 0.835 |
| Varlingam | 0.774 |
| Dynotears | 0.715 |
| F-PCMCI | 0.840 |
| SVAR-RFCI | 0.857 |
| GVAR | 0.761 |
| CrossCorrelation | 0.798 |
| Ensemble (Mean) | 0.659 |
| Ensemble (Linear) | 0.666 |
| Ensemble (MLP) | 0.685 |
| Ensemble (Transformer) | 0.666 |

Table 14: Performance assessment for individual CD methods and ensembles on CausalRivers (first 500 samples from *Random-5*)Stein et al. (2024a). We find that ensembles improve performance across the board, suggesting that ensembles can be reliable in real-world applications despite being trained on synthetic data.

### D.6 MULTI-VIOLATION ROBUSTNESS

While we always introduce only a single violation in our main section, we want to emphasize that TCD-Arena also allows an arbitrary combination of violations. To showcase this, we report results for two violation sets in which we simultaneously violate two assumptions. We test a double common violation where we generate data by combining $\mathbf{V}_{\text{inno,com}}$ and $\mathbf{V}_{\text{obs,com}}$, and a violation where we introduce both types of confounding $\mathbf{V}_{\text{conf,i}}$ and $\mathbf{V}_{\text{conf,l}}$. We use the same configuration for the violation levels and the experimental protocol as for the single violations. Results for these multi-violation scenarios are presented in Table 15 through Table 17, alongside the corresponding single-violation baselines. Generally, the method exhibiting the highest robustness in a multi-violation setting is also the top performer for at least one of the constituent single violations. However, we observe an exception with $G^{\text{LSG}}$ under the combined innovation and observation violation, where Nts-Notears emerges as the most robust approach. Given that we evaluated only two specific combinations, we refrain from over-interpreting these results. Nevertheless, as real-world data often violates multiple assumptions simultaneously, we aim to extend TCD-Arena to include more complex scenarios in the future.

| $G^{\text{LWCG}}$ (SHD) | $\mathbf{V}_{\text{inno,com}}$ | $\mathbf{V}_{\text{obs,com}}$ | $\mathbf{V}_{\text{double,com}}$ | $\mathbf{V}_{\text{conf,l}}$ | $\mathbf{V}_{\text{conf,i}}$ | $\mathbf{V}_{\text{double,conf}}$ |
|---|---|---|---|---|---|---|
| CrossCorrelation | 0.668 | 0.621 | 0.958 | 0.815 | 0.390 | 0.819 |
| CausalPretraining | 0.727 | 0.550 | 0.960 | 0.810 | 0.272 | 0.813 |
| GVAR | 0.407 | 0.395 | 0.985 | 0.456 | 0.053 | 0.530 |
| Varlingam | 0.260 | 0.425 | 0.987 | 0.384 | 0.067 | 0.469 |
| PCMCI | 0.488 | 0.905 | 0.989 | 0.896 | 0.381 | 0.904 |
| PCMCI+ | 0.633 | 0.714 | 0.973 | 0.824 | 0.346 | 0.839 |
| SVAR-RFCI | 0.536 | 0.630 | 0.971 | 0.823 | 0.205 | 0.829 |
| F-PCMCI | 0.454 | 0.672 | 0.994 | 0.775 | 0.133 | 0.788 |
| Dynotears | 0.391 | 0.413 | 0.962 | 0.376 | 0.029 | 0.436 |
| Nts-Notears | 0.350 | 0.430 | 0.931 | 0.481 | 0.102 | 0.519 |

Table 15: Robustness evaluation for multi-violation datasets ($\mathbf{V}_{\text{double,com}}$ and $\mathbf{V}_{\text{double,conf}}$) measured as **normalized SHD** and for the recovery of $G^{\text{LWCG}}$. For comparison, we also report the corresponding individual violation results. Combining violations further reduces the robustness of CD methods across the board.

| $G^{\text{INST}}$ (SHD) | $\mathbf{V}_{\text{inno,com}}$ | $\mathbf{V}_{\text{obs,com}}$ | $\mathbf{V}_{\text{double,com}}$ | $\mathbf{V}_{\text{conf,l}}$ | $\mathbf{V}_{\text{conf,i}}$ | $\mathbf{V}_{\text{conf,double}}$ |
|---|---|---|---|---|---|---|
| Varlingam | 0.906 | 0.917 | 0.952 | 0.747 | 0.770 | 0.814 |
| PCMCI+ | 1.000 | 1.000 | 1.000 | 1.000 | 1.000 | 1.000 |
| SVAR-RFCI | 1.000 | 1.000 | 1.000 | 1.000 | 1.000 | 1.000 |
| F-PCMCI | 0.992 | 0.999 | 1.000 | 0.843 | 0.695 | 0.885 |
| Dynotears | 0.757 | 0.802 | 0.940 | 0.568 | 0.474 | 0.677 |
| Nts-Notears | 0.872 | 0.832 | 0.959 | 0.716 | 0.701 | 0.773 |

Table 16: Robustness evaluation for multi-violation datasets ($\mathbf{V}_{\text{double,com}}$ and $\mathbf{V}_{\text{double,conf}}$) measured as **normalized SHD** and for the recovery of $G^{\text{INST}}$. For comparison, we also report the corresponding individual violation results. Combining violations further reduces the robustness of CD methods across the board.

| $G^{\text{LSG}}$ (SHD) | $\mathbf{V}_{\text{inno,com}}$ | $\mathbf{V}_{\text{obs,com}}$ | $\mathbf{V}_{\text{double,com}}$ | $\mathbf{V}_{\text{conf,l}}$ | $\mathbf{V}_{\text{conf,i}}$ | $\mathbf{V}_{\text{conf,double}}$ |
|---|---|---|---|---|---|---|
| CrossCorrelation | 0.533 | 0.500 | 0.853 | 0.703 | 0.281 | 0.708 |
| CausalPretraining | 0.625 | 0.468 | 0.880 | 0.725 | 0.209 | 0.728 |
| GVAR | 0.352 | 0.315 | 0.900 | 0.355 | 0.040 | 0.417 |
| Varlingam | 0.317 | 0.296 | 0.877 | 0.290 | 0.067 | 0.370 |
| PCMCI | 0.356 | 0.719 | 0.891 | 0.775 | 0.262 | 0.796 |
| PCMCI+ | 0.480 | 0.575 | 0.865 | 0.672 | 0.197 | 0.691 |
| SVAR-RFCI | 0.427 | 0.533 | 0.875 | 0.692 | 0.145 | 0.700 |
| F-PCMCI | 0.404 | 0.626 | 0.990 | 0.662 | 0.110 | 0.677 |
| Dynotears | 0.309 | 0.315 | 0.841 | 0.301 | 0.023 | 0.344 |
| Nts-Notears | 0.283 | 0.346 | 0.838 | 0.394 | 0.084 | 0.427 |

Table 17: Robustness evaluation for multi-violation datasets ($\mathbf{V}_{\text{double,com}}$ and $\mathbf{V}_{\text{double,conf}}$) measured as **normalized SHD** and for the recovery of $G^{\text{LSG}}$ . For comparison, we also report the corresponding individual violation results. Combining violations further reduces the robustness of CD methods across the board.

## D.7 NONLINEAR CONDITIONAL INDEPENDENCE TESTS

As we rely on linear conditional independence (CI) tests (Partial Correlation and Robust Partial Correlation) for constraint-based algorithms, we conducted a small study to assess potential performance gains from using nonlinear CI tests (in this case, GPDC) instead. For this, we compared performance on a particular violation ($\mathbf{V}_{\text{nl,rbf,small}}$) where we expected the highest impact from this hyperparameter extension. We report the results in Table 18. When using GPDC to run PCMCI+, we found that robustness notably improved. However, it still did not exceed that of other methods. Because the use of nonlinear CI tests increases the computational load immensely (in our settings, roughly a hundredfold), and in light of these results, we refrained from including them in the hyperparameter search space in the main section of the paper.

## D.8 LARGER GRAPHS

To explore settings beyond our standard experimental regimes, we evaluated method robustness under $\mathbf{V}_{\text{inno,com}}$ on *large* graph structures (12 variables, 3 lags), comparing it to our *small* (5, 3) and *big* (7, 4) setups. We include the results in Table 19 to Table 21. We found that system size affects robustness heterogeneously: while Nts-Notears remains relatively stable, GVAR and PCMCI+ suffer notable performance drops as size increases. This indicates that differences between methods likely become more pronounced on larger graphs. However, due to the more-than-quadratic computational scaling of many methods with graph size, we refrained from a comprehensive evaluation of this effect, as it would have forced us to slim down our empirical protocol.

| $\mathbf{V}_{\text{nl,rbf,small}}$ (SHD) | $G^{\text{LWCG}}$ | $G^{\text{INST}}$ | $G^{\text{LSG}}$ |
|---|---|---|---|
| CrossCorrelation | 0.441 | † | 0.368 |
| CausalPretraining | 0.458 | † | 0.391 |
| GVAR | 0.399 | † | 0.336 |
| Varlingam | 0.410 | 0.775 | 0.370 |
| PCMCI | 0.510 | † | 0.403 |
| PCMCI+ | 0.473 | 1.000 | 0.380 |
| SVAR-RFCI | 0.446 | 1.000 | 0.366 |
| F-PCMCI | 0.463 | 0.635 | 0.422 |
| Dynotears | 0.535 | 0.504 | 0.420 |
| Nts-Notears | 0.387 | 0.649 | 0.334 |
| PCMCI+ (GPDC) | 0.420 | 1.000 | 0.336 |

Table 18: Robustness comparison for $\mathbf{V}_{\text{nl,rbf,small}}$ of all Methods and PCMCI+ with a nonlinear conditional independence test. Even tho robustness is visibly increased, other methods still show higher robustness against this particular violation.

| $G^{\text{LWCG}}$ (SHD) | $\mathbf{V}_{\text{inno,com,small}}$ | $\mathbf{V}_{\text{inno,com,big}}$ | $\mathbf{V}_{\text{inno,com,large}}$ |
|---|---|---|---|
| CrossCorrelation | 0.588 | 0.748 | 0.895 |
| CausalPretraining | 0.727 | † | † |
| GVAR | 0.371 | 0.443 | 0.517 |
| Varlingam | 0.258 | 0.262 | 0.249 |
| PCMCI | 0.436 | 0.541 | 0.762 |
| PCMCI+ | 0.552 | 0.678 | 0.814 |
| SVAR-RFCI | 0.460 | 0.611 | 0.811 |
| F-PCMCI | 0.377 | 0.530 | 0.944 |
| Dynotears | 0.322 | 0.417 | 0.438 |
| Nts-Notears | 0.341 | 0.358 | 0.323 |

Table 19: Impact of system size on method robustness for the $\mathbf{V}_{\text{inno,com}}$ violation, measured as normalized SHD and for $G^{\text{LWCG}}$. Evaluated on **small** (5, 3), **big** (7, 4), and **large** (12, 3) settings, we observe heterogeneous scaling effects: Nts-Notears remains stable, whereas GVAR and PCMCI+ show degraded robustness as graph size increases.

| $G^{\text{INST}}$ (SHD) | $\mathbf{V}_{\text{inno,com,small}}$ | $\mathbf{V}_{\text{inno,com,big}}$ | $\mathbf{V}_{\text{inno,com,large}}$ |
|---|---|---|---|
| Varlingam | 0.905 | 0.907 | 0.929 |
| PCMCI+ | 1.000 | 1.000 | 1.000 |
| SVAR-RFCI | 1.000 | 1.000 | 1.000 |
| F-PCMCI | 0.987 | 0.998 | 1.000 |
| Dynotears | 0.743 | 0.771 | 0.768 |
| Nts-Notears | 0.909 | 0.837 | 0.828 |

Table 20: Impact of system size on method robustness for the $\mathbf{V}_{\text{inno,com}}$ violation, measured as normalized SHD and for $G^{\text{INST}}$. We found little impact of the size on method robustness.

| $G^{\mathrm{LSG}}$ | $\mathbf{V}_{\mathrm{inno,com,small}}$ | $\mathbf{V}_{\mathrm{inno,com,big}}$ | $\mathbf{V}_{\mathrm{inno,com,large}}$ |
|---|---|---|---|
| CrossCorrelation | 0.476 | 0.589 | 0.835 |
| CausalPretraining | 0.625 | † | † |
| GVAR | 0.324 | 0.380 | 0.482 |
| Varlingam | 0.318 | 0.191 | 0.205 |
| PCMCI | 0.341 | 0.370 | 0.641 |
| PCMCI+ | 0.440 | 0.519 | 0.770 |
| SVAR-RFCI | 0.368 | 0.487 | 0.740 |
| F-PCMCI | 0.335 | 0.473 | 0.938 |
| Dynotears | 0.266 | 0.318 | 0.386 |
| Nts-Notears | 0.283 | 0.283 | 0.285 |

Table 21: Impact of system size on method robustness for the $\mathbf{V}_{\mathrm{inno,com}}$ violation, measured as normalized SHD and for $G^{\mathrm{LSG}}$. Evaluated on **small** (5, 3), **big** (7, 4), and **large** (12, 3) settings, we observe heterogeneous scaling effects: Nts-Notears remains stable, whereas GVAR and PCMCI+ show degraded robustness as graph size increases.

# E  APPENDIX — LLM USAGE

During the preparation of this manuscript, we used Gemini 2.5 Pro to refine sentence structure and correct grammatical errors. We reviewed and edited all AI-generated suggestions and take full responsibility for the publication's final content.

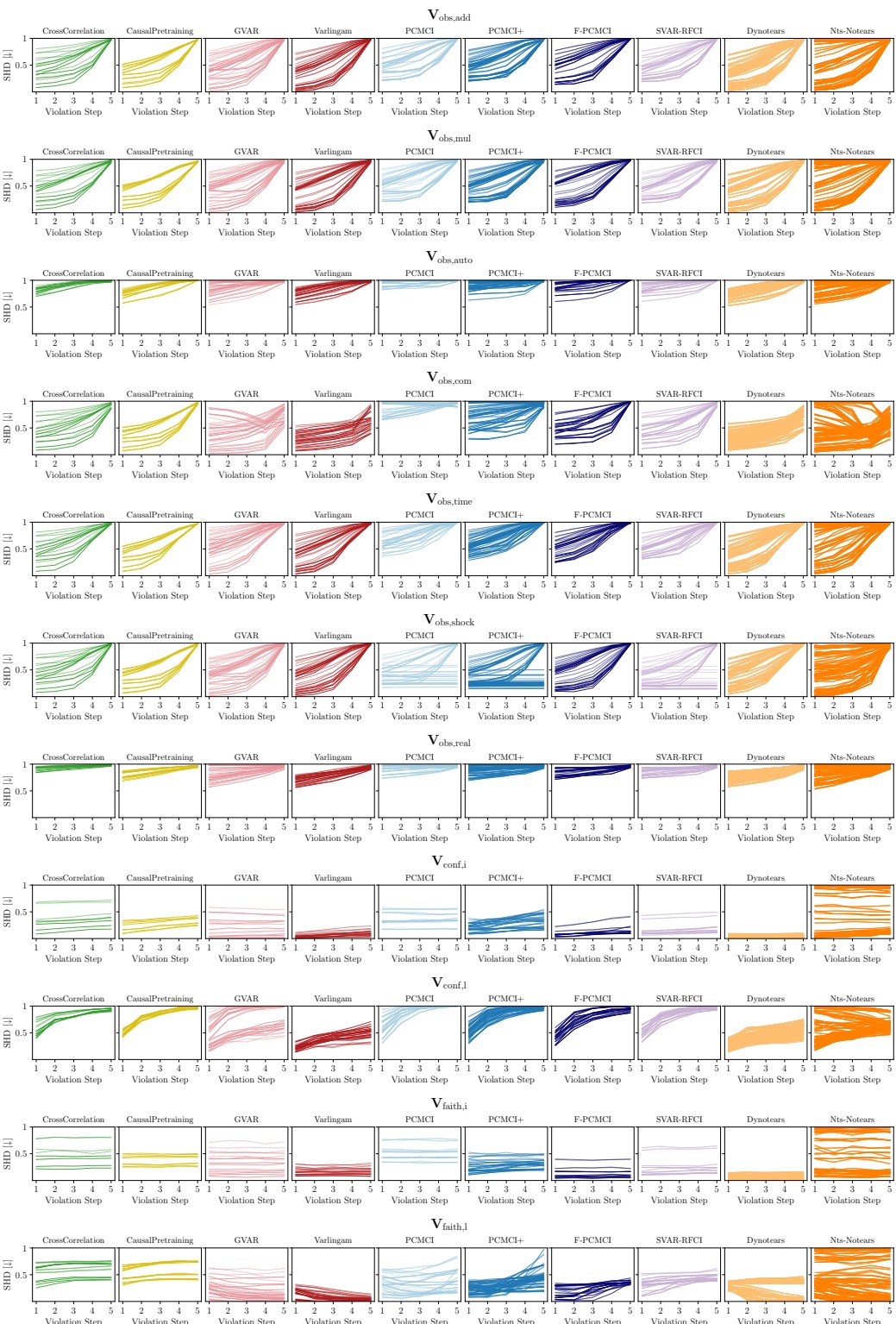

Figure 37: Performance differences (measured as normalized SHD) when uncovering the LWCG depending on hyperparameter selection and data regime for the violations $\mathbf{V}_{obs,add}$, $\mathbf{V}_{obs,mul}$, $\mathbf{V}_{obs,auto}$, $\mathbf{V}_{obs,com}$, $\mathbf{V}_{obs,time}$, $\mathbf{V}_{obs,shock}$, $\mathbf{V}_{obs,real}$ $\mathbf{V}_{conf,inst}$, $\mathbf{V}_{conf,lagged}$, $\mathbf{V}_{faith,inst}$, $\mathbf{V}_{faith,lagged}$.

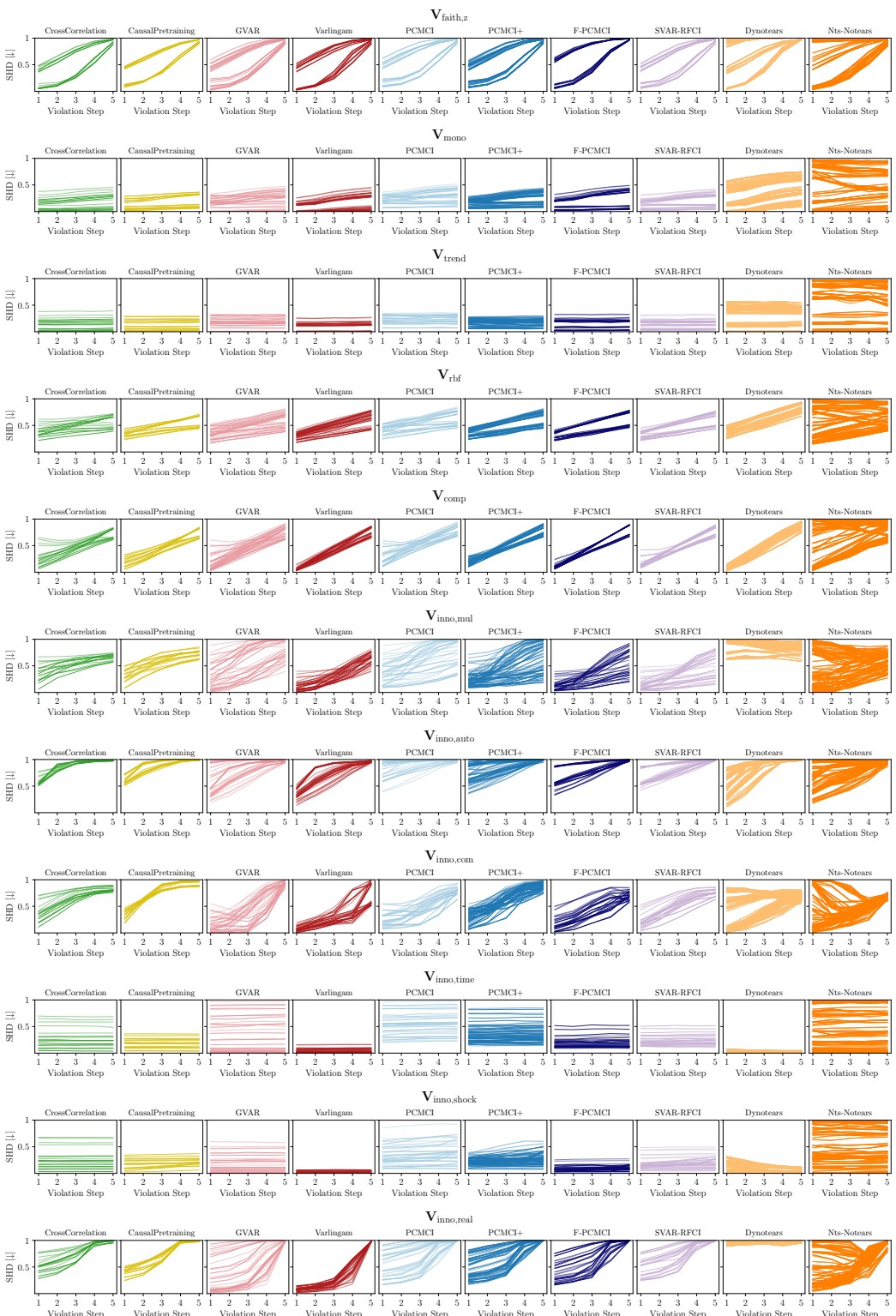

Figure 38: Performance differences when uncovering the LWCG depending on hyperparameter selection and data regime for the violations $\mathbf{V}_{\text{faith,zero}}$, $\mathbf{V}_{\text{nl,mono}}$, $\mathbf{V}_{\text{nl,trend}}$, $\mathbf{V}_{\text{nl,rbf}}$, $\mathbf{V}_{\text{nl,comp}}$ $\mathbf{V}_{\text{inno,mul}}$, $\mathbf{V}_{\text{inno,auto}}$, $\mathbf{V}_{\text{inno,com}}$, $\mathbf{V}_{\text{inno,time}}$, $\mathbf{V}_{\text{inno,shock}}$, $\mathbf{V}_{\text{inno,real}}$.

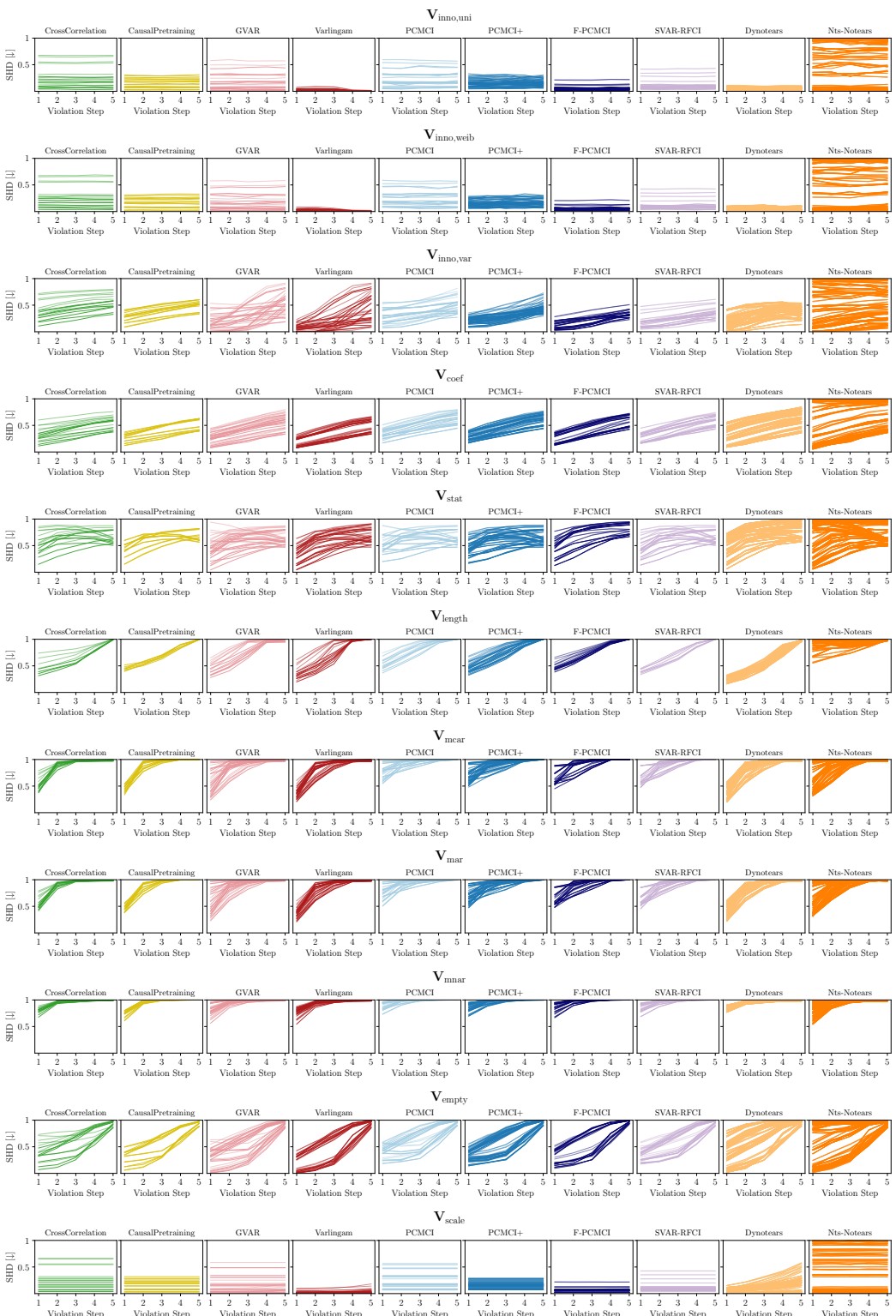

Figure 39: Performance differences (measured as normalized SHD) when uncovering the LWCG depending on hyperparameter selection and data regime for the violations $\mathbf{V}_{\text{inno,uni}}$, $\mathbf{V}_{\text{ino,weib}}$, $\mathbf{V}_{\text{inno,var}}$, $\mathbf{V}_{\text{coef}}$, $\mathbf{V}_{\text{stat}}$, $\mathbf{V}_{\text{length}}$ $\mathbf{V}_{\text{mcar}}$, $\mathbf{V}_{\text{mar}}$, $\mathbf{V}_{\text{mnar}}$, $\mathbf{V}_{\text{empty}}$, $\mathbf{V}_{\text{scale}}$.

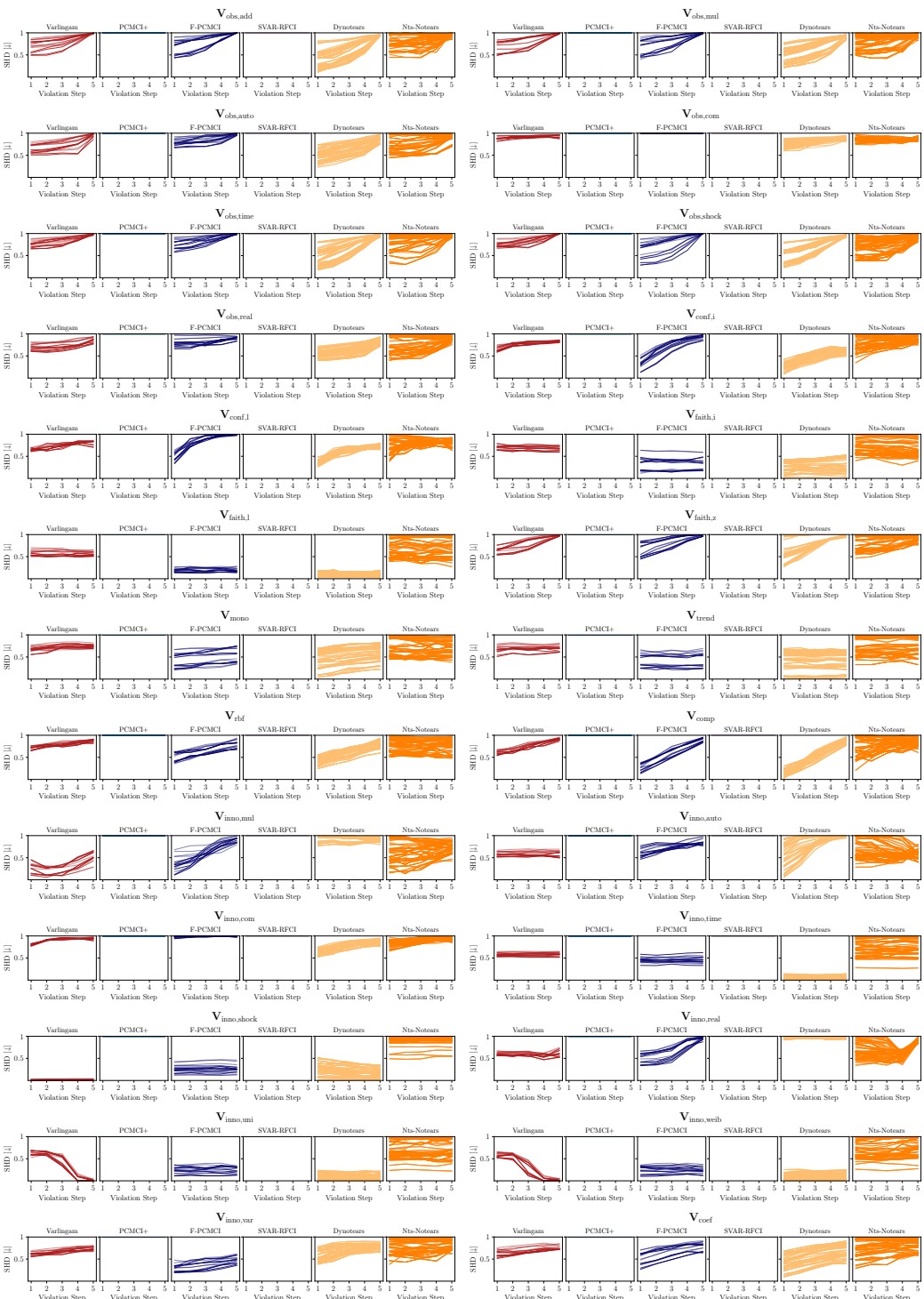

Figure 40: Performance differences (measured as normalized SHD) when uncovering INST depending on hyperparameter selection and data regime for the violations $\mathbf{V}_{obs,add}$, $\mathbf{V}_{obs,mul}$, $\mathbf{V}_{obs,auto}$, $\mathbf{V}_{obs,com}$, $\mathbf{V}_{obs,time}$, $\mathbf{V}_{obs,shock}$, $\mathbf{V}_{obs,real}$, $\mathbf{V}_{conf,inst}$, $\mathbf{V}_{conf,lag}$, $\mathbf{V}_{faith,inst}$, $\mathbf{V}_{faith,lag}$, $\mathbf{V}_{faith,zero}$, $\mathbf{V}_{nl,mono}$, $\mathbf{V}_{nl,trend}$, $\mathbf{V}_{nl,rbf}$, $\mathbf{V}_{nl,comp}$, $\mathbf{V}_{inno,mul}$, $\mathbf{V}_{inno,auto}$, $\mathbf{V}_{inno,com}$, and $\mathbf{V}_{inno,time}$, $\mathbf{V}_{inno,shock}$, $\mathbf{V}_{inno,real}$ $\mathbf{V}_{inno,uni}$, $\mathbf{V}_{inno,weib}$, $\mathbf{V}_{inno,var}$, $\mathbf{V}_{stat}$, $\mathbf{V}_{coef}$

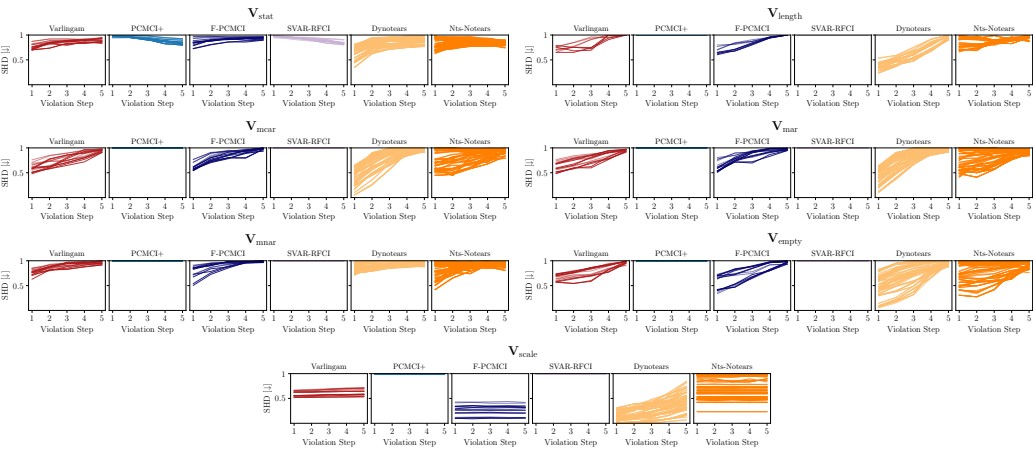

Figure 41: Performance differences (measured as normalized SHD) when uncovering INST depending on hyperparameter selection and data regime for the violations $\mathbf{V}_{\text{length}}$, $\mathbf{V}_{\text{mcar}}$, $\mathbf{V}_{\text{mar}}$, $\mathbf{V}_{\text{mnar}}$, $\mathbf{V}_{\text{empty}}$ and $\mathbf{V}_{\text{scale}}$.

