# OpenReview forum: "TCD-Arena: Assessing Robustness of Time Series Causal Discovery Methods Against Assumption Violations"
_ICLR.cc/2026/Conference — ICLR 2026 Poster_

### Official Review · Reviewer_9gRA · 2025-10-23

**Soundness:** 3
**Presentation:** 3
**Contribution:** 3
**Rating:** 6
**Confidence:** 4

**Summary:**

The paper introduces TCD-Arena, a modular, open-source testbed for stress-testing time-series causal discovery (CD) methods under stepwise assumption violations. It defines three target graphs—WCG (lag-window graph), SG (lagged summary graph), and INST (instantaneous graph)—and implements 27 violation families (e.g., observational/innovation noise structures, latent confounding, faithfulness violations, nonlinear mechanisms, non-stationarity, sample length, data quality, scaling). For each violation, five intensity levels are simulated across 8 data regimes (varying T, D, L, sparsity, and presence of contemporaneous effects). The study evaluates eight CD methods spanning major paradigms (cross-correlation, VAR/GVAR, VARLiNGAM, PCMCI/PCMCI+, DYNOTEARS, NTS-NOTEARS, and CausalPretraining), with extensive hyper-parameter searches, producing over 50 million runs.

Key findings: (i) Granger-based approaches are comparatively robust for lagged edges (WCG/SG); (ii) Dynotears and PCMCI+ are strongest on instantaneous edges (INST) while VARLiNGAM lags; (iii) deep models (CausalPretraining, NTS-NOTEARS) underperform simpler ones; (iv) under-specifying the max lag hurts all methods, whereas over-specifying is mostly benign; (v) ensembles (simple average, linear, MLP, ConvMixer) improve average robustness.

**Strengths:**

1. First comprehensive time-series-focused robustness arena with graded violation intensities rather than binary toggles; covers 27 violations with clear, reproducible generators.
2. Separating WCG/SG/INST clarifies evaluation of lagged vs. instantaneous structure—often conflated in prior work.
3. Massive run budget with systematic hyper-parameter sweeps; sensible aggregation (AUROC across regimes × levels) and ablations on lag mis-specification and hyper-parameter sensitivity.
4. Ensembling insight shows that meta-learners fed only with method outputs (no raw input) increase robustness and reduce variance across violations.

**Weaknesses:**

1. AUROC on adjacency recovery is threshold-free but may obscure class-imbalance and orientation errors. SHD, F1/AUPRC, precision, and arrow-orientation accuracy would provide complementary views; some are included in the appendix but not highlighted.
2. The DGP (Eq. 2) is additive with univariate link functions and a fixed max lag; some violations (e.g., nonlinearity via RBF/mixtures) are still univariate. Methods exploiting multivariate interactions or cycles may not be assessed.
3. Non-stationarity is simulated by redrawing coefficients A while keeping the skeleton fixed; real systems also change skeletons, contexts, regimes, and sampling rates.
4. Only one faithfulness-violation pattern is implemented; non-Gaussian noise is introduced for innovation but not matched to LiNGAM’s identifiability conditions (e.g., independent non-Gaussian; instantaneous vs lagged). The result that VARLiNGAM is not advantageous under this violation could reflect misspecification.
5. Now it seems only MCAR is used for missingness. More missingness mechanisms, such as MAR and MNAR, should be examined.
6. Meta-learners are trained on simulated violations; transfer to real data under distribution shift is acknowledged but not tested. The Pareto oracle is instructive but unattainable. Guidance on how to pick ensembles in practice is limited.
7. Given 50M runs, a brief computational cost profile (GPU hours, carbon) and per-method runtime/memory would help users scope experiments.

**Questions:**

1. Could TCD-Arena add skeleton changes and context-specific mechanisms (e.g., regime switches) to stress methods designed for heterogeneous data?
2. What meta-features (if any) about datasets improve selection? Did you try cross-violation training and measure out-of-violation generalization?
3. Interesting result that ↑𝐿 is safe while ↓𝐿 is harmful. Can you show sample-complexity curves for varying 𝑇 under ↑𝐿 (variance vs bias trade-off)?

---

> ### Author Response · Authors · 2025-11-22
> **We thank the reviewer for his comments and his appreciation of our efforts. In the following, we will comment on the remaining concerns**
>
> ---
> **AUROC and other metrics (Weakness 1)**
>
> To further improve our evaluation protocol, we currently compute additional Metrics (normalized SHD, FPR, TPR) for all violations. We have attached a sample for the $V_{inno, com}$ violation in the **general comment**.  As the ranking can be affected by the metric, we will expand the discussion of the differences in Appendix D1 and reference it accordingly in the main part.
>
>
> ---
> **Cycles, and nonstationarity (Weakness 2,3, and Question 1)**
>
> As correctly noted, we omit or restrict certain violations from the TCD-arena. We do this because the causal ground truth becomes less clear or varies for the violation, making it difficult to evaluate it alongside the other violations.
> As noted in L299-306, generating cycles through temporal aggregation creates ambiguity in the causal ground truth, as it depends on the resolution [1]. Changing causal skeletons is already integrated in TCD-arena, but we omitted it from our original study due to the changing ground truth. However, due to requests from multiple reviewers to include this case, we added it as an additional violation.  We have attached the results in the **general comment** and will incorporate them into the manuscript.
>
> ---
> **Multivariate relationships (Weakness 2)**
>
> We agree that an assessment of multivariate effects would be an interesting extension. However, as this is the first study on assumption violation for time series, we intentionally restrict the scope to the more specific form of Eq. 2, as it is arguably the simplest form of the equation. Furthermore, we include methods such as Dynotears [2], Varlingam [3], and Causal Pretraining [4], which explicitly assume this SCM.
>
> ---
> **Alternative violation implementations (Weaknesses 4, 5, and Question 1)**
>
> We find this interesting and are working on integrating four additional alternative violation implementations to address this request :
> - Unfaithfulness (through small but non-zero effects)
> - Missing-not-at-random data:  $p(R^t_i = 1) = \frac{1}{1 + e^{-(a+bx^t_i)}}$ to specify the probability of any value to be unmeasured ($R^t_i = 1$ denotes that a value is missing, $a$ and $b$ are hyperparameters.)
> - Changing causal skeleton (as described above)
> - Semi-synthetic observational noise (requested by reviewer 3).
>
> We include the results for the last two violations in the **general comment**.
>
> ---
> **Non-Gaussian Noise (Weakness 4)**
>
> We address this weakness with three points. First, VAR-LiNGAM estimates lagged relationships through an initial VAR step, which does not rely on non-Gaussianity. Therefore, our use of Gaussian innovation noise as a baseline should not affect the discovery of lagged effects (e.g., Fig. 1a).
> Second, deploying standard normal noise was a deliberate methodological choice as it serves as a unique neutral starting point. Choosing any specific non-Gaussian distribution (e.g., Weibull, Uniform) as the default would raise questions concerning the specific selection in each tested violation.
> Third, the observation that VAR-LiNGAM's performance is hindered even when innovation noise is technically non-Gaussian (as seen in our violation scenarios, Figs. 7, 24) is a compelling result, highlighting a potential sensitivity of the method.
>
> ---
> **Meta Learners (Weakness 6)**
>
> To enhance the practical applicability of ensemble approaches, we revised the evaluation protocol and conducted an additional study using the trained ensembles on a real-world benchmark (CausalRivers [5]). We include the results and some discussion on this study in the **general comment**.
>
>
>  ---
> **Pareto Oracle (Weakness 6)**
>
> The Pareto oracle is a theoretical concept designed solely to serve as a reference point for contextualizing the ensembling performance. It has no practical applications.
>
> ---
> **Computational Costs  (Weakness 7)**
>
> As there were multiple requests for more details on the computational costs, we provide information on the average computational time in the **general comment** and will integrate it into the manuscript accordingly.
>
> ---
> **Meta features for the ensembles (Question 2)**
>
> As we do not test ensembling with access to the datasets, we cannot provide concrete information on this. However, we would assume that access to the data could help ensembles to assess features such as observational noise, variability, or seasonality, or simply the length of the time series, which could provide information for CD-method selection.
>
> ---
>  **L↑ and  L↓ (Question 3)**
>
>  In  L↓, we reduce the number of estimated edges to a suboptimal number, which makes it impossible to estimate all links. This is likely also the reason why it results in a much severe performance drop. We are unsure if there is a bias-variance tradeoff here, but we may be misunderstanding the question. Could you rephrase this?

---

> > ### Author Response · Authors · 2025-11-22
> >
> > [1] Runge, Jakob. "Causal network reconstruction from time series: From theoretical assumptions to practical estimation." Chaos: An Interdisciplinary Journal of Nonlinear Science 28.7 (2018).
> >
> > [2] Pamfil, Roxana, et al. "Dynotears: Structure learning from time-series data." International Conference on Artificial Intelligence and Statistics. Pmlr, 2020.
> >
> > [3] A. Hyvärinen, K. Zhang, S. Shimizu, and P. O. Hoyer. Estimation of a structural vector autoregression model using non-Gaussianity. Journal of Machine Learning Research, 11: 1709-1731, 2010.
> >
> > [4] Stein, Gideon, Maha Shadaydeh, and Joachim Denzler. "Embracing the black box: Heading towards foundation models for causal discovery from time series data." arXiv preprint arXiv:2402.09305 (2024).
> >
> > [5]  Stein, Gideon, et al. "CausalRivers-Scaling up benchmarking of causal discovery for real-world time-series." The Thirteenth International Conference on Learning Representations.

---

> ### Comment · Reviewer_9gRA · 2025-11-24
> **Rephrasing the question 3**
>
> Thank you for the detailed rebuttal, which has solved most of my questions, and I apologize for the confusion about question 3. My question wasn’t about reducing the number of edges an estimator may output. I was asking about misspecifying the maximum lag hyperparameter (say 𝐿^) used by the methods relative to the true DGP maximum lag (𝐿*).
>
> **Rephrased question:** In your “𝐿↑ / 𝐿↓” study, could you please distinguish the effects of over-specifying the maximum allowed lag (𝐿^>𝐿*, i.e., 𝐿↑) versus under-specifying it (𝐿^<𝐿*, i.e., 𝐿↓).

---

> > ### Author Response · Authors · 2025-11-25
> >
> > Thank you for rephrasing the question.
> >
> > We observe two distinct effects:
> >
> > 1. **Overspecification:** Setting the maximum lag too high has a minor impact, as most methods demonstrate a high degree of robustness, correctly identifying the additional candidate links as non-causal.
> > 2. **Underspecification:** Setting the maximum lag too low causes a mechanistic failure. It renders the discovery of true, longer-lagged dependencies impossible, resulting in unavoidable false negatives and significantly degraded performance.
> >
> > We distinguish these cases and discuss their effects in l. 392 - 405 of the manuscript.
> >
> > As the initial question referenced the relationship between $T$ and $L\uparrow$, we attached a small study comparing all methods under $V_{length}$ specifically. While the performance drop is slightly more pronounced compared to all violations (Table 1), we still consider the performance drop to be minor. Therefore, the conclusion concerning the overspecification of lags remains the same. As we find this kind of fine-grained analysis interesting, we will include it in Appendix D.4.
> >
> > Thank you for this suggestion.
> >
> >
> > | AUROC ($V_{\text{length}}$)       | Causal Pretraining      | Pcmci   | NTS-Notears | PCMCI+ | Varlingam | Dynotears | GVAR     | Cross Corr |
> > |:--------------|:--------|:--------|:-----------|:----------|:----------|:----------|:--------|:-----------------|
> > | **Correct L** | 0.895   | 0.884   | 0.885      | 0.898     | 0.894     | 0.926     | 0.901   | 0.885            |
> > | **↑L**        | 0.895   | 0.869   | 0.887      | 0.884     | 0.863     | 0.915     | 0.874   | 0.881            |
> > | **Difference**| 0.000   | -0.015  | 0.001      | -0.014    | -0.031    | -0.011    | -0.027  | -0.004           |

---

> > > ### Comment · Reviewer_9gRA · 2025-11-25
> > > **Official Comment by Reviewer 9gRA**
> > >
> > > I thank the authors for the detailed follow-up rebuttal. I decided to raise the score of soundness from 3 to 4, but keep the same overall rating as before, considering the overall contribution of this work.

---

### Official Review · Reviewer_B3QT · 2025-10-28

**Soundness:** 3
**Presentation:** 2
**Contribution:** 1
**Rating:** 2
**Confidence:** 3

**Summary:**

The paper introduces TCD-Arena, a modular benchmarking toolkit for time-series causal discovery (CD) with a focus on robustness under stepwise assumption violations. It evaluates 8 CD methods spanning major paradigms across 27 violation types (each with graded severity) and multiple data regimes, totaling 50M runs. The study contrasts recovery of three graph notions (GWCG, GSG, GINST), analyzes hyperparameter sensitivity, and explores ensembling of CD methods as a route to improved robustness. Results suggest (i) notable variability in robustness across methods and violations, (ii) simple Granger-style approaches are comparatively robust for lagged effects, (iii) deep methods can trail simpler baselines, and (iv) ensembling provides tangible gains.

**Strengths:**

I found the paper interesting and I do think that there is a need more papers like in the causal discovery community.

This paper study the robustness to violated assumptions is central for CD adoption

Evaluating GWCG, GSG, and GINST clarifies what each method is able to recover.

It studies hyperparameter sensitivity analysis which is important and often ignored. This it has a   practical value for deployers.

If the toolkit is released as open-source with configs and seeds, TCD-Arena could become a useful community resource.

**Weaknesses:**

* It is very difficult to understand the contribution of the paper without going to the Appendix.

* Despite the motivation that real-world ground truth is scarce, the study remains fully synthetic. This limits the external validity; even a small real or semi-synthetic case  would strengthen claims.

* Many violations are reasonable, but several design choices (e.g., faithfulness violations via path cancellation; innovation noise blends; stationarity via coefficient resampling) feel one specific instantiation among many. It’s unclear how sensitive findings are to alternative parameterizations. More ablations/justifications would help.

* Some algorithms do not target GINST, others require max-lag specification, and assumptions differ (e.g., non-Gaussianity). While you note this, the aggregated “robustness” comparisons may conflate target mismatch with method weakness. Clearer tracks per target/assumption would avoid this.

* Meta-learners trained on synthetic violations may not transfer across domains/violations. The practical ensembles’ training protocol need clearer specification.

* Recent CD algorithms for time series causal discovery are not included.


* While the topic of causal discovery robustness is important, this paper is primarily a benchmarking and dataset-style contribution based entirely on synthetic data. It does not introduce a new learning algorithm, theoretical insight, modeling principle, or a new real-data benchmark, types of contributions that, to my understanding, are typically expected for ICLR’s main track. Even though ICLR does occasionally accept benchmark-oriented papers, such works usually have a broad and transformative impact (for example, by introducing a new real-world benchmark or establishing a widely adopted evaluation framework). In its current form, the contribution of this paper seems more incremental in scope (even though I think it is very interesting and needed). That said, I may not be fully aware of the current editorial stance on such submissions, and I would defer to the area and meta chairs regarding venue fit. Given its focus, the work might be better suited for a Datasets and Benchmarks track at a major AI or ML conference, where such empirical frameworks are the main focus.

**Questions:**

* Can you consider testing methods on larger graphs?

* Can you include at least one semi-synthetic or interventional real dataset to validate ranking consistency (even if partial ground truth)?

* Do your qualitative conclusions hold under AUPRC, (adj)SHD, and orientation error measures?

* Any cases where AUROC misleads?

* Why that specific cancellation pattern? Have you tried near-unfaithful settings (small but nonzero effects) and do trends persist?

* How were search budgets matched across methods? Can you report per-method best-vs-average gaps in the main text and show robustness profiles over the top-k configs?

* How do you prevent leakage from the meta-learner training to evaluation regimes? Any evidence of transfer to unseen violation types or severities? Stationarity violation.

* You resample coefficients while keeping the skeleton fixed. Have you tested changing skeletons over segments (structural breaks)?

* Can you include or at least discuss clearly recent algorithms like TiMINo[1], varFCI [2], LPCMCI [3], PCGCE [4] J-PCMCI[5], CBNB [6], SpaceTime [7]


* Can you discuss clearly the contribution of this paper with respect to other ICLR benchmark papers on causal discovery. Like for example: [8] (which was cited but not thoroughly discussed)

Minor:

* You might consider citing [9] in addition to Pearl (2009) and Peters et al. (2017) when referring to a comprehensive review of causal discovery in i.i.d. settings, as this book provides a more in-depth treatment of constraint-based causal discovery methods (PC and FCI) than the other two references.


* In much of the existing literature, the WCG and summary graph are defined differently: both typically include instantaneous relations. It may therefore be helpful to explicitly state that your definitions diverge from the conventional ones. You might also consider using distinct notation (e.g., LWCG for lagged window causal graph and LSG for lagged summary graph) to avoid confusion. Additionally, note that the literature also defines a third type of graph, the extended summary causal graph, which, as I understand it, combines features of the lagged summary graph and the instantaneous graph.


References:

* [1] Peters et al. Causal Inference on Time Series using Restricted Structural Equation Models. Neurips, 2013

* [2] Malinsky and Spirtes. Causal Structure Learning from Multivariate Time Series in Settings with Unmeasured Confounding. KDD workshop on CD. 2018

* [3] Gerhardus and Runge. High-recall causal discovery for autocorrelated time series with latent confounders. Neurips, 2020

* [4] Assaad et al. Discovery of extended summary causal graphs from time series. UAI. 2022

* [5] Gunther et al. Causal Discovery for time series from multiple datasets with latent contexts. UAI, 2023

* [6] Bystrova et al. Causal Discovery from Time Series with Hybrids of Constraint-Based and Noise-Based Algorithms. TMLR, 2024

* [7] Ameche et al. SpaceTime: Causal Discovery from Non-Stationary Time Series. AAAI, 2025

* [8] Cheng et al. CAUSALTIME: REALISTICALLY GENERATED TIMESERIES FOR BENCHMARKING OF CAUSAL DISCOVERY. ICLR, 2024.

* [9] Spirtes et al. Causation, prediction, and search. MIT press. 2000.

---

> ### Author Response · Authors · 2025-11-22
> **We would like to thank the reviewer for their extensive review and interest in our work. We appreciate the fact that the reviewer agrees on the need for more papers related to empirical evaluation in the causal discovery community while also providing constructive feedback.**
>
> ---
> **Venue fit (Weakness 7)**
>
> We would like to note that the ICLR main track has explicitly called for datasets and benchmarks (since 2024) with no further specifications regarding real-world data. Our work is also grouped in this category (See primary area). Incidentally, the invited talk “The Emerging Science of Benchmarks” from ICLR 2024 [1] actually motivated us to construct this benchmark, as we believe there is a dire need for more benchmarking in Causal Discovery.
>
> ---
> **Synthetic data source (Weakness 2 Question 2)**
>
> You are correct that TCD-arena is fully synthetic. This is by design to allow for controlled experiments not possible with real-world data (in agreement with Poinsot [2]). Nonetheless, to further alleviate this concern, we have now added a semi-synthetic violation using noise extracted from stock price data. We also note that our core findings, such as the robustness of VAR approaches, are consistent with those reported on the real-world CausalRivers benchmark [3], which supports the use of a combination of real-world and synthetic benchmarks.
>
> ---
> **Distinction to other ICLR benchmarks (Question 10)**
>
> Our key distinction from recent benchmarks, such as CausalRivers (real-world)[3] and CausalTime (semi-synthetic)[4], is our fully synthetic nature. Their reliance on real data means the underlying SCMs are partly unknown, which limits controlled evaluation. In contrast, TCD-Arena provides complete control over the SCM, enabling precise testing of how specific SCM features impact performance. Therefore, TCD-Arena fills a crucial gap for controlled robustness analysis. We have updated the related work to emphasize this.
>
> ---
> **Reliance on the Appendix (Weakness 1)**
>
> Our objective when formatting the paper was to provide a comprehensive, high-level overview of TCD-arena and our study, with the appendix serving as a reference for detailed information. Is there specific content from the Appendix that you would like to see in the main part to improve comprehensibility? We would attempt to integrate it accordingly.
>
> ---
> **Larger graphs (Question 1)**
>
> We have conducted an additional study on larger graphs (14 nodes for V_{inno,com,14} ) and will integrate the results into the Appendix. Notably, the best method (Varlingam) is consistent with the results in the main section.
>
> |Normalied SHD |Pcmci|NTS-Notears|Varlingam|Dynotears|GVAR|Cross Corr|
> |:---|:---|:---|:---|:---|:---|:---|
> |$V_{inno,com,14}$|0.704|0.439|**0.286**|0.623|0.516|0.833|
>
> ---
> **Violations design (Weakness 3 and Question 2,5,8)**
>
> We agree that for many violations, alternative implementations are possible. We therefore integrate an additional four violations ( **general comment**) that target your requests.
>
>
> ---
> **Metric differences (Question 3.4)**
>
> We are currently calculating additional metrics (SHD, TPR, FPR) and have attached a comparison between metrics for ($V_{inno, com}$) along with a discussion in the **general comment**. We will change the main metric in the paper to normalized SHD to align with the literature.
>
> ---
> **Search budgets (Question 6)**
>
> We have added information on the computational costs for each CD method in the **general comment** and will integrate it into the manuscript accordingly.
>
> ---
> **Average vs. Best performance (Question 7)**
>
> In the main section of the paper, we report the best per method as well as the average performance. While we also discuss the drop for NTS-notears in particular, we refrained from placing more emphasis on this, as the difference in average performance strongly relies on the search space and should be taken with care.
>
> ---
> **Clearer tracks per target/assumption (Weakness 4)**
>
> Our study distinguishes between $G^{INST}$ and $G^{WCG}$ to separate comparisons. The max lag is required for all methods except Causal Pretraining.
>  While our choice of standard normal noise negatively affects Varlingam's performance, the discovery of lagged effects is unaffected by our choice of a standard normal noise. We note this accordingly in our manuscript.
>
>
> ---
> **Ensemble evaluation (Weakness 5 and Question 7)**
>
> We revised the ensembling evaluation protocol to address o.o.d generalization by testing the trained ensembles directly on real-world data (CausalRivers [2]). We include the results in the **general comment** and look forward to discussing them in the manuscript.
>
> ---
> **More algorithms (Weakness 6 and Question 8)**
>
> Currently, we select fairly basic but well-established methods for our study. To further extend the results, we are integrating two additional methods (LPCMCI and SVARRFCI), and we exemplarily include their robustness on V_{inno,com} in the **general comment**.
> Notably, we purposefully excluded algorithms that were developed for specific data properties (e.g., SpaceTime [5] (multi datasets) or CUTS (gappy data) [6].
>
> ---
> **Minor:**
>
> We have changed WCG and SG to LWCG and LSG (for lagged) to clarify the distinction. We also added your suggested reference.

---

> > ### Author Response · Authors · 2025-11-22
> >
> > [1] https://iclr.cc/virtual/2024/invited-talk/21799
> >
> > [2] Poinsot, Audrey, et al. "Position: Causal machine learning requires rigorous synthetic experiments for broader adoption." arXiv preprint arXiv:2508.08883 (2025).
> >
> > [3] Stein, Gideon, et al. "CausalRivers-Scaling up benchmarking of causal discovery for real-world time-series." The Thirteenth International Conference on Learning Representations.
> >
> > [4] Cheng, Yuxiao, et al. "CausalTime: Realistically Generated Time-series for Benchmarking of Causal Discovery." The Twelfth International Conference on Learning Representations.
> >
> > [5] Mameche, Sarah, et al. "SPACETIME: Causal Discovery from Non-Stationary Time Series." Proceedings of the AAAI Conference on Artificial Intelligence. Vol. 39. No. 18. 2025.
> >
> > [6] Cheng, Yuxiao, et al. "CUTS: Neural Causal Discovery from Irregular Time-Series Data." The Eleventh International Conference on Learning Representations.

---

> > > ### Comment · Reviewer_B3QT · 2025-11-25
> > >
> > > I appreciate the authors’ rebuttal and the clarifications provided. I would like to point out a minor misinterpretation: regarding Weakness 7, my comment was not specifically about benchmarking, but rather about the fully synthetic nature of the evaluation. Nevertheless, after reviewing all the feedback, I have decided not to emphasize this limitation in my final assessment. Concerning Weakness 2, I partially agree with the authors’ response: there are a few initiatives that enable some level of control based on realistic data with possible interventions, such as the Causal Chamber. I sincerely appreciate the authors for including experiments with larger graphs and for adding additional comparison methods. After careful consideration of the rebuttal and the improvements made, I have decided to slightly increase my score.

---

> > > > ### Author Response · Authors · 2025-11-27
> > > >
> > > > Dear Reviewer,
> > > >
> > > > Thank you again for your positive re-evaluation.
> > > >
> > > > We are writing to briefly highlight that, based on your initial suggestion, **we have now added multiple additional semi-synthetic capabilities to our framework and experiments.**
> > > >
> > > >   As detailed in our general comment, this includes using real-world signals to generate semi-synthetic **missingness patterns**, **innovation noise**, and **observational noise**.
> > > >
> > > > We will integrate these cases as additional violations and believe they can bridge the gap between purely synthetic data and real-world complexity, while still allowing us to maintain the full control that our benchmark provides.
> > > >
> > > > Thank you for challenging us on this notion of our work.

---

### Official Review · Reviewer_K4FN · 2025-10-31

**Soundness:** 3
**Presentation:** 3
**Contribution:** 3
**Rating:** 6
**Confidence:** 4

**Summary:**

This paper introduces TCD-Arena, a large-scale and modular benchmark framework for evaluating the robustness of time series causal discovery (CD) algorithms under systematically controlled assumption violations. This study evaluates eight representative CD algorithms through over 50 million runs across 27 assumption violations. Additionally, this work explore hyperparameter sensitivity, model misspecification and ensembling strategies that improve robustness. TCD-Arena is released as an open-source toolkit to support reproducible, large-scale robustness evaluation for time series causal discovery research.

**Strengths:**

The paper presents a comprehensive and rigorously executed empirical benchmark that systematically evaluates the robustness of time series causal discovery methods across diverse assumption violations. By encompassing 27 distinct violation scenarios and analyzing eight representative algorithms, the study establishes a new empirical standard for assessing reliability in time series causal discovery.

**Weaknesses:**

The authors conducted a commendable and well-structured benchmark study. However, there are a few aspects that could improve the paper’s readability and completeness:
- In line 408 and line 411, the references to Fig. 3b are mistakenly written as Fig. 2b.
- In line 2420, the right panel of Figure 18 shows missing results for NTS-NOTEARS under certain conditions.
- Including a runtime comparison across different algorithms would be valuable for practitioners, as it would help guide the selection of time-series causal discovery methods in real-world applications.

**Questions:**

- In line 654 and line 771, the references to Ormaniec et al. and Yi et al. appear to be incomplete. Both works were published at ICLR 2025. Moreover, the title of Yi et al. is fully capitalized, which is inconsistent with the other citation styles. Providing complete and correctly formatted references would help readers in the community quickly identify related advances.
- Could the authors provide more details about the GVAR method and the code source used in the experiments? It seems that this corresponds to an extended version of the standard VAR model, as Table 11 indicates that the method has two hyperparameters, coeff and p-val. Including a brief methodological description would make the work more complete. Also, unless there is a specific reason to retain this name, it might be preferable to change it, since another work [1] is also named GVAR, which could lead to confusion.
- Could the authors elaborate on how AUROC was computed in Figure 1? My understanding is as follows: for each assumption violation scenario, there are five different violation levels, and for each level, 100 datasets are sampled. Each method, under each hyperparameter configuration, computes AUROC across all these datasets. The results are then averaged to obtain an average AUROC per hyperparameter configuration, and the best-performing configuration is reported. Please confirm whether this interpretation is correct.
- The paper defines robustness (line 363) in a way that differs from prior works such as Montagna et al. and Yi et al., which report results based on the best-performing hyperparameter setting per violation. Could the authors discuss the differences between these two evaluation strategies and their respective advantages or limitations? Although there may not yet be a community-wide consensus, a discussion on this distinction would be valuable.

[1] Marcinkevičs R, Vogt J E. Interpretable models for granger causality using self-explaining neural networks[J]. arXiv preprint arXiv:2101.07600, 2021.

---

> ### Author Response · Authors · 2025-11-22
> **We thank the reviewer for their careful assessment. In the following, we address the remaining weaknesses and questions:**
>
> ---
> **Formatting  (Weakness 1,2, and Question 1)**
>
> - We fixed the formatting for the citations. Thank you for pointing them out.
> - We fixed the graphic. It appears that the performance dropped below the axis limit, so it was no longer visible.
>
> ---
> **Textual clarifications (Question 2 3 4)**
>
> - We extended the description of GVAR in the method section to improve clarity.
> - Concerning the robustness computation: For each violation level, we sample 8 data regimes, and for each data regime, we sample 100 data samples. We then calculate the performance metric (e.g., AUROC) for each pack of 100 samples and average over all scores to conclude the robustness of a specific Hyperparameter configuration. Note: A depiction of this process is included in Fig. 9. Please let us know if further clarification is required.
>
> - We added further discussion on the differences between our evaluation protocol and those of Montagna [1] and Yi [2]. We also want to note that we rely on the best-performing hyperparameter configuration, which is identical to the protocol of previous studies. Additionally, provide the average performance in Table 1. In this sense, our protocol builds upon the works in the literature, which rely solely on the best-performing hyperparameter configuration while typically using smaller search spaces [2].
>
> ---
> **Runtime comparison (Weakness 3)**
>
> As multiple reviewers pointed out that information on method runtimes would be interesting, we aggregated a runtime comparison for all tested CD methods and included a summary of them in the **general comment**, along with some discussion.

---

> > ### Author Response · Authors · 2025-11-22
> >
> > [1] Montagna, Francesco, et al. "Assumption violations in causal discovery and the robustness of score matching." Advances in Neural Information Processing Systems 36 (2023): 47339-47378.
> >
> > [2] Yi, Huiyang, et al. "The Robustness of Differentiable Causal Discovery in Misspecified Scenarios." The Thirteenth International Conference on Learning Representations. 2025.

---

> > > ### Comment · Reviewer_K4FN · 2025-11-23
> > > **Official Comment by Reviewer K4FN**
> > >
> > > I appreciate the authors' efforts in addressing the concerns. However, I noticed that the revised manuscript has not yet been uploaded, so I am unable to verify the changes. If the updated version explicitly incorporates the discussed fixes and clarifications, I would be happy to provide stronger support for this work. This paper represents an important contribution to robustness evaluation of time-series causal discovery under assumption violations, and I believe it is valuable to the research community.
> > >
> > > Regarding the authors' statement that Montagna et al. [1] and Yi et al. [2] used "smaller search spaces", I believe this characterization is not entirely accurate. To my knowledge, [1] tunes α ∈ {0.001, 0.01, 0.05, 0.1} and λ ∈ {0.05, 0.5, 2, 5}, and [2] tunes α ∈ {0.001, 0.01, 0.05, 0.1} and λ ∈ {0.005, 0.01, 0.05, 0.5, 2, 5}. Moreover, hyperparameters differ between time-series and non-time-series methods, and these search spaces are therefore not directly comparable. What can be said confidently is that the present work adopts a substantially more comprehensive search space than [3]. I also encourage the authors to discuss the difference between reporting average performance versus best performance. In real-world applications, the optimal hyperparameters are typically unknown. Therefore, reporting average performance has meaningful advantages, and highlighting this point would further strengthen the paper.
> > >
> > > Additionally, I have one more question. In Figures 21–25, some methods exhibit behavior where performance increases as the violation step increases, or decreases first and then increases, or increases first and then decreases. Intuitively, one would expect performance to generally decrease as violations become stronger. For cases where performance behaves contrary to this expectation, an explanation or discussion would be valuable. Such analysis would make the paper more complete and help readers better understand under what circumstances these non-monotonic robustness patterns arise.
> > >
> > > [1] Montagna, Francesco, et al. "Assumption violations in causal discovery and the robustness of score matching." Advances in Neural Information Processing Systems 36 (2023): 47339-47378.
> > >
> > > [2] Yi, Huiyang, et al. "The Robustness of Differentiable Causal Discovery in Misspecified Scenarios." The Thirteenth International Conference on Learning Representations. 2025.
> > >
> > > [3] Ferdous, Muhammad Hasan, Emam Hossain, and Md Osman Gani. "Timegraph: Synthetic benchmark datasets for robust time-series causal discovery." Proceedings of the 31st ACM SIGKDD Conference on Knowledge Discovery and Data Mining V. 2. 2025.

---

> > > > ### Author Response · Authors · 2025-11-24
> > > > **Thank you once more for your quick and helpful review.**
> > > >
> > > > **We have now uploaded a revised manuscript incorporating your suggestions.**
> > > >
> > > > For your convenience, here is a summary of the key changes:
> > > > - Figure References are fixed (l. 409).
> > > > - Figure 18 (l. 2420) was updated.
> > > > - Table 13 now holds information on computational complexity and is included in C5.
> > > > - The citations in l. 768, 648 are updated.
> > > > - The description of GVAR was extended (l. 321-323).
> > > >
> > > > **Concerning smaller search spaces**
> > > >
> > > > We, in fact,  agree with your assessment that a direct comparison with [1] and [2] is not as straightforward as with [3] due to differences in data modalities.
> > > >
> > > > We have revised the corresponding section and also emphasized the problematic nature of optimal hyperparameter selection (l. 360 -368).
> > > >
> > > > Furthermore, we extended the discussion of the results concerning average performance in l. 407-419.
> > > >
> > > > **Additional question**
> > > >
> > > > Thank you for spotting this.
> > > >
> > > > We corrected a formatting bug that caused the x-axis ordering to be shuffled for some violations in the noted figures. This specifically affected the graphics for $V_{nonstat}$ and $V_{empty}$ for both lagged and instantaneous effects.
> > > >
> > > > We updated the affected graphics accordingly (Figures 21–25).
> > > > The curves are now decreasing as expected.
> > > > We also note that some unintuitive behavior persists in certain cases, particularly for the Dynotears algorithm.
> > > > We believe this is an interesting finding and comment on this in **Appendix D.5**

---

> > > > > ### Comment · Reviewer_K4FN · 2025-11-25
> > > > > **Official Comment by Reviewer K4FN**
> > > > >
> > > > > Thank you for the detailed response. Most of the concerns and weaknesses have been addressed well. I have one remaining point regarding the evaluation protocol that I would like to discuss further. As far as I understand, in both [1] and [2], the reported performance corresponds to the best result for each assumption violation and each data regime. This differs from your paper, which reports the highest average robustness (Figure 1 and Figure 9) and the average robustness over all hyperparameters (Table 1). These two evaluation strategies are different, and your approach provides a valuable complementary perspective to prior work. If my understanding is incorrect in any way, please let me know.

---

> ### Author Response · Authors · 2025-11-25
>
> **You are totally correct. Thank you for making this difference clear.**
>
> As we previously discussed, finding the optimal hyperparameters in real-world applications is complicated because there is no ground truth available.
>
> Our protocol highlights a single hyperparameter configuration per CD method for all violations (in addition to the average robustness).
> We believe this scoring is more practically relevant, as it references a unique configuration of a CD method that achieves high overall robustness, as opposed to blending multiple configurations.
>
> Notably, we also report the best hyperparameter configuration per violation in Fig. 17, aiming to bridge the gap between our work and [1] and [2].
>
> With the exception of some violations (e.g., $V_{obs,shock}$), the effects are minor, suggesting that there is typically a single hyperparameter configuration that is relatively reliable in our experiments.
>
> We have slightly rephrased sections 362-365 to place more emphasis on this distinction.

---

> > ### Comment · Reviewer_K4FN · 2025-11-25
> > **Official Comment by Reviewer K4FN**
> >
> > I appreciate the authors' thorough and detailed rebuttals. My concerns have been adequately addressed.
> >
> > In addition, I noticed that some other reviewers expressed a desire for even more data scenarios and additional methods. Having previously published in the area of causal discovery benchmarks myself, I fully understand that pursuing comprehensive evaluation can easily lead to an explosion of experimental combinations. The data regimes and methods considered in the current version are already sufficient to reflect the robustness of time series causal discovery, and the authors' explanations for the experiments they did not include are reasonable. Given that there are numerous possible data settings and methods that could be considered, I do not expect the authors to cover every combination, nor do I view the absence of such exhaustive coverage as a weakness. I am confident that this work represents a valuable contribution to the causal discovery benchmark community.
> >
> > Based on these considerations, I have decided to raise my score and confidence.

---

### Official Review · Reviewer_qeAM · 2025-11-01

**Soundness:** 2
**Presentation:** 3
**Contribution:** 2
**Rating:** 4
**Confidence:** 5

**Summary:**

This paper presents TCD-Arena, a modularized and extendable testing kit designed to assess the robustness of time series causal discovery (CD) algorithms against assumption violations. The authors use this toolkit to conduct a large-scale empirical study, evaluating eight distinct CD methods across 27 different assumption violations, each with stepwise increasing severity. The study involves over 50 million individual CD attempts. The paper also provides an initial investigation into ensembling CD methods, which can improve general robustness. The TCD-Arena toolkit is presented as an open-source package to facilitate future research and comparability.

**Strengths:**

1. The paper's most significant strength is the sheer scale and thoroughness of the empirical study, evaluating 8 CD methods against 27 distinct violations with increasing severity, totaling over 50 million CD attempts.
2. The paper addresses the critical and practical problem of algorithm robustness to assumption violations, which is a known barrier to the adoption of CD methods.
3. The paper provides a novel investigation into ensembling CD methods as a strategy to improve general robustness, which is an under-explored area in the literature.

**Weaknesses:**

1. This level of resource-intensive evaluation is an "overkill" and is not a realistic or practical standard for most researchers to adopt. Instead of clarifying robustness, such a convoluted framework may deter researchers, raise the barrier for reproduction, and make the results more difficult to interpret. The paper does not convincingly argue that this massive complexity offers proportionate benefits over simpler, more targeted robustness tests.
2. The main paper's evaluation relies exclusively on AUROC. This omits other standard and widely-used metrics in the CD literature, such as True Positive Rate (TPR), False Discovery Rate (FDR), and Structural Hamming Distance (SHD). This makes the results difficult to align and compare with many other benchmark studies.
3. The authors rightly concede that the ensemble results are a "theoretical proof-of-concept". The paper does not provide a clear path for how these ensembles could be deployed in a real-world scenario where ground truth is unavailable and domain adaptation is a challenge.
4. The paper restricts some methods, like PCMCI and PCMCI+, to linear conditional independence tests, even though they can handle nonlinear cases. While this is noted in the appendix, it means the study is not evaluating the full capabilities of these specific algorithms.

**Questions:**

1. Could the authors justify the decision to rely on AUROC as the primary metric in the main paper and to exclude common metrics like TPR, FDR, and SHD from the study entirely? These are standard in the field, and their omission is a significant weakness.
2. What was the rationale for the specific selection of the eight algorithms? Were more recent, state-of-the-art methods for time series causal discovery considered?
3. Regarding the ensembles: Beyond the "proof-of-concept," what do the authors see as a realistic path to practical application? How could a meta-learner, trained on this synthetic data, be reliably applied to real-world data where the types and severity of violations are unknown?

---

> ### Author Response · Authors · 2025-11-22
> **We thank the reviewer for their helpful comments.  Below, we will address the noted Questions and Weaknesess**
>
> ---
> **AUROC as the main metric (Weakness 2 and Question 1)**
>
> We chose AUROC initially as the metric, given that it is inherently threshold-free, ensuring that we do not underreport method performance.
> However, we agree that providing additional metrics commonly used in the causal discovery literature would make this study easier to compare with other works.
> Therefore, we integrate SHD, TPR, and FPR as additional metrics. We will likely change SHD as the main metric in the manuscript to further align this study with the literature.
> As this point was also raised by other reviewers, we include a metric comparison for a specific violation in the **general comment** and provide a brief discussion.
>
> ---
> **Method selection (Question 2)**
>
> Yes, we considered several other algorithms, which are partially integrated into TCD-arena  [1].
> Our rationale for algorithm selection for the studies is threefold: 1. We include the most well-established methods (which are typically the original method without further extensions). 2. We prefer algorithms with a reliable implementation available. 3. Many advanced methods are developed to target specific settings, such as extended summary graphs (PCGCE[2]), irregular data (Cuts[3]), or multiple datasets (e.g., J-PCMCI [4]). While an analysis of these methods would be interesting, we wanted to cover the most basic form of CD (a single time series and a single Window causal graph as output) first.
> Despite this, we are currently working on integrating 2 more recent algorithms (LPCMCI [5] and SVAR-RFCI [6]) to further increase the coverage of our study. We include a performance comparison for a violation with particularly pronounced differences in method robustness ($V_{inno, com}$) in the **general comment** and discuss it.
>
> ---
> **Ensembles in real-world scenarios (Weakness 3 and Question 3)**
>
> Thank you for raising this point. We agree that it would be interesting to evaluate how ensembles trained on synthetic data perform on real-world data sources.
> To address this, we evaluated the performance of the trained ensembles on a completely different real-world data source (CausalRivers [7]). We include the current results in the **general comment** and provide a brief discussion of them there.
>
> ---
> **Resource intensiveness (Weakness 1)**
>
> We understand the concern about overkill and would like to address this aspect for two audiences:
>    - For researchers, our work serves as a comprehensive reference, not a mandatory protocol. A new method can be assessed through targeted comparisons against the top performer on specific violations of interest. To facilitate this, we have made TCD-Arena fully seeded, with scripted and hashed data generation, allowing the sharing of exact experimental configurations.
>    - For practitioners, the large search space can be drastically reduced by leveraging domain-specific knowledge about the target application (How much data is accessible? How is data captured? What is known about the data-generating process?). This allows for customized comparisons if needed.
>
> While resource-intensive, we argue that this extensive protocol is necessary to uncover crucial interactive effects between data regimes, hyperparameters, and process size, which simpler evaluations would miss. Note that we include a comparison of computational costs per method in the **general comment** to further assess this.
>
> ---
> **Nonlinear independence tests (Weakness 4)**
>
> While we would have liked to integrate nonlinear independence tests, we initially struggled with the computational overload that these tests require.
> While we can run PCMCI+ with a partial correlation CI test in under a second on a single sample, the official implementation of other tests in the official package (Tigramite [7]) can run for multiple minutes, increasing the computational effort by a hundredfold. We include information on the speed of various tests (for graphs with 5 nodes) and have attached the results below, including the newly integrated methods.
> We, however, agree that, especially for the linearity violations, the appropriate CI test should be leveraged.
> In the second table below, we present results for nonlinear (GPDC) and linear CI-tests for the RBF violation for PCMCI and PCMCI+, on smaller graphs (5 nodes).
> As we find an increase in performance, we are currently working on extending these results to cover all nonlinear violations in an additional study, which will be referenced in the main part of the manuscript.
>
> **Runtimes in seconds per sample (mean ± std)**
> |Method |ParCorr|RobustParCorr|GPDC|CMIknn|
> |---|---|---|---|---|
> |PCMCI|0.67±0.1|1.03±0.1|687.53±128.7|1166.56±340.2|
> |PCMCI+|0.37±0.1|0.53±0.1|305.11±94.1|796.04±159.2|
> |LPCMCI|0.67±0.4|0.87±0.5|629.62±463.4|588.83±463.7|
> |SVAR-RFCI|0.83±0.5|1.27±0.8|1141.08±574.7|684.03±408.3|
>
> **Nonlinear CI-Test comparison (RBF violation)**
> |  | PCMCI | PCMCI+ |
> | :--- | :--- | :--- |
> | GPDC | 0.929882 | 0.936064 |
> | Linear | 0.887269 | 0.889307 |

---

> > ### Author Response · Authors · 2025-11-22
> >
> > [1] https://anonymous.4open.science/r/anonymous_submission-0EF4/README.md
> >
> > [2] Assaad, Charles K., Emilie Devijver, and Eric Gaussier. "Discovery of extended summary graphs in time series." Uncertainty in Artificial Intelligence. Pmlr, 2022.
> >
> > [3] Cheng, Yuxiao, et al. "Cuts+: High-dimensional causal discovery from irregular time-series." Proceedings of the AAAI Conference on Artificial Intelligence. Vol. 38
> >
> > [4] Günther, Wiebke, Urmi Ninad, and Jakob Runge. "Causal discovery for time series from multiple datasets with latent contexts." Uncertainty in Artificial Intelligence. PMLR, 2023.
> >
> > [5] LPCMCI: Gerhardus, A. & Runge, J. High-recall causal discovery for autocorrelated time series with latent confounders Advances in Neural Information Processing Systems, 2020, 33. https://proceedings.neurips.cc/paper/2020/hash/94e70705efae423efda1088614128d0b-Abstract.html
> >
> > [6] https://github.com/jakobrunge/tigramite
> >
> > [7] Stein, Gideon, et al. "CausalRivers-Scaling up benchmarking of causal discovery for real-world time-series." The Thirteenth International Conference on Learning Representations.

---

> > ### Comment · Reviewer_qeAM · 2025-11-24
> >
> > I appreciate the authors' rebuttal. After careful consideration, based on the technical contributions of the paper, I will keep my initial score.

---

### Author Response · Authors · 2025-11-22

---
**We would like to express our sincere gratitude to all reviewers for their feedback and their overall appreciation of the study.**

We provide individual responses to each review to initiate the discussion, but we wanted to address a common theme first.
As the nature of this benchmark invites extensions, and we agree on the value of the proposed experiments with almost no exceptions, we are working on incorporating them into the manuscript.
We think this process also serves to highlight the flexible and extendable nature of TCD-arena.

As multiple extensions require substantial computational efforts to be fully completed, we provide partial results for all requests and are committed to providing comprehensive updates to the manuscript.

Below, we include additional resources that were requested by multiple reviewers.
We also refer to them individually in each response.

---
**1. Computational Costs**

Additional information was highly requested, so we provide the average computational efforts (in seconds) per 100 samples and for a single hyperparameter configuration for each of the 8 CD strategies (measured on our specified hardware). The standard deviations denote differences between Hyperparameter configurations. As Cross Corr has no Method hyperparameters, no standard deviation is provided. Note that we run 8,000 samples per HP combination and violation (See Section 4). We will include this information in the manuscript (C.5).

|Method|Seconds per 100 samples|
|:---|---:|
|CausalPretr|38 ± 2|
|PCMCI|259 ± 85|
|NTS-NOTears|595 ± 203|
|PCMCI+|402 ± 160|
|Varlingam|25 ± 9|
|Dynotears|45 ± 12|
|GVAR|11 ± 0|
|Cross Corr|15†|

---
**2. Ensemble evaluation**

We agree with the reviewers that an out-of-distribution (o.o.d.) evaluation of our ensembling approaches would further improve the empirical protocol. We therefore integrated an assessment of the ensembles on CausalRivers [1], a real-world benchmark for causal discovery. We attached the current results below. Interestingly, the Linear Layer approach seems to be the most robust against a domain shift. We will investigate this further and integrate the results into the manuscript.

|Metric|Linear-Ensemble|MLP-Ensemble|ConvM-Ensemble|Mean-Ensemble|Cross Corr|Dynotears|Var|Varlingam|NTS-Notears|PCMCI|PCMCI+|Pareto selection|
|:---|:---|:---|:---|:---|:---|:---|:---|:---|:---|:---|:---|:---|
|AUROC|**0.720**|0.677|0.675|0.664|0.477|0.561|0.703|0.629|0.474|0.518|0.526|0.720|

---
**3. AUROC and other metrics**

To further align our study with other works, we are currently integrating additional Metrics (normalized Structured Hamming Distance (SHD), FPR, TPR). We have attached a sample for a violation below ($V_{inno, com}$), which compares the ranking between methods based on the metric. We chose this violation as it resulted in large performance gaps in our original experiments. While we find that Varlingam Ranks consistently the highest, we also observe some differences in the ordering depending on the metric. Therefore, we will expand on the discussion of the differences in Appendix D1 and reference it accordingly in the main part. Finally, we will likely switch to the SHD metric as the primary metric in the revised version to align with the literature.

|Metric|CausalPretraining|PCMCI|NTS-Notears|PCMCI+|Varlingam|Dynotears|GVar|Cross Corr|
|:---|:---|:---|:---|:---|:---|:---|:---|:---|
|AUROC|0.892 (7)|0.922 (3)|0.922 (2)|0.900 (5)|0.944 (1)|0.909 (4)|0.892 (6)|0.863 (8)|
| SHD|0.553 (5)|0.464 (3)|0.809 (8)|0.531 (4)|0.308 (1)|0.562 (6)|0.419 (2)|0.605 (7)|
|F1 |0.735 (7)|0.845 (2)|0.818 (4)|0.798 (6)|0.877 (1)|0.835 (3)|0.805 (5)|0.672 (8)|

---
**4. Additional violations**

To further extend TCD-Arena, we are working on integrating four additional violations that were explicitly requested  during the review process:

- Nonstationarity through a changing  causal skeleton ($V_{skeleton}$) (R 3,4)
- Semi-synthetic observational noise that was extracted by applying a high-pass filter to real-world stock prices. ($V_{obs,semi}$) (R 3)
- Near-Faithfulness through small but nonzero effects (R 3,4)
- Data with missing-not-at-random gaps (R 4)

 We provide robustness scores for the first two violations below:

|Normalized SHD|CausalPretraining|PCMCI|NTS-Notears|PCMCI+|Varlingam|Dynotears|GVar|Cross Corr|
|:---|---:|---:|---:|---:|---:|---:|---:|---:|
|$V_{obs,semi}$|0.837|0.874|0.897|0.832|0.777|0.801|0.877|0.890|
|$V_{skeleton}$|2.015|1.986|2.543|2.016|2.061|2.577|1.997|2.117|

---
**5. Additional Methods**

As requested (R 1 3), we are working on integrating two more recent algorithms in our study (LPCMCI, SVARRFCI [2]). Below, we present the results for smaller graphs and the $V_{ino.com} $ violation.


|SHD |Causal Pretraining|Pcmci|NTS-Notears|PCMCI+|Varlingam|Dynotears|**LPCMCI**|**Svarrfci**|GVAR|Corr Corr|
|:---|:---|:---|:---|:---|:---|:---|:---|:---|:---|:---|
|$V_{inno,com}$|0.553|0.365|0.825|0.455|0.278|0.552|0.378|0.371|0.367|0.482|

---

> ### Author Response · Authors · 2025-11-22
>
> [1] Stein, Gideon, et al. "CausalRivers-Scaling up benchmarking of causal discovery for real-world time-series." The Thirteenth International Conference on Learning Representations.
>
> [2] LPCMCI: Gerhardus, A. & Runge, J. High-recall causal discovery for autocorrelated time series with latent confounders Advances in Neural Information Processing Systems, 2020, 33. https://proceedings.neurips.cc/paper/2020/hash/94e70705efae423efda1088614128d0b-Abstract.html

---

### Author Response · Authors · 2025-11-27

**Rebuttal Update: Framework Enhancements**

We thank the reviewers for their numerous suggestions. As mentioned before, we have further integrated any requested features into our study design.

The new key enhancements include:

 - **Unfaithfulness Simulation**: We now model unfaithfulness conditions by incorporating near-zero coefficients.

- **Signal-Dependent Missingness**: The probability of a value being missing depends on the time series signal itself (MNAR).

- **Externally-Driven Missingness**: The missingness mechanism can be controlled by an external, real-world signal (MAR).

 - **Realistic Innovation Noise**: We have added the capability to sample innovation noise by relying on real-world time series.



|   normalized SHD  |    CausalPretraining | PCMCI | NTS-Notears | PCMCI+ | Varlingam | Dynotears |   GVAR | Cross Corr |
|:------------|------:|------:|-----------:|----------:|----------:|----------:|------:|-----------------:|
| MAR         | 0.795 | 0.849 |      0.907 |     0.839 |     0.759 |     0.800 | 0.856 |            0.825 |
| $\text{Innovation}_{real}$ | 0.430 | 0.452 |      0.774 |     0.414 |     0.136 |     0.863 | 0.457 |            0.525 |
| MNAR        | 0.865 | 0.895 |      0.904 |     0.892 |     0.868 |     0.882 | 0.895 |            0.871 |
| $\text{Faith}_{zero}$   | 0.451 | 0.475 |      0.910 |     0.447 |     0.400 |     0.680 | 0.466 |            0.470 |


Notably, **TCD-arena now supports multiple procedures for semi-synthetic time-series**.

We also updated the anonymous repo to reflect these changes:
https://anonymous.4open.science/r/anonymous_submission-0EF4


**We remain available for the rest of the rebuttal phase and would be pleased to implement any further requested extensions** to demonstrate the flexibility and extendability of TCD-Arena.

---

### Meta-Review · Area_Chair_o2py · 2025-12-19

**Summary:**

This paper proposes a benchmark for causal discovery in time series, specifically focused to robustness to assumption violation. While there are other benchmarks for causal discovery, including in the time series setting, this paper argues that their fully synthetic nature allows to specifically test a single aspect of the methods in a controlled way, which is an approach I like.

I would recommend the authors to try to take a closer look at their result and try to understand why things work the way they do. Assumptions violations in causality often render the setup non-identifiable statistically. There, any non-random performance should be judged with suspicion, potentially stemming from shortcuts on the generative model.  The stepwise analysis is very nice, assumptions are often not on/off, which is usally ignored in other benchmarks.

Overall, this paper fills a niche area in the community and I would recommend acceptance. I strongly recommend that the authors take a closer look at their data-generating process for shortcuts under assumption violations. This is not to say that this is an issue with their synthetic data setup (real data could have the same issues).

**Reviewer Concerns:**

Reviwers had concerns on the breadth of the experimental setup, which I don't share. Being this a benchmark paper, it's nice to see that the authors propose a comprehensive evaluation. I think future causal discovery approaches should consider these robustness tests. As real world data is often scarce in causality, this type of stress-testing can be quite insightful in the development of new algorithms (e.g., even if there is an identifiability result, it is not obvious that the setting is easy to estimate statistically). Other concerns about adding metrics and methods were well addressed.

**Reviewer Scores:**

In the discussion, two reviewers changed their score and one changed the confidence, in light of the new experiments. As mentioned above, I don't share the concerns of qeAM and disagree with B3QT that data sets like causal chambers are a better stress test for causal methods (the fact that it is a physical device does not imply it is representative or real-world problems either, i.e., that rankings on causal chambers transfer to other data). It's possible they would have changed their scores; otherwise, I would have raised my points in the discussion.

---

### Decision · Program_Chairs · 2026-01-26

Accept (Poster)